# Distribution-Specific Agnostic Conditional Classification With Halfspaces

**Jizhou Huang, Brendan Juba**
Washington University in St. Louis
{huang.jizhou,bjuba}@wustl.edu

## Abstract

We study "selective" or "conditional" classification problems under an agnostic setting. Classification tasks commonly focus on modeling the relationship between features and categories that captures the vast majority of data. In contrast to common machine learning frameworks, conditional classification intends to model such relationships only on a subset of the data defined by some selection rule. Most works on conditional classification either solve the problem in a realizable setting or do not provide error guarantees compared to an optimal solution. In this work, we consider conditional classification by sparse linear classifiers for subsets defined by halfspaces. We give both positive and negative results for Gaussian feature distributions. On the positive side, we present the first PAC-learning algorithm for homogeneous halfspace selectors with error guarantee $\tilde{O}(\sqrt{\mathrm{opt}})$, where $\mathrm{opt}$ is the optimal conditional 0-1 loss over the given class of classifiers and homogeneous halfspaces. On the negative side, we find that, under cryptographic assumptions, approximating the conditional loss within a small additive error is computationally hard even under Gaussian distribution. We prove that approximating conditional classification is at least as hard as approximating agnostic classification in both additive and multiplicative form.

## 1 Introduction

Classification is the task of modeling the relationship between some features and membership in some category. Classification tasks are common across various fields, such as spam detection (classifying emails as "spam" or "not spam"), and image recognition (identifying objects like "cat" or "dog"). Standard classification approaches seek to model the whole data distribution. By contrast, we consider the problems where a **better classifier** exists on a subset of the data. In particular, we will consider cases in which classifiers are sparse linear functions (or more generally, any small set of functions), and subsets are described by **selector** functions, given here by homogeneous halfspaces.

We study the distribution-specific PAC-learnability (Kearns et al., 1994) of the classifier-selector class in the presence of adversarial label noise. In the literature, this problem is known as "conditional" classification, but also part of a family that is generally known as "selective" classification.

### 1.1 Background and Motivation

The first "selective classification" problem was introduced decades ago (Chow, 1957; 1970). The focus was on finding Bayes classifiers for the case where the data distribution is fully known. The appeal of effective selective classification is clear in applications where partial domain coverage is acceptable, or in scenarios where achieving extremely low risk is essential but unattainable with standard classification methods. Classification tasks in medical diagnosis and bioinformatics often fall into this category (Khan et al., 2001; Hanczar & Dougherty, 2008).

El-Yaniv et al. (2010) gave a thorough theoretical analysis for selective classification based on a "risk-coverage" model. They proved that, for the optimal classifier and selector, there exists a natural trade-off between the performance of the classifier on the selected subset and the size of the subset.

Prior work has either considered the "realizable" case (El-Yaniv & Wiener, 2012; Gangrade et al., 2021), where there exists a classifier-selector pair that does not make any errors, or endowed the learner with a rejection mechanism using heuristic rules or confidence scores (Geifman & El-Yaniv, 2017; Pugnana & Ruggieri, 2023). For the "agnostic" case, where no perfect classifier-selector pair exists, few works had been done on model-based selective learning (Wiener & El-Yaniv, 2011; 2015; Gelbhart & El-Yaniv, 2019). More importantly, these works do not guarantee both computational efficiency together with good performance with respect to the optimal classifier and selector.

We consider a more general formulation of **agnostic selective classification** under the PAC-learning semantics in Definition 1.1. In particular, we do not make any assumptions on the labels while the performance of the learned classifier and selector are guaranteed to be close to the optimal solution.

**Definition 1.1** (Agnostic Conditional Classification). *Let $\mathcal{D}$ be any distribution on $\mathbb{R}^d \times \{0,1\}$, $\mathcal{C}$ be a finite class of classifiers on $\mathbb{R}^d \times \{0,1\}$, and $\mathcal{G}_{\mathcal{D}}^{a,b} = \{S \subseteq \mathbb{R}^d \mid \Pr_{\mathbf{x} \sim \mathcal{D}_{\mathbf{x}}}\{\mathbf{x} \in S\} \in [a,b]\}$. For any $0 \le a \le b \le 1$, suppose $\min_{S \in \mathcal{G}_{\mathcal{D}}^{a,b}, c \in \mathcal{C}} \Pr_{(\mathbf{x},\mathbf{y}) \sim \mathcal{D}}\{y \ne c(\mathbf{x}) \mid \mathbf{x} \in S\} \le \text{opt}$, a $C$-approximate algorithm (or an algorithm with approximation factor $C$) for some $C > 1$, given sample access to $\mathcal{D}$, outputs an $S' \in \mathcal{G}_{\mathcal{D}}^{a,b}$ and $c \in \mathcal{C}$ such that $\Pr_{\mathcal{D}}\{y \ne c(\mathbf{x}) \mid \mathbf{x} \in S'\} \le C\text{opt}$ with high probability.*

The imposed "population" bounds on the subsets $S \in \mathcal{G}_{\mathcal{D}}^{a,b}$ are critical. On the one hand, the lower bound, $\Pr\{\mathbf{x} \in S\} \ge a$ can both prevent trivial optimal solutions such as $S' = \varnothing$ and make the selected subsets statistically meaningful. On the other hand, if the selector chooses a majority of the data, the performance advantage of the optimal solution of selective classification could vanish compared with that of the regular classification model (El-Yaniv et al., 2010; Hainline et al., 2019).

Consider a halfspace $h$, i.e., a subset of $\mathbb{R}^d$ such that the membership in $h$ is defined by some linear threshold function. In this work, we wish to solve the problem of agnostic conditional classification with halfspace selectors under standard normal distributions described as follows.

**Problem 1.2** (Distribution-Specific Agnostic Conditional Classification With Halfspaces). *Let $\mathcal{D}$ be any distribution on $\mathbb{R}^d \times \{0,1\}$ with standard normal $\mathbf{x}$-marginal on $\mathbb{R}^d$, $\mathcal{C}$ be a finite class of classifiers on $\mathbb{R}^d \times \{0,1\}$, and $\mathcal{H}_{\mathcal{D}}^{a,b}$ be the class of halfspaces on $\mathbb{R}^d$ with population size in the range of $[a,b]$ over $\mathcal{D}_{\mathbf{x}}$. For any $0 \le a \le b \le 1$, suppose $\min_{h \in \mathcal{H}_{\mathcal{D}}^{a,b}, c \in \mathcal{C}} \Pr_{(\mathbf{x},y) \sim \mathcal{D}}\{y \ne c(\mathbf{x}) \mid \mathbf{x} \in h\} \le \text{opt}$, how close to $\text{opt}$ can a polynomial-time algorithm achieve on $\mathcal{H}_{\mathcal{D}}^{a,b}$ with high probability?*

An algorithm for Problem 1.2 may be leveraged to perform conditional classification for large or infinite classes $\mathcal{C}$ by using an algorithm for list learning of classifiers for some richer class (Charikar et al., 2017), taking $\mathcal{C}$ in Problem 1.2 to be the list of classifiers produced by the list learning algorithm:

**Definition 1.3** (Robust list learning). *Let $\mathcal{D} = \alpha \mathcal{D}^* + (1 - \alpha)\tilde{\mathcal{D}}$ for an inlier distribution $\mathcal{D}^*$ and outlier distribution $\tilde{\mathcal{D}}$ each supported on $\mathbb{R}^d \times \{0,1\}$, with $\alpha \in (0,1)$. A robust list learning algorithm for a class of Boolean classifiers $\mathcal{C}$, given $\alpha$ and parameters $\epsilon, \delta \in (0,1)$, and sample access to $\mathcal{D}$ such that for $(\mathbf{x}, b)$ in the support of $\mathcal{D}^*$, $b = c^*(\mathbf{x})$ for some $c^* \in \mathcal{C}$, runs in time $\text{poly}(d, \frac{1}{\alpha}, \frac{1}{\epsilon}, \log \frac{1}{\delta})$, and with probability $1 - \delta$ returns a list of $\ell = \text{poly}(d, \frac{1}{\alpha}, \frac{1}{\epsilon}, \log \frac{1}{\delta})$ classifiers $\{h_1, \ldots, h_\ell\}$ such that for some $h_i$ in the list, $\Pr_{\mathcal{D}^*}\{c_i(\mathbf{x}) = c^*(\mathbf{x})\} \ge 1 - \epsilon$.*

As we will review, it is known in particular that, for sparse linear classifiers (with $s = O(1)$ nonzero coefficients), list learning from a sample of size $m = O(\frac{1}{\alpha\epsilon}(s \log d + \log \frac{1}{\delta}))$ is possible in time and list size $O(md^s)$ (Juba, 2017; Mossel & Sudan, 2016).

## 1.2 CHALLENGES OF DISTRIBUTION-SPECIFIC CONDITIONAL CLASSIFICATION

Problem 1.2 is similar to **agnostic linear classification**, where we seek to minimize the classification error over the vast majority of data. In particular, it was clear that agnostic classification can be reduced to (distribution-free) conditional learning (Juba, 2017). Agnostic linear classification has been extensively studied over decades, and it is known to be computationally hard in both distribution-free (Kearns et al., 1994) and distribution-specific settings (Diakonikolas et al., 2023).

Despite the intractability of agnostic learning, numerous distribution-specific approximation schemes have been developed with approximation factor of $O(1/\sqrt{\text{opt}})$ or even constants (Frei et al., 2021; Diakonikolas et al., 2020c; 2022; 2024; Shen, 2021). Given the similarity between agnostic linear

classification and Problem 1.2, and that it was the only formal barrier known, it is natural to ask if we can leverage the existing techniques for standard agnostic classification in conditional classification.

However, it is not clear how these could lead to a meaningful error guarantee for conditional classification. Directly, Definition 1.1 (correspondingly, Problem 1.2) can be reduced to a "one-sided" classification problem, where we seek to minimize the error rate of the classifier on only the selected halfspace (one side). As the error rate could be extremely unbalanced across the selected side and its complement, a constant factor approximation scheme for the agnostic linear classification problem may not yield approximation guarantees for the one-sided agnostic classification problem.

An analogous difficulty arose in "fairness auditing" (Kearns et al., 2018). In the problem of fairness auditing, instead of minimizing the classification error, we wish to verify some specific fairness criteria for a subset of the data. Kearns et al. (2018) showed that the auditing problem is equivalent to agnostic classification for any simple representation classes (including halfspaces) under distribution-free settings. Despite the similarity between these two problems, as well as the existence of constant factor approximation algorithms for agnostic linear classification under distributional assumptions, recent work by Hsu et al. (2024) showed there does not exist any nontrivial multiplicative factor approximation algorithm for auditing halfspace subgroup fairness even under Gaussian distributions. The connection in the distribution-free setting simply does not carry over to Gaussian data.

## 1.3 OUR CONTRIBUTION

Let opt be as defined in Problem 1.2 for $\mathcal{H}$ being the class of **homogeneous** halfspaces. Our first contribution is a polynomial-time $\tilde{O}(1/\sqrt{\text{opt}})$-approximation algorithm to learn a pair of classifier and selector for Problem 1.2 with homogeneous halfspace selectors for finite $\mathcal{C}$ (cf. Theorem 3.1). We also generalized our approach to work with any sparse linear classifiers (cf. Theorem 3.5). This is the first polynomial-time algorithm for agnostic conditional/selective classification with a provable approximation guarantee w.r.t. the optimal solution.

*Remark* 1. Even for homogeneous halfspace selectors, the imbalance of error rates between classes could still exist, as we will show in our hardness result that the difference between the error rates of different classes of the homogeneous halfspace always equals to the amount that the probability of either label deviates from $1/2$; see Lemma 4.4 for details.

*Remark* 2. Note that the extension of our approach on sparse classes leverages the power of an list learning algorithm for linear sparse classifiers (cf. Theorem A.1). It is an open question whether our method can be extended to other families of classifiers. Polynomial-time algorithms for list-decodable (dense) linear regression (as opposed to classification) have been found (Karmalkar et al., 2019; Raghavendra & Yau, 2020), so as other list-decodable estimation tasks as mentioned earlier, but little is known for learning classifiers at present. It is an interesting broader open question which families of classifiers are learnable in the list-decodable setting.

Our second contribution is a negative result for Problem 1.2. We show that agnostic conditional classification in Definition 1.1 is at least as hard as agnostic linear classification under any distribution (cf. Proposition 4.5). With the distribution-specific hardness result of agnostic linear classification (Diakonikolas et al., 2023), we prove that no polynomial-time algorithm can achieve an error guarantee of $\text{opt} + O(1/\log^{1/2+\alpha} d)$ for any constant $\alpha > 0$ for Problem 1.2 (cf. Theorem 4.3). We show more generally that approximating the conditional classification objective is at least as hard as approximating the regular classification objective (cf. Proposition 4.5 & Claim 4.7).

*Remark* 3. We believe our upper bound $\tilde{O}(\sqrt{\text{opt}})$ is sub-optimal. Obviously, it is consistent with the known negative results, even for general halfspaces. Meanwhile, it was possible to reach $O(\text{opt})$ error for agnostic learning of halfspaces w.r.t. Gaussian data (Diakonikolas et al., 2022). Also, in other, simpler list-decodable robust estimation tasks (Kothari et al., 2018), it was possible to use $k$th moments to obtain $\text{opt}^{1-O(1/k)}$ error, and for list-decodable linear regression it is similarly possible to approach optimal error by spending more on the running time (Bakshi & Kothari, 2021). Thus, there does not seem to be a hard barrier at some suboptimal error rate.

**Organization.** In Section 2, we give some necessary background. We will present our algorithmic results in Section 3. The distribution-specific hardness result for conditional classification with general halfspaces is in Section 4. In the last section, we will discuss the limitations of our results and a few possible directions for extensions.

### 1.4 RELATED WORKS

**Selective Learning.** For realizable setting, El-Yaniv & Wiener (2012) reduced active learning to selective learning and proved a exponential lower bound on label complexity for linear classification . Gangrade et al. (2021) proposed a optimization-based selective learning framework that guarantees to maximize the classifiers' coverage with a specified one-side prediction error rate. They proved that any representation class with finite VC-dimension can be used successfully in their models. For the agnostic cases, Wiener & El-Yaniv (2011; 2015); Gelbhart & El-Yaniv (2019) presented a selective learning approach to learn a classifier-selector pair that is at least as competitive as the ERM of the non-selective learning task. However, the computation of both the classifier and selector in these methods relies on an agnostic learning oracle, and the selector function is not guaranteed to minimize the conditional classification error down to any approximation factor. Geifman & El-Yaniv (2017) proposed a heuristic method to design selector functions for any given deep neural network with provable guarantees. Empirically, Pugnana & Ruggieri (2023) developed an model-agnostic learning algorithm to learn a confidence-based selective classifier that seeks to minimize the AUC-based loss within the selected region. Geifman & El-Yaniv (2019) proposed the SelectiveNet architecture that simultaneously learns a pair of classifier and selector in a single neural networks with given coverage.

**Conditional Learning.** The problem of conditional learning (including conditional classification) incorporates two sub-problems, obtaining a finite list of predictors as well as learning a predictor-selector pair out this finite list and some class of selector functions. For the former task, a series of positive results (Charikar et al., 2017; Kothari et al., 2018; Calderon et al., 2020; Bakshi & Kothari, 2021) have been obtained under the "list-decodable" setting of Definition 1.3. For the latter task, Juba (2016) introduced the problem of learning abduction, where they propose to learn a subset of the data distribution where e.g., no errors occur. In their work, they showed that subsets defined by $k$-DNFs can be efficiently learned in realizable cases without any distributional assumptions. Subsequent improvements were obtained for the agnostic setting (Zhang et al., 2017; Juba et al., 2018).

**Learning To Abstain.** Cortes et al. (2016) considered a different formulation of selective classification. Instead of optimizing the classification error conditioned the selected subgroup, they proposed to minimize the classification error jointly with the selector function while enforcing a cost for "abstaining". They designed a few convex surrogate losses to upper bound the joint classification loss in the setting that abstaining has a cost. Later works (Mao et al., 2024c;a;b) proposed new families of surrogate losses to approximate the classification loss with abstaining and proved various upper bounds classification error of any classifier-selector pair in terms of different surrogate loss measures for two different selective learning strategies.

## 2 PRELIMINARIES

In general, we use lowercase italic font characters to represent scalars, e.g. $x \in \mathbb{R}$, lowercase bold italic font characters to represent vectors, e.g. $\boldsymbol{x} \in \mathbb{R}^d$, and uppercase bold italic font characters to represent matrix, e.g. $\boldsymbol{A} \in \mathbb{R}^{m \times d}$. In particular, subscripts will be used to index the coordinates of any vector, e.g., $x_i$ represents the $i$th coordinate of the vector $\boldsymbol{x}$. Similarly, we use normal font characters to represent random variables, e.g. $\mathrm{x} \in \mathbb{R}$, $\mathbf{x} \in \mathbb{R}^d$ and $\mathbf{A} \in \mathbb{R}^{m \times d}$.

For $\mathbf{x} \in \mathbb{R}^d$, let $\|\mathbf{x}\|_p = (\sum_{i=1}^d |\mathrm{x}_i|^p)^{1/p}$ denote the $l_p$-norm of $\mathbf{x}$, and $\bar{\mathbf{x}} = \mathbf{x}/\|\mathbf{x}\|_2$ denote the normalized vector of $\mathbf{x}$. We will use $\langle \mathbf{x}, \boldsymbol{u} \rangle$ to represent the inner product of $\mathbf{x}, \boldsymbol{u} \in \mathbb{R}^d$ and $\mathbf{x}^{\otimes k}$ to represent the outer product of $\mathbf{x} \in \mathbb{R}^d$ to the $k$th degree. Additionally, we will use $\theta(\boldsymbol{u}, \boldsymbol{w})$ to denote the angle between two vectors $\boldsymbol{u}, \boldsymbol{w} \in \mathbb{R}^d$. For any subspace $V \subseteq \mathbb{R}^d$, let $\mathbf{x}_V$ denote the projection of $\mathbf{x}$ onto $V$. Further, we will write $\boldsymbol{w}^\perp = \{\boldsymbol{u} \in \mathbb{R}^d \mid \langle \boldsymbol{u}, \boldsymbol{w} \rangle = 0\}$ as the orthogonal space of $\boldsymbol{w} \in \mathbb{R}^d$, and, therefore, $\mathbf{x}_{\boldsymbol{w}^\perp} = (I - \bar{\boldsymbol{w}}^{\otimes 2})\mathbf{x}$ as the projection of $\mathbf{x} \in \mathbb{R}^d$ onto $\boldsymbol{w}^\perp$.

We use $\mathcal{D}_\mathbf{x}$ to denote the marginal distribution of $\mathcal{D}$ on $\mathbf{x} \in \mathbb{R}^d$, $\mathrm{Pr}_{\mathbf{z} \sim \mathcal{D}}\{\mathbf{z} \in E\}$ to denote the probability of an event $E$, $\mathbb{E}_{\mathbf{z} \sim \mathcal{D}}[f(\mathbf{z})]$ to denote the expectation of some statistic $f(\mathbf{z})$, and therefore, $\hat{\|}f(\mathbf{z})\hat{\|}_p = (\mathbb{E}_{\mathbf{z} \sim \mathcal{D}}[\|f(\mathbf{z})\|_p^p])^{1/p}$. In particular, for an i.i.d. sample $\hat{\mathcal{D}} \sim \mathcal{D}$, we define the empirical probability and expectation as $\mathrm{Pr}_{\mathbf{z} \sim \hat{\mathcal{D}}}\{\mathbf{z} \in E\} = 1/|\hat{\mathcal{D}}| \sum_{\mathbf{z} \in \hat{\mathcal{D}}} \mathbb{1}\{\mathbf{z} \in E\}$ and $\mathbb{E}_{\mathbf{z} \sim \hat{\mathcal{D}}}[f(\mathbf{z})] = 1/|\hat{\mathcal{D}}| \sum_{\mathbf{z} \in \hat{\mathcal{D}}} f(\mathbf{z})$. In particular, let $\mathcal{N}^d(0, 1)$ denotes the $d$-dimensional standard normal distribution. We may drop $\mathcal{D}$ from the subscript when it the context is clear, i.e., we may write $\mathrm{Pr}\{\mathbf{z} \in E\}, \mathbb{E}[f(\mathbf{z})]$ for $\mathrm{Pr}_{\mathbf{z} \sim \mathcal{D}}\{\mathbf{z} \in E\}, \mathbb{E}_{\mathbf{z} \sim \mathcal{D}}[f(\mathbf{z})]$.

**In this paper, we denote halfspaces as a subset of $\mathbb{R}^d$ in the following way.** For any $t \in \mathbb{R}$, $\boldsymbol{w} \in \mathbb{R}^d$, let $l_t : \mathbb{R}^d \to \mathbb{R}$ be an affine function such that $l_t(\mathbf{x}, \boldsymbol{w}) = \langle \mathbf{x}, \boldsymbol{w} \rangle - t$. Then, a halfspace in $\mathbb{R}^d$ with threshold $t \in \mathbb{R}$ and normal vector $\boldsymbol{w}$ is defined as $h_t(\boldsymbol{w}) = \{\mathbf{x} \in \mathbb{R}^d \mid l_t(\mathbf{x}, \boldsymbol{w}) \geq 0\}$ (resp. $h_t^c(\boldsymbol{w}) = \{\mathbf{x} \in \mathbb{R}^d \mid l_t(\mathbf{x}, \boldsymbol{w}) \leq 0\}$). When a halfspace is **homogeneous**, we will drop the threshold from the subscript, i.e., when $t = 0$, we will write $h(\boldsymbol{w})$ instead of $h_0(\boldsymbol{w})$.

## 3 CONDITIONAL CLASSIFICATION WITH HOMOGENEOUS HALFSPACES

In this section, we present our algorithmic results for conditional classification with **homogeneous halfspaces (selectors)** on $\mathbb{R}^d$ for sparse linear classifiers or, more generally **any small set of binary classifiers** $\mathcal{C}$ under any distribution $\mathcal{D}$ with **standard normal x-marginals**.

In the case that $\mathcal{C}$ is finite, we find a homogeneous halfspace as the selector that minimizes its conditional classification loss, $\Pr\{c(\mathbf{x}) \neq y \mid \mathbf{x} \in h(\boldsymbol{w})\}$, for each classifier $c \in \mathcal{C}$. Eventually, we choose the best classifier-selector pair as the output. To extend to the case of any sparse linear classes, our strategy is to generate a finite $\mathcal{C}$ using a list-learning algorithm, then run our conditional learning algorithm on the obtained $\mathcal{C}$ to find a classifier-selector pair.

Notice that, for homogeneous halfspaces under Gaussian x-marginals, minimizing $\Pr\{c(\mathbf{x}) \neq y \mid \mathbf{x} \in h(\boldsymbol{w})\}$ equals to minimizing $\Pr\{c(\mathbf{x}) \neq y \cap \mathbf{x} \in h(\boldsymbol{w})\}$ since every homogeneous halfspace satisfies $\Pr_{\mathbf{x} \sim \mathcal{N}^d(0,1)}\{\mathbf{x} \in h(\boldsymbol{w})\} = 1/2$. Hence, we will only consider minimizing $\Pr\{c(\mathbf{x}) \neq y \cap \mathbf{x} \in h(\boldsymbol{w})\}$ in this section. The core challenge for our strategy is finding such a halfspace for each $c \in \mathcal{C}$. We give the details in the following sections.

### 3.1 ALGORITHM OVERVIEW

---

**Algorithm 1:** Conditional Classification For Finite $\mathcal{C}$

---
1   **procedure** CCFC$(\mathcal{D}, \mathcal{C}, \epsilon, \delta)$
2      $T \leftarrow 12d^2 \ln(8 \,|\mathcal{C}|\, /\delta)/\epsilon^4$
3      $N \leftarrow O(\ln(16T \,|\mathcal{C}|\, /\delta)/\epsilon^2 \ln \epsilon^{-1})$
4      $\hat{\mathcal{D}} \leftarrow \ln(4 \,|\mathcal{C}|\, T/\delta)/2\epsilon$ i.i.d. examples from $\mathcal{D}$
5      $\boldsymbol{w}^{(0)} \leftarrow$ any basis
6      **for** $c \in \mathcal{C}$ **do**
7          $\mathcal{D}^{(c)} \leftarrow \mathcal{D}_{\mathbf{x}} \times \mathbb{1}\{c(\mathbf{x}) \neq y\}$
8          $\mathcal{W}^{(c)} \leftarrow$ PSGD $\left(\mathcal{D}^{(c)}, T, N, \boldsymbol{w}^{(0)}\right) \,\cup\,$ PSGD $\left(\mathcal{D}^{(c)}, T, N, -\boldsymbol{w}^{(0)}\right)$
9          $\boldsymbol{w}^{(c)} \leftarrow \operatorname{argmin}_{\boldsymbol{w} \in \mathcal{W}^{(c)}} \Pr_{\hat{\mathcal{D}}}\{\mathbf{x} \in h(\boldsymbol{w}) \cap c(\mathbf{x}) \neq y\}$
10      **end**
11      **return** $\operatorname{argmin}_{\boldsymbol{w}^{(c)}} \Pr_{\hat{\mathcal{D}}} \left\{\mathbf{x} \in h(\boldsymbol{w}^{(c)}) \cap c(\mathbf{x}) \neq y\right\}$

---

In Algorithm 1, for each binary classifier $c \in \mathcal{C}$, we map the label y from $\mathcal{D}$ to $\mathbb{1}\{c(\mathbf{x}) \neq y\}$ to form a new distribution $\mathcal{D}^{(c)}$, then pass $\mathcal{D}^{(c)}$ to Algorithm 2 to obtain a sequence of halfspaces, and only keep $h(\boldsymbol{w}^{(c)})$ with the smallest empirical conditional 0-1 loss for this classifier $c$. At last, it returns the classifier-selector pair that performs the best among all $c \in \mathcal{C}$ estimated on $\hat{\mathcal{D}}$.

Notably, the mapping step (line 7) for each $c \in \mathcal{C}$ essentially just creates another adversarial distribution $\mathcal{D}^{(c)}$, which is a key step to reduce the conditional classification problem to a **"one-sided" agnostic linear classification** problem. While directly optimizing over the conditional classification loss $\Pr\{\mathbf{x} \in h(\boldsymbol{w}) \cap c(\mathbf{x}) \neq y\}$ is intractable in general, it turns out that an **approximately stationary point** of a simple convex surrogate approximation to the classification loss, $\mathcal{L}_{\mathcal{D}}(\boldsymbol{w})$, suffices to approximately capture an optimal halfspace selector under Gaussian distributions. Formally, we define $\mathcal{L}_{\mathcal{D}}(\boldsymbol{w}) = \mathbb{E}_{(\mathbf{x}, y) \sim \mathcal{D}}[y \cdot \max(0, \langle \mathbf{x}, \boldsymbol{w} \rangle)]$ with respect to the distribution $\mathcal{D}$.

Algorithm 2 is inspired by Diakonikolas et al. (2020b) that the gradient step (line 5) uses the projected gradient $g_{\boldsymbol{w}}(\mathbf{x}, y)$, defined as $g_{\boldsymbol{w}}(\mathbf{x}, y) = y \cdot \mathbf{x}_{\boldsymbol{w}^\perp} \mathbb{1}\{\mathbf{x} \in h(\boldsymbol{w})\}$. We will show in the next section that Algorithm 2 is guaranteed to converge to an approximately stationary point, i.e., small $\|\mathbb{E}[g_{\boldsymbol{w}}]\|_2$.

---

**Algorithm 2:** Projected SGD for $\mathcal{L}_\mathcal{D}(\boldsymbol{w})$

---

1 **procedure** PSGD$(\mathcal{D}, T, N, \boldsymbol{w}^{(0)})$
2    $\beta \leftarrow \sqrt{1/Td}$
3    **for** $i = 1, \dots, T$ **do**
4       $\hat{\mathcal{D}}^{(i)} \leftarrow N$ i.i.d. samples from $\mathcal{D}$
5       $\boldsymbol{u}^{(i)} \leftarrow \boldsymbol{w}^{(i-1)} - \beta\,\mathbb{E}_{(\mathbf{x},\mathrm{y})\sim\hat{\mathcal{D}}^{(i)}}[g_{\boldsymbol{w}^{(i-1)}}(\mathbf{x},\mathrm{y})]$
6       $\boldsymbol{w}^{(i)} \leftarrow \boldsymbol{u}^{(i)}/\|\boldsymbol{u}^{(i)}\|_2$
7    **end**
8    **return** $(\boldsymbol{w}^{(1)}, \dots, \boldsymbol{w}^{(T)})$

---

Note that the objective function considered in Diakonikolas et al. (2020b) is completely different from ours so that their convergence analysis does not obviously hold for our surrogate loss $\mathcal{L}_\mathcal{D}(\boldsymbol{w})$. Also, our choice of $g_{\boldsymbol{w}}(\mathbf{x}, \mathrm{y})$ is similar to that of Shen (2021). Nonetheless, the problem they were solving is agnostic linear classification and they used a quite different gradient descent policy.

---

**Algorithm 3:** Conditional Classification For Sparse Linear $\mathcal{C}$

---

1 **procedure** CCSLC$(\mathcal{D}, \epsilon, \delta, m)$
2    $\mathcal{C} \leftarrow$ SPARSELIST$(\mathcal{D}, m)$
3    **return** CCFC$(\mathcal{D}, \mathcal{C}, \epsilon, \delta)$

---

Algorithm 3 solves conditional learning of sparse linear classifiers. Specifically, SPARSELIST (cf. Algorithm 4) generates a list of sparse linear classifiers, at least one of which is guaranteed to be close to a minimizer of the conditional classification error with homogeneous halfspace selectors (cf. Theorem A.1). Run Algorithm 1 on the obtained $\mathcal{C}$ gives the optimal classifier-selector pair.

## 3.2 CONDITIONAL CLASSIFICATION FOR FINITE CLASSES

We introduce our main guarantee at first, but postpone the proof to Appendix C due to the page limit. As a sketch of the proof, we will see Proposition 3.2 and Proposition 3.3 together indicate the optimality of Projected SGD, as captured by Lemma 3.4. Combined with a standard concentration analysis, this implies our main theorem.

**Theorem 3.1** (Main Theorem). *Let $\mathcal{D}$ be a distribution on $\mathbb{R}^d \times \{0, 1\}$ with standard normal $\mathbf{x}$-marginal, and $\mathcal{C}$ be a class of binary classifiers on $\mathbb{R}^d \times \{0, 1\}$. If there exists a unit vector $\boldsymbol{v} \in \mathbb{R}^d$ and a $c \in \mathcal{C}$ such that, for some sufficiently small $\epsilon \in [0, 1/e]$, $\Pr_{(\mathbf{x},\mathrm{y})\sim\mathcal{D}}\{\mathbf{x} \in h(\boldsymbol{v}) \cap c(\mathbf{x}) \neq \mathrm{y}\} \leq \epsilon$, then, with at most $\tilde{O}(d^2/\epsilon^6)$ examples, Algorithm 1 will return a $\boldsymbol{w}^{(c')}$, with probability at least $1 - \delta$, such that $\Pr_{(\mathbf{x},\mathrm{y})\sim\mathcal{D}}\{\mathbf{x} \in h(\boldsymbol{w}^{(c')}) \cap c'(\mathbf{x}) \neq \mathrm{y}\} = \tilde{O}(\sqrt{\epsilon})$ and run in time $O(d^2\,|\mathcal{C}|\,/\epsilon^6)$.*

The most important component that enables our approach is the following proposition, which states that, for any sub-optimal halfspace $h(\boldsymbol{w})$, the projected negative gradient $\mathbb{E}[-g_{\boldsymbol{w}}]$ of the surrogate loss $\mathcal{L}_\mathcal{D}(\boldsymbol{w})$ must have non-negligible projection on the normal vector of the optimal halfspace $h(\boldsymbol{v})$.

**Proposition 3.2.** *Let $\mathcal{D}$ be a distribution on $\mathbb{R}^d \times \{0, 1\}$ with standard normal $\mathbf{x}$-marginal, and $g_{\boldsymbol{w}}(\mathbf{x}, \mathrm{y}) = \mathrm{y} \cdot \mathbf{x}_{\boldsymbol{w}\perp} \mathbb{1}\{\mathbf{x} \in h(\boldsymbol{w})\}$. Suppose $\boldsymbol{v}, \boldsymbol{w} \in \mathbb{R}^d$ are unit vectors such that $\theta(\boldsymbol{v}, \boldsymbol{w}) \in [0, \pi/2)$ and $\Pr_{(\mathbf{x},\mathrm{y})\sim\mathcal{D}}\{\mathbf{x} \in h(\boldsymbol{v}) \cap \mathrm{y} = 1\} \leq \epsilon$, then, if $\Pr_{(\mathbf{x},\mathrm{y})\sim\mathcal{D}}\{\mathbf{x} \in h(\boldsymbol{w}) \cap \mathrm{y} = 1\} \geq \frac{5}{2}(\epsilon\sqrt{\ln \epsilon^{-1}})^{1/2}$, we have $\langle \mathbb{E}_{(\mathbf{x},\mathrm{y})\sim\mathcal{D}}[-g_{\boldsymbol{w}}(\mathbf{x},\mathrm{y})], \bar{\boldsymbol{v}}_{\boldsymbol{w}\perp} \rangle \geq \frac{2}{5}\epsilon\sqrt{\ln \epsilon^{-1}}$ for sufficiently small $\epsilon$.*

We leave the formal proof to Appendix C due to the page limit. The proof is based on the following observation (also see Figure 1): When a homogeneous halfspace $h(\boldsymbol{w})$ is substantially sub-optimal, the probability of labels being true within the domain that the optimal halfspace $h(\boldsymbol{v})$ disagrees with it, i.e. $h(\boldsymbol{w})\backslash h(\boldsymbol{v})$, must be large. However, the same probability cannot be too large in the optimal halfspace $h(\boldsymbol{v})$ and, hence, $h(\boldsymbol{v}) \cap h(\boldsymbol{w})$. Then, if the underlying distribution has a well-behaved $\mathbf{x}$-marginal, the $l_2$ norm of the expectation of $\mathbf{x}$ within that domain should also be large.

In fact, the observation gives an insight of why $\mathcal{L}_\mathcal{D}(\boldsymbol{w})$ suits for the one-sided classification problems. As we are concerned about the one-sided loss, $\Pr\{\mathbf{x} \in h(\boldsymbol{w}) \cap \mathrm{y} = 1\}$, there are error no assumptions

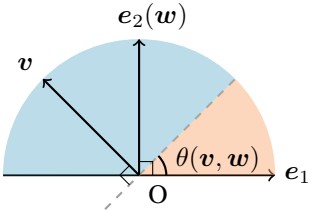

Figure 1: Blue area represents $h(\boldsymbol{v}) \cap h(\boldsymbol{w})$, orange area represents $h(\boldsymbol{w}) \backslash h(\boldsymbol{v})$.

on $h^c(\boldsymbol{w})$, which is also the key difference between regular classification and conditional classification. Therefore, we want to ensure the projection of $\mathbb{E}[g_{\boldsymbol{w}}(\mathbf{x}, \mathbf{y})]$ on to $\bar{\boldsymbol{v}}_{\boldsymbol{w}\top}$ is zero on $h^c(\boldsymbol{w})$, which, in turn, requires the objective loss to be zero on $h^c(\boldsymbol{w})$. Since $\mathcal{L}_{\mathcal{D}}(\boldsymbol{w}) = 0$ on $h^c(\boldsymbol{w})$, we will only need to discuss the properties of $\mathbb{E}[g_{\boldsymbol{w}}(\mathbf{x}, \mathbf{y})]$ on the domain where we have control.

Besides, an important implication of Proposition 3.2 is that, once $\theta(\boldsymbol{v}, \boldsymbol{w}) \in [0, \pi/2)$ and $h(\boldsymbol{w})$ is sub-optimal, $\mathbb{E}[-g_{\boldsymbol{w}}(\mathbf{x}, \mathbf{y})]$ always "points" to $\boldsymbol{v}$. Then, the update step (line 5) in Algorithm 2 will make $\theta(\boldsymbol{v}, \boldsymbol{w})$ contractive, which will, in turn, guarantee that the assumption $\theta(\boldsymbol{v}, \boldsymbol{w}) \in [0, \pi/2)$ is satisfied in the next iteration. This property plays a key role in proving Lemma 3.4.

To effectively utilize Proposition 3.2, we also have to show that its assumption is satisfied. That is, at least one of the weight vectors, $\boldsymbol{w}^{(1)}, \ldots, \boldsymbol{w}^{(T)}$, produced by Algorithm 2 has small $\|\mathbb{E}[g_{\boldsymbol{w}}(\mathbf{x}, \mathbf{y})]\|_2$. We show this can be achieved within a bounded number of iterations as the proposition below.

**Proposition 3.3.** *Let $\mathcal{D}$ be a distribution on $\mathbb{R}^d \times \{0, 1\}$ with standard normal $\mathbf{x}$-marginal, $g_{\boldsymbol{w}}(\mathbf{x}, \mathbf{y}) = \mathbf{y} \cdot \mathbf{x}_{\boldsymbol{w}\perp} \mathbb{1}\{\mathbf{x} \in h(\boldsymbol{w})\}$, and $\mathcal{L}_{\mathcal{D}}(\boldsymbol{w}) = \mathbb{E}_{(\mathbf{x}, \mathbf{y}) \sim \mathcal{D}}[\mathbf{y} \cdot \max(0, \langle \mathbf{x}, \boldsymbol{w} \rangle)]$. With $\beta = \sqrt{1/Td}$, after $T \geq 12d^2 \ln(1/\delta)/\epsilon^4$ iterations, the output $\boldsymbol{w}^{(1)}, \ldots, \boldsymbol{w}^{(T)}$ in Algorithm 2 will satisfy $\min_{i=1,\ldots,T} \|\mathbb{E}_{(\mathbf{x}, \mathbf{y}) \sim \mathcal{D}}[g_{\boldsymbol{w}^{(i)}}(\mathbf{x}, \mathbf{y})]\|_2 \leq \epsilon$ with probability at least $1 - \delta$.*

We defer the proof to Appendix B. Our technique resembles the work of Diakonikolas et al. (2020b), which showed that, if the objective function is **bounded** and has **Lipschitz continuous** gradient, then the norm of its gradient converges in boundedly many iterations of (Projected) SGD.

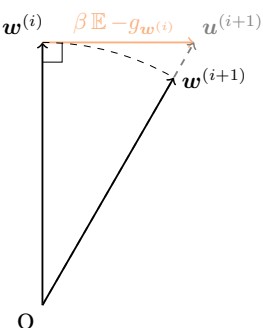
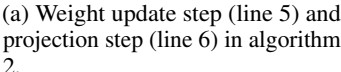
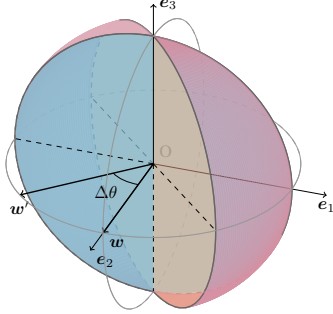

(a) Weight update step (line 5) and projection step (line 6) in algorithm 2.

(b) Orange plane is the decision boundary of $h(\boldsymbol{w}')$, while blue plane is that of $h(\boldsymbol{w})$. $\nabla_{\boldsymbol{w}} \mathcal{L}_{\mathcal{D}}(\boldsymbol{w})$ and $\nabla_{\boldsymbol{w}} \mathcal{L}_{\mathcal{D}}(\boldsymbol{w}')$ only differs in the two pink spherical sectors, which is dominated by $\Delta \theta$.

Figure 2: Boundedness of $\mathcal{L}_{\mathcal{D}}(\boldsymbol{w}^{(i)})$ and almost Lipschitz continuity of $\nabla_{\boldsymbol{w}} \mathcal{L}_{\mathcal{D}}(\boldsymbol{w})$.

However, the magnitude of $\mathcal{L}_{\mathcal{D}}(\boldsymbol{w})$ is dominated by $\|\boldsymbol{w}\|_2$, which could grow unbounded after many iterations, and its gradient $\nabla_{\boldsymbol{w}} \mathcal{L}_{\mathcal{D}}(\boldsymbol{w})$ has a "jumping" point at zero, which is not Lipschitz continuous in general. So, the key to proving Proposition 3.3 is to overcome these issues.

Observe that the gradient update (line 5) of Algorithm 2 will always produce $\|\boldsymbol{w}^{(i)}\|_2 \geq \|\boldsymbol{w}^{(i-1)}\|_2$, while the projection step (line 6) of Algorithm 2 will always make $\mathcal{L}_{\mathcal{D}}(\boldsymbol{w})$ bounded, cf. Figure 2a.

On the other hand, it turns out that $\nabla_{\boldsymbol{w}} \mathcal{L}_{\mathcal{D}}(\boldsymbol{w})$ is almost Lipschitz continuous under nice distributions such as a standard normal. Intuitively, if we perturb $\boldsymbol{w}$ a little bit to change it to $\boldsymbol{w}'$, it will only rotate the halfspace $h(\boldsymbol{w})$ by a very small angle, i.e. $\Delta\theta = \theta(\boldsymbol{w}, \boldsymbol{w}')$ is small. And, it suffices to consider the difference between $\nabla_{\boldsymbol{w}} \mathcal{L}_{\mathcal{D}}(\boldsymbol{w})$ and $\nabla_{\boldsymbol{w}} \mathcal{L}_{\mathcal{D}}(\boldsymbol{w}')$ on a 3-dimensional subspace as shown in figure 2b. Now, if the density of distribution $\mathcal{D}$ is not concentrated too much in any small spherical sectors in the subspace, it implies that the change of $\nabla_{\boldsymbol{w}} \mathcal{L}_{\mathcal{D}}(\boldsymbol{w})$ is dominated by $\Delta\theta$ (see Figure 2b), which is insignificant. This observation indicates that $\nabla_{\boldsymbol{w}} \mathcal{L}_{\mathcal{D}}(\boldsymbol{w})$ is Lipschitz continuous under anti-concentrated distributions unless $\|\boldsymbol{w}\|_2$ is extremely small.

Given Proposition 3.2 and Proposition 3.3, we show that in the list of parameters returned by Algorithm 2, at least one of them is approximately optimal:

**Lemma 3.4.** *Let $\mathcal{D}$ be a distribution on $\mathbb{R}^d \times \{0,1\}$ with standard normal* **x***-marginal, and $g_{\boldsymbol{w}}(\mathbf{x}, \mathrm{y}) = \mathrm{y} \cdot \mathbf{x}_{\boldsymbol{w}\perp} \mathbb{1}\{\mathbf{x} \in h(\boldsymbol{w})\}$. Suppose $\boldsymbol{v} \in \mathbb{R}^d$ is a unit vectors such that $\Pr_{(\mathbf{x}, \mathrm{y}) \sim \mathcal{D}}\{\mathbf{x} \in h(\boldsymbol{v}) \cap \mathrm{y} = 1\} \leq \epsilon$, if $T \geq 12d^2 \ln(2/\delta)/\epsilon^4$, $N \geq \ln(4T/\delta)/C\epsilon^2 \ln \epsilon^{-1}$ for some constant $C > 0$, and $\theta(\boldsymbol{v}, \boldsymbol{w}^{(0)}) \in [0, \pi/2)$, it holds that at least one $\boldsymbol{w} \in \mathcal{W}$ returned by Algorithm 2 satisfies $\Pr_{(\mathbf{x}, \mathrm{y}) \sim \mathcal{D}}\{\mathbf{x} \in h(\boldsymbol{w}) \cap \mathrm{y} = 1\} \leq \frac{5}{2}(\epsilon\sqrt{\ln \epsilon^{-1}})^{1/2}$ w.p. $1 - \delta$ for some sufficiently small $\epsilon \in [0, 1/e]$.*

We defer the formal proof to Appendix C, but sketch the idea here. Observe that combining the negation of Proposition 3.2 and Proposition 3.3 already yields Lemma 3.4. So, all we need to do is make sure that the assumption $\theta(\boldsymbol{v}, \boldsymbol{w}) \in [0, \pi/2)$ in Proposition 3.2 is satisfied.

Notice that, in the sequence of parameters $\boldsymbol{w}^{(1)}, \ldots, \boldsymbol{w}^{(T)}$ returned by Algorithm 2, every $\boldsymbol{w}^{(i)}$ must be significantly sub-optimal until we see a $\boldsymbol{w}$ such that $\Pr\{\mathbf{x} \in h(\boldsymbol{w}) \cap \mathrm{y} = 1\} \leq \frac{5}{2}(\epsilon\sqrt{\ln \epsilon^{-1}})^{1/2}$. If such a sub-optimal halfspace $h(\boldsymbol{w}^{(i)})$ also satisfies $\theta(\boldsymbol{v}, \boldsymbol{w}^{(i)}) \in [0, \pi/2)$, its negative projected gradient $\mathbb{E}[-g_{\boldsymbol{w}^{(i)}}]$ must has positive projection on $\bar{\boldsymbol{v}}_{\boldsymbol{w}\perp}$ by Proposition 3.2. Using such a $\mathbb{E}[-g_{\boldsymbol{w}^{(i)}}]$ to update $\boldsymbol{w}^{(i)}$ in Algorithm 2 will always produce $\theta(\boldsymbol{v}, \boldsymbol{w}^{(i+1)}) \leq \theta(\boldsymbol{v}, \boldsymbol{w}^{(i)})$. Thus, by an inductive argument, we can show that the first $\boldsymbol{w}^{(t)}$ such that $\|\mathbb{E}[g_{\boldsymbol{w}^{(t)}}]\|_2 < \frac{2}{5}\epsilon\sqrt{\ln \epsilon^{-1}}$ must satisfy $\theta(\boldsymbol{v}, \boldsymbol{w}^{(t)}) \in [0, \pi/2)$, which enables the application of Proposition 3.2.

### 3.3 GENERALIZATION TO SPARSE LINEAR CLASSEIFIERS

Although Algorithm 1 only applies to finite $\mathcal{C}$, we can generalize our approach to work with infinite classes of classifiers whenever they are list-learnable (Definition 1.3); for example, sparse linear classifiers are list-learnable in polynomial time. We present the corresponding performance guarantee for Algorithm 3 as follows while deferring the formal proof to Appendix D due to the page limit.

**Theorem 3.5.** *Let $\mathcal{D}$ be a distribution on $\mathbb{R}^d \times \{0,1\}$ with standard normal* **x***-marginal, and $\mathcal{C}$ be a class of sparse linear classifiers on $\mathbb{R}^d \times \{0,1\}$ with sparsity $s = O(1)$. If there exists a unit vector $\boldsymbol{v} \in \mathbb{R}^d$ and a $c \in \mathcal{C}$ such that, for some sufficiently small $\epsilon \in [0, 1/e]$, $\Pr_{(\mathbf{x}, \mathrm{y}) \sim \mathcal{D}}\{\mathbf{x} \in h(\boldsymbol{v}) \cap c(\mathbf{x}) \neq \mathrm{y}\} \leq \epsilon$, then, with at most $\mathrm{poly}(d, 1/\epsilon, 1/\delta)$ examples, Algorithm 3 will return a $\boldsymbol{w}^{(c')}$, with probability at least $1 - \delta$, such that $\Pr_{(\mathbf{x}, \mathrm{y}) \sim \mathcal{D}}\{\mathbf{x} \in h(\boldsymbol{w}^{(c')}) \cap c'(\mathbf{x}) \neq \mathrm{y}\} = \tilde{O}(\sqrt{\epsilon})$ and run in time $\mathrm{poly}(d, 1/\epsilon, 1/\delta)$.*

## 4 CONDITIONAL CLASSIFICATION WITH GENERAL HALFSPACES IS HARD

In this section, we show that it is computationally hard to obtain a small additive error for conditional classification with general halfspaces for any finite class of classifiers $\mathcal{C}$, even under distributions with standard normal **x**-marginals. Specifically, we show that, for each classifier $c \in \mathcal{C}$, approximating the optimal conditional classification loss over the class of general halfspaces on $\mathbb{R}^d$ with an **additive error** is at least as hard as achieving the same additive error for agnostic linear classification, which is known to be computationally hard (Diakonikolas et al., 2023). Further, we show that any $(1 + \alpha)$-approximation algorithm for conditional classification implies an $(1 + \alpha)$-approximation algorithm for standard classification, down to polynomially small losses. (The converse is not known to hold.)

The hardness of distribution-specific conditional classification is based on the sub-exponential hardness of "continuous Learning With Errors" (cLWE), which is a variant of the "Learning With Errors" (LWE) assumption. Informally speaking, in the problem of LWE, we are given labelled examples from two hypothesis cases. In one case, the labels are biased by some secret vector, while,

in another case, the labels are generated uniformly at random. We wish to distinguish between these cases. We formally define the problem of LWE (Regev, 2009), following Diakonikolas et al. (2023):

**Definition 4.1** (Learning With Errors). *For $m, d \in \mathbb{N}$, $q \in \mathbb{R}_+$, let $\mathcal{D}_{sample}, \mathcal{D}_{secret}, \mathcal{D}_{noise}$ be distributions on $\mathbb{R}^d, \mathbb{R}^d, \mathbb{R}$ respectively. In the $LWE(m, \mathcal{D}_{sample}, \mathcal{D}_{secret}, \mathcal{D}_{noise}, \mathrm{mod}_q)$ problem, with $m$ independent samples $\{(\mathbf{x}^{(1)}, \mathrm{y}^{(1)}), \ldots, (\mathbf{x}^{(m)}, \mathrm{y}^{(m)})\}$, we want to distinguish between the following two cases:*

- *Alternative hypothesis: each $(\mathbf{x}^{(i)}, \mathrm{y}^{(i)})$ is generated as $\mathrm{y}^{(i)} = \mathrm{mod}_q(\langle \mathbf{x}^{(i)}, \mathbf{s} \rangle + z)$, where $\mathbf{x}^{(i)} \sim \mathcal{D}_{sample}, \mathbf{s} \sim \mathcal{D}_{secret}, z \sim \mathcal{D}_{noise}$.*
- *Null hypothesis: each $\mathrm{y}^{(i)}$ is sampled uniformly at random on the support of its marginal distribution in the alternative hypothesis, independent of $\mathbf{x}^{(i)} \sim \mathcal{D}_{sample}$.*

An algorithm is said to be able to *solve the LWE problem with $\Delta$ advantage* if the probability that the algorithm outputs "alternative hypothesis" is $\Delta$ larger than the probability that it outputs "null hypothesis" when the given data is sampled from the alternative hypothesis distribution.

Let $\mathbb{S}^{d-1} := \{\mathbf{x} \in \mathbb{R}^d \mid \|\mathbf{x}\|_2 = 1\}$, $\mathbb{R}_q := [0, q)$, and $\mathrm{mod}_q : \mathbb{R}^d \to \mathbb{R}_q^d$ to be the function that applies $\mathrm{mod}_q$ operation on each coordinate of $\mathbf{x}$. Essentially, the hardness of cLWE is based on the sub-exponential hardness of LWE (see Appendix E). We formally state the assumption of sub-exponential hardness of cLWE as follows.

**Assumption 4.2** ((Gupte et al., 2022; Diakonikolas et al., 2023) Sub-exponential cLWE Assumption). *For any $d \in \mathbb{N}$, any constants $\kappa \in \mathbb{N}, \alpha \in (0, 1), \beta \in \mathbb{R}_+$ and any $\log^\beta d \leq k \leq Cd$ where $C > 0$ is a sufficiently small universal constant, the problem $LWE(d^{O(k^\alpha)}, \mathcal{N}^d(0, 1), \mathbb{S}^{d-1}, \mathcal{N}(0, \sigma^2), \mathrm{mod}_T)$ over $\mathbb{R}^d$ with $\sigma \geq k^{-\kappa}$ and $T = 1/C'\sqrt{k \log d}$, where $C' > 0$ is a sufficiently large universal constant, cannot be solved in time $d^{O(k^\alpha)}$ with $d^{-O(k^\alpha)}$ advantage.*

For simplicity, we define $\mathrm{y} \equiv \mathbb{1}\{c(\mathbf{x}) \neq \mathrm{y}'\}$ for $(\mathbf{x}, \mathrm{y}') \sim \mathcal{D}'$ and construct the distribution $(\mathbf{x}, \mathrm{y}) \sim \mathcal{D}$. Notice that, in agnostic settings, since $\mathcal{D}'$ is worst case, $\mathcal{D}$ is also worst case. Therefore, this replacement does not affect the difficulty of the problems we consider.

Normally, one would consider the classification loss to be the expected disagreement between the classifier and the labelling. However, it is more convenient for us to view a labelling $\mathrm{y} = 1$ as an "occurrence of error" and define the loss in terms of such occurrences. Specifically, for any subset $S \subseteq \mathbb{R}^d$ and any distribution $\mathcal{D}$ on $\mathbb{R}^d \times \{0, 1\}$, we define the classification loss as $\mathrm{err}_{\mathcal{D}}(S) = \Pr_{(\mathbf{x}, \mathrm{y}) \sim \mathcal{D}}\{\mathrm{y} = \mathbb{1}\{\mathbf{x} \in S\}\}$. Note that this definition of classification loss is essentially the same as the "traditional" classification loss that defined in terms of disagreement since we can convert from one to another by simply negating the labelling. Analogously, for any subsets $S, T \subseteq \mathbb{R}^d$ and any distribution $\mathcal{D}$ on $\mathbb{R}^d \times \{0, 1\}$, we denote the conditional loss by $\mathrm{err}_{\mathcal{D}|T}(S) = \Pr_{(\mathbf{x}, \mathrm{y}) \sim \mathcal{D}}\{\mathrm{y} = \mathbb{1}\{\mathbf{x} \in S\} \mid \mathbf{x} \in T\}$. When $S \equiv T$, we abbreviate $\mathrm{err}_{\mathcal{D}|T}(S)$ to $\mathrm{err}_{\mathcal{D}|T}$.

**Theorem 4.3** (Hardness Of Conditional Classification). *Let $\mathcal{D}$ be any distribution on $\mathbb{R}^d \times \{0, 1\}$ with standard normal $\mathbf{x}$-marginals, $\mathcal{H}$ be the class of halfspaces on $\mathbb{R}^d$, and define $\mathcal{H}_{\mathcal{D}}^{a,b} = \{h_t(\boldsymbol{w}) \in \mathcal{H} \mid \Pr_{\mathbf{x} \sim \mathcal{D}_\mathbf{x}}\{\mathbf{x} \in h_t(\boldsymbol{w})\} \in [a, b]\}$ for any $0 \leq a \leq b \leq 1$. Under Assumption 4.2, for any constant $\alpha \in (0, 2), \gamma > 1/2$ and any $c/\sqrt{d \log d} \leq \epsilon \leq 1/\log^\gamma d$ where $c$ is a sufficiently large constant, there is no algorithm that can find a halfspace $h_{t'}(\boldsymbol{w}) \in \mathcal{H}_{\mathcal{D}}^{a,b}$ such that $\mathrm{err}_{\mathcal{D}|h_{t'}(\boldsymbol{w})} \leq \min_{h_t(\boldsymbol{u}) \in \mathcal{H}_{\mathcal{D}}^{a,b}} \mathrm{err}_{\mathcal{D}|h_t(\boldsymbol{u})} + \epsilon$ and runs in time $d^{O(1/(\epsilon \sqrt{\log d})^\alpha)}$.*

Theorem 4.3 states our hardness result for conditional classification, which is a simple consequence of Proposition 4.5 and Lemma 4.6. The former one shows that conditional classification is at least as hard as agnostic classification and the latter one shows that agnostically learning halfspaces is hard.

Our main contribution is Proposition 4.5, but before getting into it, we first show a simple but critical observation that reveals the relationship between $\mathrm{err}_{\mathcal{D}}(S)$ and $\mathrm{err}_{\mathcal{D}|S}$. That is, the loss of agnostic classification can be explictly expressed by the loss of conditional classification.

**Lemma 4.4** (Classification Error Decomposition). *Let $\mathcal{D}$ be any distribution on $\mathbb{R}^d \times \{0, 1\}$ and $S$ be any subset of $\mathbb{R}^d$, we have $\mathrm{err}_{\mathcal{D}}(S) = 2\mathrm{err}_{\mathcal{D}|S} \Pr_{\mathcal{D}}\{\mathbf{x} \in S\} + \Pr_{\mathcal{D}}\{\mathrm{y} = 0\} - \Pr_{\mathcal{D}}\{\mathbf{x} \in S\}$ as well as $\mathrm{err}_{\mathcal{D}}(S) = 2\mathrm{err}_{\mathcal{D}|S^c}(S) \Pr_{\mathcal{D}}\{\mathbf{x} \in S^c\} + \Pr_{\mathcal{D}}\{\mathrm{y} = 1\} - \Pr_{\mathcal{D}}\{\mathbf{x} \in S^c\}$.*

Due to page limits, we defer its proof to Appendix E. Lemma 4.4 is a powerful result since it allows us to establish a reduction from classification to conditional classification.

Briefly speaking, if we know $\Pr\{\mathbf{x} \in S^*\}$ for some optimal solution $S^*$ to the agnostic classification problem, we can approximate $\mathrm{err}_{\mathcal{D}}(S^*)$ by approximating its conditional classification loss, i.e. $\mathrm{err}_{\mathcal{D}|S^*}$. Even though we do not know $\Pr\{\mathbf{x} \in S^*\}$, we can guess a small range that contains $\Pr\{\mathbf{x} \in S^*\}$, and enforce such a constraint just as in Definition 1.1. Then, we sweep over all such small intervals and one of the instances being solved must include $\Pr\{\mathbf{x} \in S\}$. Once we take these intervals small enough, it won't incur a significant error. We use this strategy to prove Proposition 4.5, but the formal proof is deferred to Appendix E due to the page limit.

**Proposition 4.5** (Reduction In Additive Form). *Let $\mathcal{D}$ be any distribution on $\mathbb{R}^d \times \{0,1\}$, $\mathcal{H}$ be any subset of the power set of $\mathbb{R}^d$ closed under complement, and define $\mathcal{H}_{\mathcal{D}}^{a,b} = \{S \in \mathcal{H} \mid \Pr_{\mathcal{D}}\{\mathbf{x} \in S\} \in [a,b]\}$ for any $0 \leq a \leq b \leq 1$. For any such $a, b$ and $\epsilon, \delta > 0$, given sample access to $\mathcal{D}$, if there exists an algorithm $\mathcal{A}_1(\epsilon, \delta, a, b)$ running in time $\mathrm{poly}(d, 1/\epsilon, 1/\delta)$, that outputs $S_1 \in \mathcal{H}_{\mathcal{D}}^{a,b}$ such that $\mathrm{err}_{\mathcal{D}|S_1} \leq \min_{S \in \mathcal{H}_{\mathcal{D}}^{a,b}} \mathrm{err}_{\mathcal{D}|S} + \epsilon$ with probability as least $1 - \delta$, there exists another algorithm $\mathcal{A}_2(\epsilon, \delta)$, that runs in time $\mathrm{poly}(d, 1/\epsilon, 1/\delta)$, and outputs $S_2 \in \mathcal{H}$ such that $\mathrm{err}_{\mathcal{D}}(S_2) \leq \min_{S \in \mathcal{H}} \mathrm{err}_{\mathcal{D}}(S) + 6\epsilon$ with probability at least $1 - \delta$.*

Furthermore, Lemma 4.6 states that agnostically learning halfspaces up to small additive error is computationally hard. Since Proposition 4.5 holds for halfspaces on $\mathbb{R}^d$, conditional learning has at least the same hardness by combining Proposition 4.5 and Lemma 4.6.

**Lemma 4.6** (Corollary 3.2 of Diakonikolas et al. (2023)). *Let $\mathcal{D}$ be any distribution on $\mathbb{R}^d \times \{0,1\}$ with standard normal $\mathbf{x}$-marginals, and $\mathcal{H}$ be the class of halfspaces on $\mathbb{R}^d$. Under Assumption 4.2, for any constant $\alpha \in (0,2)$, $\gamma > 1/2$ and any $c/\sqrt{d \log d} \leq \epsilon \leq 1/\log^{\gamma} d$ where $c$ is a sufficiently large constant, there is no algorithm that can find a halfspace $h_{t'}(\mathbf{v}) \in \mathcal{H}$ such that $\mathrm{err}_{\mathcal{D}}(h_{t'}(\mathbf{v})) \leq \min_{h_t(\mathbf{u}) \in \mathcal{H}} \mathrm{err}_{\mathcal{D}}(h_t(\mathbf{u})) + \epsilon$ and runs in time $d^{O(1/(\epsilon \sqrt{\log d})^{\alpha})}$.*

Analogously, a reduction in multiplicative form can also be obtained using a similar analysis to that in the proof of Proposition 4.5. In particular, we show that if there exists a multiplicative approximation algorithm for conditional classification with factor $1 + \alpha$, there must exist another multiplicative approximation algorithm for classification in agnostic setting with the same factor $1 + \alpha$.

**Claim 4.7** (Reduction In Multiplicative Form). *Let $\mathcal{D}$ be any distribution on $\mathbb{R}^d \times \{0,1\}$, $\mathcal{H}$ be any subset of the power set of $\mathbb{R}^d$ closed under complement, and define $\mathcal{H}_{\mathcal{D}}^{a,b} = \{S \in \mathcal{H} \mid \Pr_{\mathcal{D}}\{\mathbf{x} \in S\} \in [a,b]\}$ for any $0 \leq a \leq b \leq 1$. If there exists an algorithm $\mathcal{A}_1(\alpha, \delta, a, b)$ that given sample access to $\mathcal{D}$, any such $a, b$, and $\alpha, \epsilon, \delta > 0$, runs in time $\mathrm{poly}(d, 1/\alpha, 1/\delta)$, and outputs $S_1 \in \mathcal{H}_{\mathcal{D}}^{a,b}$ such that $\mathrm{err}_{\mathcal{D}|S_1} \leq (1 + \alpha) \min_{S \in \mathcal{H}_{\mathcal{D}}^{a,b}} \mathrm{err}_{\mathcal{D}|S}$ with probability as least $1 - \delta$, there exists another algorithm $\mathcal{A}_2(\alpha, \epsilon, \delta)$ that runs in time $\mathrm{poly}(d, 1/\alpha, 1/\epsilon, 1/\delta)$, and outputs $S_2 \in \mathcal{H}$ such that $\mathrm{err}_{\mathcal{D}}(S_2) \leq (1 + \alpha)(\min_{S \in \mathcal{H}} \mathrm{err}_{\mathcal{D}}(S) + 4\epsilon)$ with probability at least $1 - \delta$.*

Again, we defer the proof to Appendix E because of page limits. Although there is an extra $4\epsilon$ additive error in the final guarantee of Claim 4.7, we can afford to take $\epsilon$ polynomially small w.r.t. $d, \alpha, \delta$, thus obtaining the multiplicative error guarantee down to polynomially small error. Informally we observe that Proposition 4.5 and Claim 4.7 indicate that any form of approximation algorithm for conditional classification yields an approximation algorithm of the same factor for agnostic classification. In the case of multiplicative approximation in particular, the reverse is not known and we observe that it might be strictly harder to approximate the conditional classification objective.

## 5    LIMITATIONS AND FUTURE WORK

Our algorithmic result is limited in three aspects. First and foremost, the restriction of selectors to homogeneous halfspaces is a major drawback especially for the task of conditional classification. Indeed, the advantage of conditional classification with halfspaces compared with regular linear classification really shines when we have the ability to select a minority of the population. Therefore, moving from homogeneous halfspaces to general halfspaces would constitute a significant advance even with worse error bound. Another limitation of our result is the strong assumption on the marginal distribution. Real-world data almost never has standard normal marginals, and testing for a standard normal distribution is costly. Hence, it's worth trying to extend our result to more general classes of distributions, such as log-concave distributions. Last but not the least, one can also try to improve our error guarantee under the current setting as the error guarantee $O(\sqrt{\mathrm{opt}})$ appears sub-optimal.

ACKNOWLEDGMENTS

We are grateful for many discussions with and feedback from Daniel Hsu. This work was partially supported by NSF awards IIS-1908287, and IIS-1942336, and NSF-Amazon award IIS-1939677.

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
