## A    Review of Robust List Learning of Sparse Linear Classifiers

---

**Algorithm 4:** Robust list learning of sparse linear classifiers

---

1 **procedure** SPARSELIST$(\mathcal{D}, m)$
2 $\quad L \leftarrow \varnothing$
3 $\quad \nu \leftarrow 2^{-(bs+s\log s)}$
4 $\quad$ Sample $(\mathbf{x}^{(1)}, \mathrm{y}^{(1)}), \ldots, (\mathbf{x}^{(m)}, \mathrm{y}^{(m)}) \sim \mathcal{D}$
5 $\quad$ **foreach** $(i_1, \ldots, i_s) \in [d]^s$ *and* $(j_s) \in [m]^s$ **do**
6 $\qquad \boldsymbol{w} \leftarrow \begin{bmatrix} \mathrm{y}^{(j_1)}\mathrm{x}_{i_1}^{(j_1)} & \cdots & \mathrm{y}^{(j_1)}\mathrm{x}_{i_s}^{(j_1)} \\ & \vdots & \\ \mathrm{y}^{(j_s)}\mathrm{x}_{i_1}^{(j_s)} & \cdots & \mathrm{y}^{(j_s)}\mathrm{x}_{i_s}^{(j_s)} \end{bmatrix}^{-1} \begin{bmatrix} \mathrm{y}^{(j_1)} - \nu \\ \vdots \\ \mathrm{y}^{(j_s)} - \nu \end{bmatrix}$
7 $\qquad L \leftarrow L \cup \{\boldsymbol{w}\}$
8 $\quad$ **end**
9 $\quad$ **return** $L$

---

For completeness, we now describe an algorithm to solve the robust list learning problem for sparse linear classifiers. It is based on the approach used in the algorithm for conditional sparse linear regression (Juba, 2017), using an observation by Mossel & Sudan (2016). We will prove the following:

**Theorem A.1.** *Suppose the numbers are b-bit rational values, Algorithm 4 solves robust list-learning of linear classifiers with $s = O(1)$ nonzero coefficients and margin $\nu \geq 2^{-(bs+s\log s)}$ from $m = O(\frac{1}{\alpha\epsilon}(s\log d + \log\frac{1}{\delta}))$ examples in polynomial time with list size $O((md)^s)$.*

*Proof.* We observe that the running time and list size of Algorithm 4 is clearly as promised. To see that it solves the problem, we first recall that results by Blumer et al. (1989) and Hanneke (2016) showed that given $O(\frac{1}{\epsilon}(D + \log\frac{1}{\delta}))$ examples labeled by a class of VC-dimension $D$, any consistent hypotheses achieves error $\epsilon$ with probability $1 - \delta$. In particular, halfspaces in $\mathbb{R}^d$ have VC-dimension $d$; Haussler (1988) observed that $s$-sparse linear classifiers in $\mathbb{R}^d$ have VC-dimension $s\log d$. Hence, if the data includes a set $S$ of at least $\Omega(\frac{1}{\epsilon}(s\log d + \log\frac{1}{\delta}))$ inliers and we find a $s$-sparse classifier that agrees with the labels on $S$, it achieves error $1 - \epsilon$ on $S$ with probability $1 - \delta/2$. Observe that in a sample of size $O(\frac{1}{\alpha\epsilon}(s\log d + \log\frac{1}{\delta}))$, with an $\alpha$ fraction of inliers, we indeed obtain $\Omega(\frac{1}{\epsilon}(s\log d + \log\frac{1}{\delta}))$ inliers with probability $1 - \delta/2$.

Now, suppose we write our linear threshold function with a standard threshold of 1, and suppose are examples are drawn from $\mathbb{R}^d \times \{-1, 1\}$. Then a classifier with weight vector $\boldsymbol{w}$ labels $\mathbf{x}$ with 1 if $\langle \boldsymbol{w}, \mathbf{x} \rangle \geq 1$, and labels $\mathbf{x}$ with $-1$ if $\langle \boldsymbol{w}, \mathbf{x} \rangle < 1$. We observe that by Cramer's rule, we can find a value $\nu^* > 0$ (of size at least $2^{-(bs+s\log s)}$ if the numbers are $b$-bit rational values) such that if $\langle \boldsymbol{w}, \mathbf{x} \rangle < 1$, $\langle \boldsymbol{w}, \mathbf{x} \rangle \leq 1 - \nu^*$. So, it is sufficient for $\langle \boldsymbol{w}, \mathrm{y}\mathbf{x} \rangle \geq \mathrm{y} - \nu$ for a given $(\mathbf{x}, \mathrm{y})$, for some margin $\nu \geq 2^{-(bs+s\log s)}$. Thus, to find a consistent $\boldsymbol{w}$, it suffices to solve the linear program $\langle \boldsymbol{w}, \mathrm{y}^{(j)}\mathbf{x}^{(j)} \rangle \geq \mathrm{y}^{(j)} - \nu$ for each $j$th example in $S$. Observe that if we parameterize $\boldsymbol{w}$ by only the nonzero coefficients, we obtain a linear program in $s$ dimensions, for which we can obtain a feasible solution at a vertex, given by $s$ tight constraints. Now, Algorithm 4 enumerates *all* $s$-tuples of indices and examples, which in particular therefore must include any $s$-tuple of examples in the inlier set $S$ and the $s$ nonzero coordinates of $\boldsymbol{w}$. Hence, with probability at least $1 - \delta$, $L$ indeed contains some $\boldsymbol{w}$ that attains error $\epsilon$ on $S$, as needed. $\qquad\square$

## B    Convergence Analysis of Projected SGD

We show our formal analysis of the Projected SGD (Algorithm 2) in this section.

We first show that the gradient of statistic ReLU, $\nabla_{\boldsymbol{w}}\mathcal{L}_{\mathcal{D}}(\boldsymbol{w})$, is almost Lipschitz continuous, which will be a critical piece in our convergence analysis of Projected SGD.

**Lemma B.1** (Relative Smoothness Of Statistic ReLU). *Let $\mathcal{D}$ be a distribution on $\mathbb{R}^d \times [-1, +1]$ with standard normal $\mathbf{x}$-marginal, $\mathcal{L}_{\mathcal{D}}(\boldsymbol{w}) = \mathbb{E}_{(\mathbf{x},\mathrm{y})\sim\mathcal{D}}[\mathrm{y} \cdot \max(0, \langle \mathbf{x}, \boldsymbol{w} \rangle)]$. Then, for any $\boldsymbol{v}, \boldsymbol{w} \in \mathbb{R}^d$*

*such that at least one of $\|v\|_2, \|w\|_2$ is non-zero, we have*

$$\|\nabla_w \mathcal{L}_\mathcal{D}(w) - \nabla_v \mathcal{L}_\mathcal{D}(v)\|_2 \leq \frac{2}{\max(\|v\|_2, \|w\|_2)}\|w - v\|_2 \tag{1}$$

*Proof.* Without loss of generosity, assume $\|v\|_2 \geq \|w\|_2$, we prove the approximate Lipschitz continuity of $\nabla_w \mathcal{L}_\mathcal{D}(w)$ by showing that, for every $w, v \in \mathbb{R}^d$, $\|\nabla_w \mathcal{L}_\mathcal{D}(w) - \nabla_v \mathcal{L}_\mathcal{D}(v)\|_2$ is upper bound by $L \cdot \|w - v\|_2 / \|v\|_2$ for some constant $L \geq 1$.

We firstly show that $\|\nabla_w \mathcal{L}_\mathcal{D}(w) - \nabla_v \mathcal{L}_\mathcal{D}(v)\|_2 = O(\theta(v, w))$, then we will prove $\theta(v, w)$ can be upper bounded by $\|w - v\|_2 / \|v\|_2$ asymptotically for $\theta(v, w) \in [0, \pi/2]$ and $\theta(v, w) \in [\pi/2, \pi]$ separately.

Recall that $\nabla_w \mathcal{L}_\mathcal{D}(w) = \mathbb{E}_{(\mathbf{x}, \mathrm{y}) \sim \mathcal{D}}[\mathrm{y} \cdot \mathbf{x} \cdot \mathbb{1}\{\mathbf{x} \in h(w)\}]$. For conciseness of the proof, we define

$$u = \underset{\|z\|_2 = 1}{\operatorname{argmax}} \langle \nabla_w \mathcal{L}_\mathcal{D}(w) - \nabla_v \mathcal{L}_\mathcal{D}(v), z \rangle.$$

Then, we have

$$\begin{aligned}
\|\nabla_w \mathcal{L}_\mathcal{D}(w) - \nabla_v \mathcal{L}_\mathcal{D}(v)\|_2 &= \langle \nabla_w \mathcal{L}_\mathcal{D}(w) - \nabla_v \mathcal{L}_\mathcal{D}(v), u \rangle \\
&= \mathbb{E}[\mathrm{y} \cdot \langle \mathbf{x}, u \rangle (\mathbb{1}\{\mathbf{x} \in h(w) \cap h^c(v)\} - \mathbb{1}\{\mathbf{x} \in h^c(w) \cap h(v)\})] \\
&\leq \mathbb{E}[|\langle \mathbf{x}, u \rangle| (\mathbb{1}\{\mathbf{x} \in h(w) \cap h^c(v)\} + \mathbb{1}\{\mathbf{x} \in h^c(w) \cap h(v)\})]. \tag{2}
\end{aligned}$$

Now, let's notice that the expectation above only has constraints on a 2-dimensional subspace spanned by $\{v, w\}$. Thus, will show that $|\langle \mathbf{x}, u \rangle|$ is essentially upper bounded by the $l_2$ norm of the projection of $\mathbf{x}$ onto a 3-dimensional subspace, which will allow us to use polar coordinates to calculate the above expectation.

We construct a set of orthonormal basis $v = \{e_1, e_2, e_3\}$ as follow. At first, let $\theta_1 = \theta(v, w)$ so that $\theta_1 \in [0, \pi]$, and we define $\bar{w} = e_2$ as well as $\bar{v} = -e_1 \sin\theta_1 + e_2 \cos\theta_1$. Then, denote $u_W$ to be the projection of $u$ on to the subspace spanned by $W = \{e_1, e_2\}$ and $\theta_2 = \theta(u_W, e_1)$ so that $\bar{u}_W = e_1 \cos\theta_2 + e_2 \sin\theta_2$. At last, we define $\theta_3 = \theta(u, u_W)$ so that $\theta_3 \in [0, \pi/2]$ and $e_3$ to be such that

$$\begin{aligned}
u &= \bar{u}_W \cos\theta_3 + e_3 \sin\theta_3 \\
&= e_1 \cos\theta_2 \cos\theta_3 + e_2 \sin\theta_2 \cos\theta_3 + e_3 \sin\theta_3.
\end{aligned}$$

Denote $\mathrm{x}_i = \langle \mathbf{x}, e_i \rangle$ and $\mathbf{x}_V$ to be the projection of $\mathbf{x}$ onto the subspace spanned by $V$, by Cauchy inequality, there is

$$\begin{aligned}
\langle \mathbf{x}, u \rangle &= \mathrm{x}_1 \cos\theta_2 \cos\theta_3 + \mathrm{x}_2 \sin\theta_2 \cos\theta_3 + \mathrm{x}_3 \sin\theta_3 \\
&= \langle \mathbf{x}_V, u \rangle \\
&\leq \|\mathbf{x}_V\|_2
\end{aligned}$$

Then, we transform the standard 3-dimensional coordinate system into a spherical coordinate system, also see figure 3. For any $\mathbf{x}_V = (\mathrm{x}_1, \mathrm{x}_2, \mathrm{x}_3)$, let $\phi = \theta(\mathbf{x}_V, e_3)$, $\theta = \theta(\mathbf{x}_{Ve_3^\perp}, e_1)$, and $r = \|\mathbf{x}_V\|_2$, then we have $\mathrm{x}_3 = r \cos\phi$, $\mathrm{x}_1 = r \sin\phi \cos\theta$, and $\mathrm{x}_2 = r \sin\phi \sin\theta$. Now, applying the standard Jacobian matrix that maps the spherical coordinates to 3-dimensional Cartesian coordinates yields $d\mathrm{x}_1 d\mathrm{x}_2 d\mathrm{x}_3 = r^2 \sin\phi \, dr \, d\phi \, d\theta$. Therefore, following with inequality (2), we have

$$\begin{aligned}
\|\nabla_w \mathcal{L}_\mathcal{D}(w) - \nabla_v \mathcal{L}_\mathcal{D}(v)\|_2 &\leq 2 \, \mathbb{E}[\|\mathbf{x}_V\|_2 \mathbb{1}\{\mathbf{x} \in h(w) \cap h^c(v)\}] \\
&= 2 \, \mathbb{E}[\|\mathbf{x}_V\|_2 \mathbb{1}\{\mathrm{x}_2 \cos\theta_1 \leq \mathrm{x}_1 \sin\theta_1, \mathrm{x}_2 \geq 0, \mathrm{x}_3 \in \mathbb{R}\}] \\
&\overset{\text{(i)}}{=} \frac{1}{\sqrt{2\pi^3}} \int_0^{\theta_1} \int_0^\pi \int_0^{+\infty} r^3 \sin\phi e^{-r^2/2} dr \, d\phi \, d\theta \\
&= \theta_1 \sqrt{\frac{2}{\pi^3}} \int_0^{+\infty} r^3 e^{-r^2/2} dr \\
&\overset{\text{(ii)}}{=} \theta_1^{(2/\pi)^{3/2}} \int_0^{+\infty} r e^{-r^2/2} dr \\
&= \theta_1^{(2/\pi)^{3/2}} \tag{3}
\end{aligned}$$

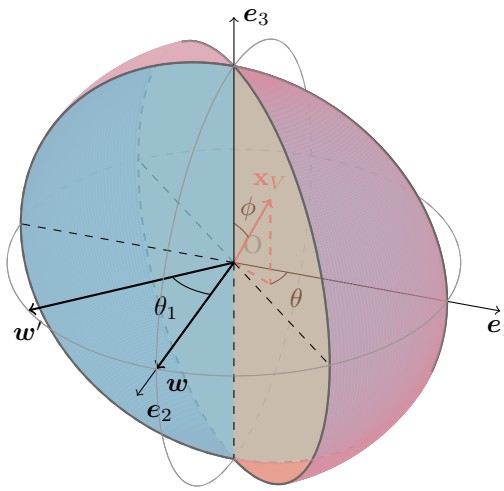

Figure 3: Spherical coordinate interpretation.

where the first inequality holds because $\{\mathbf{x}_V \mid \mathbf{x}_V \in h(\boldsymbol{w}) \cap h^c(\boldsymbol{v})\}$ and $\{\mathbf{x}_V \mid \mathbf{x}_V \in h^c(\boldsymbol{w}) \cap h(\boldsymbol{v})\}$ are symmetric under Gaussian measure, the integral domain in equation (i) is valid for $\theta, \theta_1 \in [0, \pi]$ because $\mathrm{x}_2 \cos\theta_1 \leq \mathrm{x}_1 \sin\theta_1$ implies $\cot\theta_1 \leq \cot\theta$, which, in turn, indicates $0 \leq \theta \leq \theta_1$ as $\cot\theta$ is a monotone decreasing function on $\theta \in (0, \pi)$, and $\mathrm{x}_2 \geq 0$ implies $\phi \in [0, \pi]$ as we know $r, \sin\theta \geq 0$ by construction, inequality (ii) is obtained by using the law of *integration by parts*.

**For the case of** $\theta_1 \in [0, \pi/2]$, it is easy to see that

$$
\begin{aligned}
\|\boldsymbol{w} - \boldsymbol{v}\|_2 &\geq \|\bar{\boldsymbol{w}} \cdot \|\boldsymbol{v}\|_2 \cos\theta_1 - \boldsymbol{v}\|_2 \\
&= \|\boldsymbol{v}\|_2 \sin\theta_1 \\
&\overset{\text{(i)}}{\geq} \|\boldsymbol{v}\|_2 \theta_1 \cos\frac{\pi}{2\sqrt{3}} \\
&\overset{\text{(ii)}}{\geq} \left(\frac{\pi}{2}\right)^{3/2} \cos\frac{\pi}{2\sqrt{3}} \|\boldsymbol{v}\|_2 \cdot \|\nabla_{\boldsymbol{w}}\mathcal{L}_\mathcal{D}(\boldsymbol{w}) - \nabla_{\boldsymbol{v}}\mathcal{L}_\mathcal{D}(\boldsymbol{v})\|_2 \\
&\geq \|\boldsymbol{v}\|_2 \cdot \|\nabla_{\boldsymbol{w}}\mathcal{L}_\mathcal{D}(\boldsymbol{w}) - \nabla_{\boldsymbol{v}}\mathcal{L}_\mathcal{D}(\boldsymbol{v})\|_2
\end{aligned}
$$

where the first inequality holds because the RHS represent the shortest distance from vector $\boldsymbol{v}$ to vector $\boldsymbol{w}$, (i) is by the elementary inequality $x \cos(x/\sqrt{3}) \leq \sin x$ as well as the assumption $\theta_1 \in [0, \pi/2]$, (ii) is by inequality (3), the last inequality holds due to $3/5 < \cos(\pi/2\sqrt{3})$ and $5/3 < (\pi/2)^{3/2}$.

**For the case of** $\theta_1 \in [\pi/2, \pi]$, by inequality (3) and $\|\boldsymbol{w} - \boldsymbol{v}\|_2 \geq \|\boldsymbol{v}\|_2$, we simply have

$$
\|\nabla_{\boldsymbol{w}}\mathcal{L}_\mathcal{D}(\boldsymbol{w}) - \nabla_{\boldsymbol{v}}\mathcal{L}_\mathcal{D}(\boldsymbol{v})\|_2 \leq \sqrt{\pi/2} \leq \frac{2}{\|\boldsymbol{v}\|_2}\|\boldsymbol{w} - \boldsymbol{v}\|_2
$$

which completes the proof by taking $L = 2$. $\qquad\square$

Now we are ready to show the convergence of the gradient norm in Algorithm 2. We first prove a more general version of Proposition 3.3 as follow, and then give Proposition 3.3 as its corollary.

**Proposition B.2.** *Let $\mathcal{D}$ be any distribution on $\mathbb{R}^d \times \{0, 1\}$ with $\mathbf{x}$-marginal such that $\max_{\boldsymbol{u} \in \mathbb{R}^d} \left\|\langle \bar{\boldsymbol{u}}, \mathbf{x}\rangle\right\|_p \leq K$ for all $p \in \{1, 2\}$ and some absolute constant $K > 0$. Define $\mathcal{L}_\mathcal{D}(\boldsymbol{w}) = \mathbb{E}_{(\mathbf{x},\mathrm{y})\sim\mathcal{D}}[\mathrm{y} \cdot \max(0, \langle \mathbf{x}, \boldsymbol{w}\rangle)]$ as well as $g_{\boldsymbol{w}}(\mathbf{x}, \mathrm{y}) = \mathrm{y} \cdot \mathbf{x}_{\boldsymbol{w}^\perp} \mathbb{1}\{\mathbf{x} \in h(\boldsymbol{w})\}$. Suppose $\|\nabla_{\boldsymbol{u}}\mathcal{L}_\mathcal{D}(\boldsymbol{u}) - \nabla_{\boldsymbol{w}}\mathcal{L}_\mathcal{D}(\boldsymbol{w})\|_2 \leq L \cdot \|\boldsymbol{u} - \boldsymbol{w}\|_2 / \max(\|\boldsymbol{u}\|_2, \|\boldsymbol{w}\|_2)$ for some constant $L > 0$, then, with $\beta = \sqrt{2/TKLd}$, after $T$ iterations, the output $(\boldsymbol{w}^{(1)}, \dots, \boldsymbol{w}^{(T)})$ in algorithm 2 will satisfies*

$$
\mathbb{E}_{\hat{\mathcal{D}}^{(1)},\dots,\hat{\mathcal{D}}^{(T)}\sim\mathcal{D}} \left[\frac{1}{T}\sum_{i=1}^{T}\left\|\mathbb{E}_{(\mathbf{x},\mathrm{y})\sim\mathcal{D}}[g_{\boldsymbol{w}^{(i)}}(\mathbf{x}, \mathrm{y})]\right\|_2^2\right] \leq \sqrt{\frac{K^3 L d}{2T}}.
$$

*In addition, if $T \geq (2K^3 Ld + 8K^4 d^2 \ln(1/\delta))/\epsilon^4$, it holds $\min_{i=1,\dots,T} \|\mathbb{E}_{(\mathbf{x},\mathrm{y})\sim\mathcal{D}}[g_{\boldsymbol{w}^{(i)}}(\mathbf{x},\mathrm{y})]\|_2 \leq \epsilon$, with probability at least $1 - \delta$.*

*Proof.* Consider the $i$th iteration of algorithm 2. Based on the gradient step $\boldsymbol{u}^{(i)} = \boldsymbol{w}^{(i-1)} - \beta\,\mathbb{E}_{\hat{\mathcal{D}}^{(i)}}[g_{\boldsymbol{w}^{(i-1)}}(\mathbf{x},\mathrm{y})]$, we have

$$
\begin{aligned}
&\mathcal{L}_{\mathcal{D}}(\boldsymbol{u}^{(i)}) - \mathcal{L}_{\mathcal{D}}(\boldsymbol{w}^{(i-1)}) \\
&= \left\langle \nabla_{\boldsymbol{w}}\mathcal{L}_{\mathcal{D}}(\boldsymbol{w}^{(i-1)}), \boldsymbol{u}^{(i)} - \boldsymbol{w}^{(i-1)} \right\rangle \\
&\quad + \int_0^1 \left\langle \nabla_{\boldsymbol{w}}\mathcal{L}_{\mathcal{D}}(\boldsymbol{w}^{(i-1)} + t(\boldsymbol{u}^{(i)} - \boldsymbol{w}^{(i-1)})) - \nabla_{\boldsymbol{w}}\mathcal{L}_{\mathcal{D}}(\boldsymbol{w}^{(i-1)}), \boldsymbol{u}^{(i)} - \boldsymbol{w}^{(i-1)} \right\rangle dt \\
&\leq -\beta \left\langle \nabla_{\boldsymbol{w}}\mathcal{L}_{\mathcal{D}}(\boldsymbol{w}^{(i-1)}), \mathbb{E}_{\hat{\mathcal{D}}^{(i)}}[g_{\boldsymbol{w}^{(i-1)}}(\mathbf{x},\mathrm{y})] \right\rangle \\
&\quad + \int_0^1 \|\nabla_{\boldsymbol{w}}\mathcal{L}_{\mathcal{D}}(\boldsymbol{w}^{(i-1)} + t(\boldsymbol{u}^{(i)} - \boldsymbol{w}^{(i-1)})) - \nabla_{\boldsymbol{w}}\mathcal{L}_{\mathcal{D}}(\boldsymbol{w}^{(i-1)})\|_2 \|\boldsymbol{u}^{(i)} - \boldsymbol{w}^{(i-1)}\|_2 dt \\
&\overset{(i)}{\leq} -\beta \left\langle \mathbb{E}_{\mathcal{D}}[g_{\boldsymbol{w}^{(i-1)}}(\mathbf{x},\mathrm{y})], \mathbb{E}_{\hat{\mathcal{D}}^{(i)}}[g_{\boldsymbol{w}^{(i-1)}}(\mathbf{x},\mathrm{y})] \right\rangle \\
&\quad + \int_0^1 \frac{t}{\max(\|(1-t)\boldsymbol{w}^{(i-1)} + t\boldsymbol{u}^{(i)}\|_2, \|\boldsymbol{w}^{(i-1)}\|_2)} \|\boldsymbol{u}^{(i)} - \boldsymbol{w}^{(i-1)}\|_2^2 \\
&\leq -\beta \left\langle \mathbb{E}_{\mathcal{D}}[g_{\boldsymbol{w}^{(i-1)}}(\mathbf{x},\mathrm{y})], \mathbb{E}_{\hat{\mathcal{D}}^{(i)}}[g_{\boldsymbol{w}^{(i-1)}}(\mathbf{x},\mathrm{y})] \right\rangle + \frac{\beta^2 L}{2} \left\| \mathbb{E}_{\hat{\mathcal{D}}^{(i)}}[g_{\boldsymbol{w}^{(i-1)}}(\mathbf{x},\mathrm{y})] \right\|_2^2 \quad (4)
\end{aligned}
$$

where the first term of inequality (i) holds because $\mathbf{x} = \boldsymbol{w}^{(i-1)\otimes 2}\mathbf{x} + \mathbf{x}_{\boldsymbol{w}^{(i-1)}\perp}$ and $\langle \mathbf{z}_{\boldsymbol{w}^{(i-1)}\perp}, \boldsymbol{w}^{(i-1)\otimes 2}\mathbf{x}\rangle = 0$ for any $\mathbf{z} \in \mathbb{R}^d$, which implies $\langle \mathbf{x}, \mathbb{E}[g_{\boldsymbol{w}^{(i-1)}}(\mathbf{x},\mathrm{y})]\rangle = \langle \mathbf{x}_{\boldsymbol{w}^{(i-1)}\perp}, \mathbb{E}[g_{\boldsymbol{w}^{(i-1)}}(\mathbf{x},\mathrm{y})]\rangle$, the second term holds due to our assumption, the last inequality is obtained by observing that $\mathbb{E}_{\hat{\mathcal{D}}^{(i)}}[g_{\boldsymbol{w}^{(i-1)}}(\mathbf{x},\mathrm{y})]$ lies on the orthogonal subspace of $\boldsymbol{w}^{(i-1)}$ so that $\|\boldsymbol{u}^{(i)}\|_2^2 = \|\boldsymbol{w}^{(i-1)}\|_2^2 + \beta\|\mathbb{E}_{\hat{\mathcal{D}}^{(i)}}[g_{\boldsymbol{w}^{(i-1)}}(\mathbf{x},\mathrm{y})]\|_2^2 \geq \|\boldsymbol{w}^{(i-1)}\|_2^2$ and that $1 = \|\boldsymbol{w}^{(i-1)}\|_2^2 \leq \|(1-t)\boldsymbol{w}^{(i-1)} + t\boldsymbol{u}^{(i)}\|_2^2$ because of the projection step (line 6) of Algorithm 2 and that $(1-t)\boldsymbol{w}^{(i-1)} + t\boldsymbol{u}^{(i)}$ is a convex combination of $\boldsymbol{u}^{(i)}$ and $\boldsymbol{w}^{(i-1)}$.

Now, since $\mathcal{L}_{\mathcal{D}}(\boldsymbol{w}) = \mathbb{E}_{(\mathbf{x},\mathrm{y})\sim\mathcal{D}}[\mathrm{y}\cdot\max(0,\langle\mathbf{x},\boldsymbol{w}\rangle)]$ by definition, there is $\mathcal{L}_{\mathcal{D}}(\boldsymbol{w}) = \|\boldsymbol{w}\|_2 \cdot \mathcal{L}_{\mathcal{D}}(\bar{\boldsymbol{w}})$, which, along with the fact that $\|\boldsymbol{u}^{(i)}\|_2 \geq \|\boldsymbol{w}^{(i-1)}\|_2 = 1$, indicates $\mathcal{L}_{\mathcal{D}}(\boldsymbol{w}^{(i)}) \leq \mathcal{L}_{\mathcal{D}}(\boldsymbol{u}^{(i)})$. Therefore, applying $\mathcal{L}_{\mathcal{D}}(\boldsymbol{w}^{(i)}) \leq \mathcal{L}_{\mathcal{D}}(\boldsymbol{u}^{(i)})$ to the LHS of inequality (4) gives

$$
\begin{aligned}
\mathcal{L}_{\mathcal{D}}(\boldsymbol{w}^{(i)}) - \mathcal{L}_{\mathcal{D}}(\boldsymbol{w}^{(i-1)}) \leq &-\beta \left\langle \mathbb{E}_{\mathcal{D}}[g_{\boldsymbol{w}^{(i-1)}}(\mathbf{x},\mathrm{y})], \mathbb{E}_{\hat{\mathcal{D}}^{(i)}}[g_{\boldsymbol{w}^{(i-1)}}(\mathbf{x},\mathrm{y})] \right\rangle \\
&+ \frac{\beta^2 L}{2} \left\| \mathbb{E}_{\hat{\mathcal{D}}^{(i)}}[g_{\boldsymbol{w}^{(i-1)}}(\mathbf{x},\mathrm{y})] \right\|_2^2 .
\end{aligned}
$$

Then, conditioning on the previous samples $\hat{\mathcal{D}}^{(1)}, \dots, \hat{\mathcal{D}}^{(i-1)}$, we have, by the independence between $\mathcal{D}$ and $\hat{\mathcal{D}}^{(i)}$ and Jensen's inequality, that

$$
\begin{aligned}
&\mathbb{E}_{\hat{\mathcal{D}}^{(i)}\sim\mathcal{D}}[\mathcal{L}_{\mathcal{D}}(\boldsymbol{w}^{(i)}) - \mathcal{L}_{\mathcal{D}}(\boldsymbol{w}^{(i-1)}) \mid \hat{\mathcal{D}}^{(1)}, \dots, \hat{\mathcal{D}}^{(i-1)}] \\
&= -\beta \left\| \mathbb{E}_{\mathcal{D}}[g_{\boldsymbol{w}^{(i-1)}}(\mathbf{x},\mathrm{y})] \right\|_2^2 + \frac{\beta^2 L}{2} \mathbb{E}_{\hat{\mathcal{D}}^{(i)}\sim\mathcal{D}}\left[ \left\| \mathbb{E}_{\hat{\mathcal{D}}^{(i)}}[g_{\boldsymbol{w}^{(i-1)}}(\mathbf{x},y)] \right\|_2^2 \right] \\
&\overset{(i)}{\leq} -\beta \left\| \mathbb{E}_{\mathcal{D}}[g_{\boldsymbol{w}^{(i-1)}}(\mathbf{x},\mathrm{y})] \right\|_2^2 + \frac{\beta^2 L}{2} \mathbb{E}_{\hat{\mathcal{D}}^{(i)}\sim\mathcal{D}}\left[ \mathbb{E}_{\hat{\mathcal{D}}^{(i)}}\left[ \|g_{\boldsymbol{w}^{(i-1)}}(\mathbf{x},y)\|_2^2 \right] \right] \\
&\leq -\beta \left\| \mathbb{E}_{\mathcal{D}}[g_{\boldsymbol{w}^{(i-1)}}(\mathbf{x},\mathrm{y})] \right\|_2^2 + \beta^2 K^2 Ld/2
\end{aligned}
$$

where inequality (i) is obtained by applying Jensen's inequality to the second term, and the last inequality holds because $\mathbb{E}_{\hat{\mathcal{D}}^{(i)}\sim\mathcal{D}}[\mathbb{E}_{\hat{\mathcal{D}}^{(i)}}[\|g_{\boldsymbol{w}^{(i-1)}}(\mathbf{x},y)\|_2^2]] = \mathbb{E}_{\mathcal{D}}[\|g_{\boldsymbol{w}^{(i-1)}}(\mathbf{x},y)\|_2^2]$ and property (3) of

lemma B.5 gives $\mathbb{E}_{\mathcal{D}}[\|g_{\boldsymbol{w}^{(i-1)}}(\mathbf{x}, y)\|_2^2] \leq d \max_{\|\boldsymbol{u}\|_2=1} \hat{\|}\langle \mathbf{x}, \boldsymbol{u}\rangle\hat{\|}_2^2$, where $\max_{\|\boldsymbol{u}\|_2=1} \hat{\|}\langle \mathbf{x}, \boldsymbol{u}\rangle\hat{\|}_2^2 \leq K^2$ because of our assumption on $\mathcal{D}_{\mathbf{x}}$. Averaging the above inequality over all $T$ iterations and using the *law of total expectation* gives

$$\frac{1}{T}\sum_{i=1}^{T}\left\|\mathbb{E}_{\mathcal{D}}[g_{\boldsymbol{w}^{(i)}}(\mathbf{x}, y)]\right\|_2^2 \leq \frac{\mathcal{L}_{\mathcal{D}}(\boldsymbol{w}^{(0)}) - \mathbb{E}_{\hat{\mathcal{D}}^{(T+1)}\sim\mathcal{D}}[\mathcal{L}_{\mathcal{D}}(\boldsymbol{w}^{(T+1)})]}{\beta T} + \beta K^2 L d/2$$

$$\overset{(i)}{\leq} \frac{\mathcal{L}_{\mathcal{D}}(\boldsymbol{w}^{(0)})}{\beta T} + \beta K^2 L d/2$$

$$\leq \frac{K}{\beta T} + \beta K^2 L d/2$$

where inequality (i) holds because $\mathcal{L}_{\mathcal{D}}(\boldsymbol{w}) \geq 0$, the last inequality is derived by property (1) of lemma B.5 with $\|\boldsymbol{w}^{(i)}\|_2 = 1$ for all $i = 0, \ldots, T+1$ and the $K$-bounded property of $\mathcal{D}_{\mathbf{x}}$. Taking $\beta = \sqrt{2/TKLd}$ gives the first claim.

To obtain the high-probability version, we define

$$G_T(\boldsymbol{w}^{(1)}, \ldots, \boldsymbol{w}^{(T)}) = \frac{1}{T}\sum_{i=1}^{T}\left\|\mathbb{E}_{(\mathbf{x},y)\sim\mathcal{D}}[g_{\boldsymbol{w}^{(i)}}(\mathbf{x}, y)]\right\|_2^2$$

which implies

$$\left|G_T(\boldsymbol{w}^{(1)}, \ldots, \boldsymbol{w}^{(i)}, \ldots, \boldsymbol{w}^{(T)}) - G_T(\boldsymbol{w}^{(1)}, \ldots, \boldsymbol{w}^{(i)'}, \ldots, \boldsymbol{w}^{(T)})\right|$$

$$\leq \frac{1}{T}\left|\|\mathbb{E}[g_{\boldsymbol{w}^{(i)}}(\mathbf{x}, y)]\|_2^2 - \|\mathbb{E}[g_{\boldsymbol{w}^{(i)'}}(\mathbf{x}, y)]\|_2^2\right|$$

$$\leq \frac{2dK^2}{T}$$

where the last step holds due to property (2) of lemma B.5 and that $\mathcal{D}_{\mathbf{x}}$ is $K$-bounded. Now using lemma B.4, we get

$$\Pr\left\{G_T(\boldsymbol{w}^{(1)}, \ldots, \boldsymbol{w}^{(T)}) - \mathbb{E}_{\hat{\mathcal{D}}^{(1)}, \ldots, \hat{\mathcal{D}}^{(T)}\sim\mathcal{D}}[G_T(\boldsymbol{w}^{(1)}, \ldots, \boldsymbol{w}^{(T)})] \geq t\right\} \leq \exp\left(-t^2 T/2K^4 d^2\right)$$

Choosing $T \geq (2K^3 Ld + 8K^4 d^2 \ln(1/\delta))/\epsilon^4$ gives both $\mathbb{E}[G_T(\boldsymbol{w}^{(1)}, \ldots, \boldsymbol{w}^{(T)})] \leq \epsilon^2/2$, by our first claim, and

$$\Pr\left\{\frac{1}{T}\sum_{i=1}^{T}\|\mathbb{E}[g_{\boldsymbol{w}^{(i)}}(\mathbf{x}, y)]\|_2^2 \leq \epsilon^2\right\} = \Pr\left\{G_T(\mathbf{w}^{(1)}, \ldots, \mathbf{w}^{(T)}) \leq \epsilon^2\right\}$$

$$\geq 1 - \delta$$

Finally, since $\min_{i=1,\ldots,T}\|\mathbb{E}[g_{\boldsymbol{w}^{(i)}}(\mathbf{x}, y)]\|_2^2$ is at most the average, we obtain the second claim. $\qquad\square$

An immediate consequence of Lemma B.1 and Proposition B.2 is as follow.

**Corollary B.3** (Proposition 3.3). *Let $\mathcal{D}$ be any distribution on $\mathbb{R}^d \times \{0, 1\}$ with standard normal $\mathbf{x}$-marginal. Define $\mathcal{L}_{\mathcal{D}}(\boldsymbol{w}) = \mathbb{E}_{(\mathbf{x},y)\sim\mathcal{D}}[y \cdot \max(0, \langle \mathbf{x}, \boldsymbol{w}\rangle)]$ and $g_{\boldsymbol{w}}(\mathbf{x}, y) = y \cdot \mathbf{x}_{\boldsymbol{w}^\perp} \mathbb{1}\{\mathbf{x} \in h(\boldsymbol{w})\}$, then, with $\beta = \sqrt{1/Td}$, after $T \geq 12d^2 \ln(1/\delta)/\epsilon^4$ iterations, the output $(\boldsymbol{w}^{(1)}, \ldots, \boldsymbol{w}^{(T)})$ in algorithm 2 will satisfies that $\min_{i=1,\ldots,T}\|\mathbb{E}_{(\mathbf{x},y)\sim\mathcal{D}}[g_{\boldsymbol{w}^{(i)}}(\mathbf{x}, y)]\|_2 \leq \epsilon$, with probability at least $1 - \delta$.*

Below are a few tools we needed in the proof of proposition B.2.

**Lemma B.4** (Theorem 2.2 of Devroye & Lugosi (2001)). *Suppose that $\mathbf{x}_1, \ldots, \mathbf{x}_d \in \mathcal{X}$ are independent random variables, and let $f : \mathcal{X}^d \to \mathbb{R}$. Let $c_1, \ldots, c_n$ satisfies*

$$\sup_{\mathbf{x}_1,\ldots,\mathbf{x}_d,\mathbf{x}_i'}|f(\mathbf{x}_1, \ldots, \mathbf{x}_i, \ldots, \mathbf{x}_d) - f(\mathbf{x}_1, \ldots, \mathbf{x}_i', \ldots, \mathbf{x}_d)| \leq c_i$$

*for $i \in [d]$. Then*

$$\Pr\{f(\mathbf{x}) - \mathbb{E}[f(\mathbf{x})] \geq t\} \leq \exp\left(-\frac{2t^2}{\sum_{i\in[d]}c_2^2}\right).$$

**Lemma B.5.** *Let $\mathcal{D}$ be any distribution on $\mathbb{R}^d \times [-1, +1]$. Define $\mathcal{L}_{\mathcal{D}}(\boldsymbol{w}) = \mathbb{E}_{(\mathbf{x}, \mathbf{y}) \sim \mathcal{D}}[\mathbf{y} \cdot \max(0, \langle \mathbf{x}, \boldsymbol{w} \rangle)]$ and $g_{\boldsymbol{w}}(\mathbf{x}, \mathbf{y}) = \mathbf{y} \cdot \mathbf{x}_{\boldsymbol{w}^{\perp}} \mathbb{1}\{\mathbf{x} \in h(\boldsymbol{w})\}$, then, for any $\boldsymbol{w} \in \mathbb{R}^d$, we have the following properties:*

1. *$\mathcal{L}_{\mathcal{D}}(\boldsymbol{w}) \leq \|\boldsymbol{w}\|_2 \hat{\|} \langle \mathbf{x}, \bar{\boldsymbol{w}} \rangle \hat{\|}_1$,*

2. *$\|\mathbb{E}_{(\mathbf{x}, \mathbf{y}) \sim \mathcal{D}}[g_{\boldsymbol{w}}(\mathbf{x}, \mathbf{y})]\|_2 \leq \sqrt{d} \max_{\|\mathbf{u}\|_2 = 1} \hat{\|} \langle \mathbf{x}, \boldsymbol{u} \rangle \hat{\|}_1$,*

3. *$\mathbb{E}_{(\mathbf{x}, \mathbf{y}) \sim \mathcal{D}}[\|g_{\boldsymbol{w}}(\mathbf{x}, \mathbf{y})\|_2^2] \leq d \max_{\|\boldsymbol{u}\|_2 = 1} \hat{\|} \langle \mathbf{x}, \boldsymbol{u} \rangle \hat{\|}_2^2$.*

*Proof.* To show the first claim, notice that $\mathbf{y} \leq 1$, so we have

$$
\begin{aligned}
\mathcal{L}_{\mathcal{D}}(\boldsymbol{w}) &= \mathbb{E}[\mathbf{y} \cdot \max(0, \langle \mathbf{x}, \boldsymbol{w} \rangle)] \\
&\leq \mathbb{E}[\langle \mathbf{x}, \boldsymbol{w} \rangle \cdot \mathbb{1}\{\langle \mathbf{x}, \boldsymbol{w} \rangle \geq 0\}] \\
&\leq \mathbb{E}[|\langle \mathbf{x}, \boldsymbol{w} \rangle|] \\
&= \|\boldsymbol{w}\|_2 \hat{\|} \langle \mathbf{x}, \bar{\boldsymbol{w}} \rangle \hat{\|}_1
\end{aligned}
$$

where inequality (i) holds because $(\mathbb{E}[\mathbf{x}^p])^{1/p}$ is a increasing function in $p$, and the last inequality holds since $\mathcal{D}$ is in isotropic position.

To prove property (2), because, again, $\mathbf{y} \leq 1$, we have

$$
\begin{aligned}
\|\mathbb{E}[\mathbf{y} \cdot \mathbf{x}_{\boldsymbol{w}^{\perp}} \mathbb{1}\{\mathbf{x} \in h(\boldsymbol{w})\}]\|_2 &\leq \|\mathbb{E}[|\mathbf{x}|]\|_2 \\
&\leq \sqrt{d} \max_{i \in [d]} \mathbb{E}[|\mathbf{x}_i|] \\
&\leq \sqrt{d} \max_{\|\mathbf{u}\|_2 = 1} \hat{\|} \langle \mathbf{x}, \boldsymbol{u} \rangle \hat{\|}_1
\end{aligned}
$$

where the absolute operator on the RHS of the first inequality is an element-wise operation.

To obtain the last property, notice that $\|\mathbf{x}_{\boldsymbol{w}^{\perp}}\|_2 \leq \|\mathbf{x}\|_2$ because $\mathbf{x}_{\boldsymbol{w}^{\perp}}$ is a projection of $\mathbf{x}$, then we have

$$
\begin{aligned}
\mathbb{E}[\|\mathbf{y} \cdot \mathbf{x}_{\boldsymbol{w}^{\perp}} \mathbb{1}\{\langle \mathbf{x}, \boldsymbol{w} \rangle > 0\}\|_2^2] &\leq \mathbb{E}[\|\mathbf{x}\|_2^2] \\
&\leq d \max_{\|\boldsymbol{u}\|_2 = 1} \hat{\|} \langle \mathbf{x}, \boldsymbol{u} \rangle \hat{\|}_2^2.
\end{aligned}
$$

$\square$

# C OPTIMALITY ANALYSIS OF APPROXIMATE STATIONARY POINT

We present our analysis for the main theorem of our algorithmic results in this section.

**Theorem C.1** (Theorem 3.1). *Let $\mathcal{D}$ be a distribution on $\mathbb{R}^d \times \{0, 1\}$ with standard normal $\mathbf{x}$-marginal, and $\mathcal{C}$ be a class of binary classifiers on $\mathbb{R}^d \times \{0, 1\}$. If there exists a unit vector $\boldsymbol{v} \in \mathbb{R}^d$ and a $c \in \mathcal{C}$ such that, for some sufficiently small $\epsilon \in [0, 1/e]$, $\Pr_{(\mathbf{x}, \mathbf{y}) \sim \mathcal{D}}\{\mathbf{x} \in h(\boldsymbol{v}) \cap c(\mathbf{x}) \neq \mathbf{y}\} \leq \epsilon$, then, with at most $\tilde{O}(d^2/\epsilon^6)$ examples, Algorithm 1 will return a $\boldsymbol{w}^{(c')}$, with probability at least $1 - \delta$, such that $\Pr_{(\mathbf{x}, \mathbf{y}) \sim \mathcal{D}}\{\mathbf{x} \in h(\boldsymbol{w}^{(c')}) \cap c'(\mathbf{x}) \neq \mathbf{y}\} = \tilde{O}(\sqrt{\epsilon})$ and run in time $O(d^2 |\mathcal{C}| / \epsilon^6)$.*

*Proof.* For conciseness of the proof, let the error indicator function $f_{\boldsymbol{w}}^{(c)} : \mathbb{R}^d \times \{0, 1\} \to \{0, 1\}$ be such that $f_{\boldsymbol{w}}^{(c)}(\mathbf{x}, \mathbf{y}) = \mathbb{1}\{\mathbf{x} \in h(\boldsymbol{w}) \cap c(\mathbf{x}) \neq \mathbf{y}\}$.

Consider the $c \in \mathcal{C}$ that satisfies $\min_{\boldsymbol{w}} \Pr_{\mathcal{D}}\left\{f_{\boldsymbol{w}}^{(c)}(\mathbf{x}, \mathbf{y}) = 1\right\} \leq \epsilon$. For $T = 12d^2 \ln(8/\delta_1)/\epsilon^4$, $N \geq \Omega(\ln(16T/\delta_1)/\epsilon^2 \ln \epsilon^{-1})$, lemma C.5 and a union bound over the two calls of algorithm 2 guarantees that there exists a $\boldsymbol{w}' \in \mathcal{W}^{(c)}$ such that $\Pr_{(\mathbf{x}, \mathbf{y}) \sim \mathcal{D}}\{f_{\boldsymbol{w}'}^{(c)}(\mathbf{x}, \mathbf{y})\} \leq \frac{5}{2}(\epsilon \sqrt{\ln \epsilon^{-1}})^{1/2}$ with probability at least $1 - \delta_1/2$.

While estimating each $w \in \mathcal{W}^{(c)}$ at line 9 with $\ln(4T/\delta_1)/2\epsilon$ samples in $\hat{\mathcal{D}}$, we know that

$$\Pr\left\{\left|\mathbb{E}_{\hat{\mathcal{D}}}[f_{w}^{(c)}(\mathbf{x}, y)] - \mathbb{E}_{\mathcal{D}}[f_{w}^{(c)}(\mathbf{x}, y)]\right| > \sqrt{\epsilon}\right\} \le \delta_1/2T$$

by lemma F.4. Taking a union bound over all $w \in \mathcal{W}^{(c)}$ gives

$$\Pr\left\{\mathbb{E}_{\hat{\mathcal{D}}}[f_{w^{(c)}}^{(c)}(\mathbf{x}, y)] > \mathbb{E}_{\mathcal{D}}[f_{w'}^{(c)}(\mathbf{x}, y)] + 2\sqrt{\epsilon}\right\} \le \delta_1$$

Therefore, we can conclude that $\Pr_{(\mathbf{x}, y) \sim \mathcal{D}}\left\{\mathbf{x} \in h(w^{(c)}) \cap c(\mathbf{x}) \ne y\right\} = \tilde{O}(\sqrt{\epsilon})$ with probability at least $1 - \delta_1$ in this iteration.

Finally, taking an union bound again over all $c \in \mathcal{C}$ and choosing $\delta_1 = \delta/|\mathcal{C}|$, we know that the total number of examples needed is $O(TN) = O(d^2 \ln(16T|\mathcal{C}|/\delta)/\epsilon^6) = \tilde{O}(d^2/\epsilon^6)$ and the running time is simply $O(|\mathcal{C}|TN) = \tilde{O}(d^2|\mathcal{C}|/\epsilon^6)$, since we can reuse the example for each $c \in \mathcal{C}$. □

**Proposition C.2** (Proposition 3.2). *Let $\mathcal{D}$ be a distribution on $\mathbb{R}^d \times \{0, 1\}$ with standard normal $\mathbf{x}$-marginal, and $g_w(\mathbf{x}, y) = y \cdot \mathbf{x}_{w^\perp} \mathbb{1}\{\mathbf{x} \in h(w)\}$. Suppose $v, w \in \mathbb{R}^d$ are unit vectors such that $\Pr_{(\mathbf{x}, y) \sim \mathcal{D}}\{\mathbf{x} \in h(v) \cap y = 1\} \le \epsilon$ and $\theta(v, w) \in [0, \pi/2)$, then, if $\Pr_{(\mathbf{x}, y) \sim \mathcal{D}}\{\mathbf{x} \in h(w) \cap y = 1\} \ge \frac{5}{2}(\epsilon\sqrt{\ln \epsilon^{-1}})^{1/2}$, it holds that*

$$\left\langle \mathbb{E}_{(\mathbf{x}, y) \sim \mathcal{D}}[-g_w(\mathbf{x}, y)], \bar{v}_{w^\perp} \right\rangle \ge \frac{2}{5}\epsilon\sqrt{\ln \epsilon^{-1}}$$

*for some sufficiently small $\epsilon \in [0, 1/e]$.*

*Proof.* For conciseness, let $\theta = \theta(v, w)$ and define two orthonormal basis $e_1, e_2$ such that $w = e_2$ and $v = -e_1 \sin\theta + e_2 \cos\theta$, which implies $e_1 = -\bar{v}_{w^\perp}$. Denote $\mathbf{x}_i = \langle \mathbf{x}, e_i \rangle$ so that $\langle \mathbf{x}, w \rangle = \mathbf{x}_2$ and $\langle \mathbf{x}, v \rangle = -\mathbf{x}_1 \sin\theta + \mathbf{x}_2 \cos\theta$. Because $\langle \mathbf{x}, e_1 \rangle = \langle \mathbf{x}_2 e_2 + \mathbf{x}_{e_2^\perp}, e_1 \rangle = -\langle \mathbf{x}_{w^\perp}, \bar{v}_{w^\perp} \rangle$, we have

$$
\begin{aligned}
\langle \mathbb{E}[-g_w(\mathbf{x}, y)], \bar{v}_{w^\perp} \rangle &= -\mathbb{E}[y \cdot \langle \mathbf{x}_{w^\perp}, \bar{v}_{w^\perp} \rangle \mathbb{1}\{\mathbf{x} \in h(w)\}] \\
&= \mathbb{E}[y \cdot \langle \mathbf{x}, e_1 \rangle \mathbb{1}\{\mathbf{x} \in h(w)\}] \\
&= \mathbb{E}[y \cdot \mathbf{x}_1 \cdot (\mathbb{1}\{\mathbf{x} \in h(w) \cap h(v)\} + \mathbb{1}\{\mathbf{x} \in h(w) \cap h^c(v)\})] \\
&\ge \underbrace{\mathbb{E}[|\mathbf{x}_1| \mathbb{1}\{\mathbf{x}_1 \tan\theta > \mathbf{x}_2 \ge 0, y = 1\}]}_{I_1} \\
&\quad - \underbrace{\mathbb{E}[|\mathbf{x}_1| \mathbb{1}\{\mathbf{x}_2 \ge 0, \mathbf{x}_2 \ge \mathbf{x}_1 \tan\theta, y = 1\}]}_{I_2}.
\end{aligned}
\tag{5}
$$

where the last inequality holds because $\cos\theta > 0$ by our assumption that $\theta(v, w) \in [0, \pi/2)$, and $h(w) = \{\mathbf{x} \mid \langle \mathbf{x}, w \rangle \ge 0\}, h(v) = \{\mathbf{x} \mid \langle \mathbf{x}, v \rangle \ge 0\}$ imply that $\mathbf{x}_2 \ge 0, \mathbf{x}_2 \ge \mathbf{x}_1 \tan\theta$ by construction. This decomposition above can also be seen from figure 4. Then, we will apply lemma C.3 to bound the above two terms.

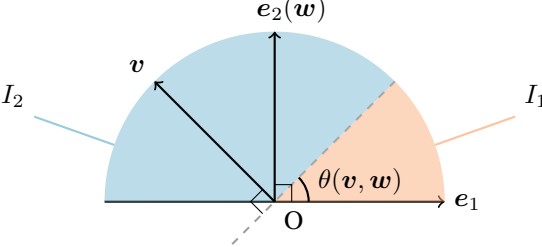

Figure 4: Blue area represent $h(v) \cap h(w)$, while orange area represents $h(w) \cap h^c(v)$.

Observe that, since $\mathbf{x}$ is sampled from a standard normal distribution and $e_1, e_2$ are two orthonormal basis, $\mathbf{x}_1, \mathbf{x}_2$ are two independent one-dimension standard normal random variables. Then, observe that we can bound $I_1$ and $I_2$ by applying lemma C.3 with carefully chosen $\alpha$ and $\beta$.

**To apply lemma C.3 on** $I_2$, by treating $x_2 \geq 0$ to be the event $T$ and the rest to be $S$ in lemma C.3, we show that there exists an $\alpha > 0$ such that $\Pr\{x_2 \geq 0 \cap x_2 \geq x_1 \tan\theta \cap y = 1\} \leq \Pr\{x_2 \geq 0 \cap |x_1| \geq \alpha\}$.

First of all, notice that $\Pr\{x_2 \geq 0 \cap x_2 \geq x_1 \tan\theta \cap y = 1\} = \Pr\{\mathbf{x} \in h(\boldsymbol{v}) \cap y = 1\} \leq \epsilon$ by our assumption. Suppose $\alpha = \sqrt{2\ln\epsilon^{-1} - 2\ln(\kappa\sqrt{\ln\epsilon^{-1}})}$ for some $\kappa > 1$, then, due to the independence between $x_1, x_2$ as well as lemma F.9, there is

$$\Pr\{x_2 \geq 0 \cap |x_1| \geq \alpha\} \geq \frac{\exp\left(-\ln\epsilon^{-1} + \ln(\kappa\sqrt{\ln\epsilon^{-1}})\right)}{\sqrt{2\pi}\left(\sqrt{2\ln\epsilon^{-1} - 2\ln(\kappa\sqrt{\ln\epsilon^{-1}})} + 1\right)}$$

$$= \frac{\epsilon\kappa}{\sqrt{2\pi}\left(\sqrt{2 - 2\ln(\kappa\sqrt{\ln\epsilon^{-1}})/\ln\epsilon^{-1}} + 1/\sqrt{\ln\epsilon^{-1}}\right)}$$

$$\geq \frac{\epsilon\kappa}{\sqrt{2\pi}\left(\sqrt{2} + 1\right)}$$

where the last inequality holds because $\kappa > 1$ and $\epsilon \in [0, 1/e]$ so that $\ln(\kappa\sqrt{\ln\epsilon^{-1}})/\ln\epsilon^{-1} \geq 0$ as well as $\ln\epsilon^{-1} \geq 1$. Taking $\kappa = \sqrt{2\pi}\left(\sqrt{2} + 1\right)$ results to $\Pr\{x_2 \geq 0 \cap |x_1| \geq \alpha\} \geq \epsilon$. Then, lemma C.3 gives

$$I_2 \leq \mathbb{E}[|x_1|\,\mathbb{1}\{x_2 \geq 0, |x_1| \geq \alpha\}]$$

$$= \frac{1}{\sqrt{2\pi}}\int_{\geq\alpha} x_1 e^{-x_1^2/2}dx_1$$

$$= \frac{\exp\left(\ln\epsilon + \ln\left(\sqrt{2\pi}\left(\sqrt{2} + 1\right)\sqrt{\ln\epsilon^{-1}}\right)\right)}{\sqrt{2\pi}}$$

$$\leq 3\epsilon\sqrt{\ln\epsilon^{-1}}. \tag{6}$$

**To apply lemma C.3 on** $I_1$, notice that the event $x_1 \tan\theta > x_2 \geq 0$ in $I_1$ is a subset of event $x_1 \geq 0 \cap x_2 \geq 0$ because $\theta(\boldsymbol{v}, \boldsymbol{w}) \in [0, \pi/2)$. Therefore, we can view the event $x_1 \geq 0 \cap x_2 \geq 0$ as $T$ in lemma C.3 and show that there exists a $\beta > 0$ such that $\Pr\{0 \leq x_1 \leq \beta \cap x_2 \geq 0\} \leq \Pr\{x_1 \tan\theta > x_2 \geq 0 \cap y = 1\}$ to apply lemma C.3.

At first, observe that, by our assumption that $\Pr\{\mathbf{x} \in h(\boldsymbol{v}) \cap y = 1\} \leq \epsilon$ as well as $\Pr\{\mathbf{x} \in h(\boldsymbol{w}) \cap y = 1\} \geq \frac{5}{2}\left(\epsilon\sqrt{\ln\epsilon^{-1}}\right)^{1/2}$, there is

$$\left(e^{-1/2} + e^{1/2}\right)\left(\epsilon\sqrt{\ln\epsilon^{-1}}\right)^{1/2} - \epsilon < \frac{5}{2}\left(\epsilon\sqrt{\ln\epsilon^{-1}}\right)^{1/2} - \epsilon$$

$$\leq \Pr\{\mathbf{x} \in h(\boldsymbol{w}) \cap y = 1\} - \Pr\{\mathbf{x} \in h(\boldsymbol{v}) \cap \mathbf{x} \in h(\boldsymbol{w}) \cap y = 1\}$$

$$= \Pr\{\mathbf{x} \in h^c(\boldsymbol{v}) \cap \mathbf{x} \in h(\boldsymbol{w}) \cap y = 1\}$$

$$= \Pr\{x_1 \tan\theta > x_2 \geq 0 \cap y = 1\}$$

where the first inequality holds because $e^{-1/2} + e^{1/2} \leq 5/2$. Then, taking $\beta = 2\sqrt{2e\pi}\left(\epsilon\sqrt{\ln\epsilon^{-1}}\right)^{1/2}$ yields

$$\Pr\{0 \leq x_1 \leq \beta \cap x_2 \geq 0\} = \frac{1}{2}\Pr\{0 \leq x_1 \leq \beta\}$$

$$\overset{(i)}{\leq} \sqrt{e}\left(\epsilon\sqrt{\ln\epsilon^{-1}}\right)^{1/2}$$

$$= \left(e^{-1/2} + e^{1/2}\right)\left(\epsilon\sqrt{\ln\epsilon^{-1}}\right)^{1/2} - e^{-1/2}\left(\epsilon\sqrt{\ln\epsilon^{-1}}\right)^{1/2}$$

$$\leq \left(e^{-1/2} + e^{1/2}\right)\left(\epsilon\sqrt{\ln\epsilon^{-1}}\right)^{1/2} - \epsilon$$

where the first equation holds because $x_1, x_2$ are independent, inequality (i) holds due to the fact that standard normal density is never greater than $1/\sqrt{2\pi}$, and the last inequality holds because $\epsilon \in [0, 1/e]$ so that $e^{-1/2} \geq \sqrt{\epsilon}/\ln^{1/4}\epsilon^{-1}$. Applying lemma C.3 gives

$$
\begin{aligned}
I_1 &\geq \mathbb{E}[x_1 \cdot \mathbb{1}\{0 \leq x_1 \leq 2\sqrt{2e\pi}(\epsilon\sqrt{\ln\epsilon^{-1}})^{1/2}, x_2 \geq 0\}]\\
&= \frac{1}{2\sqrt{2\pi}}\int_0^{2\sqrt{2e\pi}(\epsilon\sqrt{\ln\epsilon^{-1}})^{1/2}} x_1 e^{-x_1^2/2} dx_1\\
&= \frac{1 - \exp\left(-4e\pi\epsilon\sqrt{\ln\epsilon^{-1}}\right)}{2\sqrt{2\pi}}\\
&\geq \sqrt{\frac{\pi}{2}}e\epsilon\sqrt{\ln\epsilon^{-1}}
\end{aligned}
\tag{7}
$$

where the last inequality holds because of the fundamental inequality $x/2 \leq 1 - e^{-x}$ for $x \in [0, 1.59]$.

At last, since $e\sqrt{\pi/2} - 3 > 2/5$, taking inequalities (7) and (6) back to inequality (5) gives the desired result. $\square$

The following lemma plays a key role in proving proposition 3.2.

**Lemma C.3.** *Let $\mathcal{D}$ be an arbitrary distribution on $\mathbb{R}^d$, and $S, T$ be any events such that $\Pr_{\mathcal{D}}\{S \cap T\} = p$ for some $p \in (0, 1)$. Then, for any unit vector $\boldsymbol{u} \in \mathbb{R}^d$, and parameters $\alpha, \beta$ that satisfies $\Pr\{T \cap |\langle \mathbf{x}, \boldsymbol{u}\rangle| \leq \beta\} \leq p \leq \Pr\{T \cap |\langle \mathbf{x}, \boldsymbol{u}\rangle| \geq \alpha\}$, it holds that*

$$
\mathbb{E}_{\mathcal{D}}[|\langle \mathbf{x}, \boldsymbol{u}\rangle| \mathbb{1}\{T, |\langle \mathbf{x}, \boldsymbol{u}\rangle| \leq \beta\}] \leq \mathbb{E}_{\mathcal{D}}[|\langle \mathbf{x}, \boldsymbol{u}\rangle| \mathbb{1}\{S, T\}] \leq \mathbb{E}_{\mathcal{D}}[|\langle \mathbf{x}, \boldsymbol{u}\rangle| \mathbb{1}\{T, |\langle \mathbf{x}, \boldsymbol{u}\rangle| \geq \alpha\}].
$$

*Proof.* For conciseness of the proof, we denote $\mathcal{E}_{\geq t} = \{\mathbf{x} \mid T \cap |\langle \mathbf{x}, \boldsymbol{u}\rangle| \geq t\}$, $\mathcal{E}_{\leq t} = \{\mathbf{x} \mid T \cap |\langle \mathbf{x}, \boldsymbol{u}\rangle| \leq t\}$, and $\mathcal{E}_S = \{\mathbf{x} \mid S \cap T\}$.

To show the first property, let $\alpha > 0$ be such that $p \leq \Pr\{T \cap |\langle \mathbf{x}, \boldsymbol{u}\rangle| \geq \alpha\} = \Pr\{\mathbf{x} \in \mathcal{E}_{\geq \alpha}\}$. Then, if $\mathbf{x} \in \mathcal{E}_S \backslash \mathcal{E}_{\geq \alpha}$, there must be $|\langle \mathbf{x}, \boldsymbol{u}\rangle| \leq \alpha$. Therefore, we have

$$
\begin{aligned}
\mathbb{E}_{\mathbf{x}\sim\mathcal{D}}[|\langle \mathbf{x}, \boldsymbol{u}\rangle| \mathbb{1}\{S, T\}] &= \mathbb{E}[|\langle \mathbf{x}, \boldsymbol{u}\rangle| \mathbb{1}\{\mathbf{x} \in \mathcal{E}_S\}]\\
&= \mathbb{E}[|\langle \mathbf{x}, \boldsymbol{u}\rangle| \mathbb{1}\{\mathbf{x} \in \mathcal{E}_S \cap \mathcal{E}_{\geq \alpha}\}] + \mathbb{E}[|\langle \mathbf{x}, \boldsymbol{u}\rangle| \mathbb{1}\{\mathbf{x} \in \mathcal{E}_S \backslash \mathcal{E}_{\geq \alpha}\}]\\
&\leq \mathbb{E}[|\langle \mathbf{x}, \boldsymbol{u}\rangle| \mathbb{1}\{\mathbf{x} \in \mathcal{E}_S \cap \mathcal{E}_{\geq \alpha}\}] + \mathbb{E}[\alpha\mathbb{1}\{\mathbf{x} \in \mathcal{E}_S \backslash \mathcal{E}_{\geq \alpha}\}]\\
&\overset{(i)}{\leq} \mathbb{E}[|\langle \mathbf{x}, \boldsymbol{u}\rangle| \mathbb{1}\{\mathbf{x} \in \mathcal{E}_S \cap \mathcal{E}_{\geq \alpha}\}] + \mathbb{E}[|\langle \mathbf{x}, \boldsymbol{u}\rangle| \mathbb{1}\{\mathbf{x} \in \mathcal{E}_{\geq \alpha} \backslash \mathcal{E}_S\}]\\
&= \mathbb{E}[|\langle \mathbf{x}, \boldsymbol{u}\rangle| \mathbb{1}\{T, |\langle \mathbf{x}, \boldsymbol{u}\rangle| \geq \alpha\}]
\end{aligned}
$$

where inequality (i) holds because $\Pr\{\mathbf{x} \in \mathcal{E}_S\} \leq \Pr\{\mathbf{x} \in \mathcal{E}_{\geq \alpha}\}$ by construction, which implies $\Pr\{\mathbf{x} \in \mathcal{E}_S \backslash \mathcal{E}_{\geq \alpha}\} \leq \Pr\{\mathbf{x} \in \mathcal{E}_{\geq \alpha} \backslash \mathcal{E}_S\}$, and every $\mathbf{x} \in \mathcal{E}_{\geq \alpha}$ satisfies $|\langle \mathbf{x}, \boldsymbol{u}\rangle| \geq \alpha$.

To prove the second claim, we similarly define $\beta > 0$ be such that $p \geq \Pr\{T \cap |\langle \mathbf{x}, \boldsymbol{u}\rangle| \leq \beta\} = \Pr\{\mathbf{x} \in \mathcal{E}_{\leq \beta}\}$. Similar to the case of $|\langle \mathbf{x}, \boldsymbol{u}\rangle| \leq \alpha$, we should notice that, if $\mathbf{x} \in \mathcal{E}_S \backslash \mathcal{E}_{\leq \beta}$, there is $|\langle \mathbf{x}, \boldsymbol{u}\rangle| \geq \beta$. Hence, with a similar argument as above, we have

$$
\begin{aligned}
\mathbb{E}_{\mathbf{x}\sim\mathcal{D}}[|\langle \mathbf{x}, \boldsymbol{u}\rangle| \mathbb{1}\{S, T\}] &= \mathbb{E}[|\langle \mathbf{x}, \boldsymbol{u}\rangle| \mathbb{1}\{\mathbf{x} \in \mathcal{E}_S\}]\\
&= \mathbb{E}[|\langle \mathbf{x}, \boldsymbol{u}\rangle| \mathbb{1}\{\mathbf{x} \in \mathcal{E}_S \cap \mathcal{E}_{\leq \beta}\}] + \mathbb{E}[|\langle \mathbf{x}, \boldsymbol{u}\rangle| \mathbb{1}\{\mathbf{x} \in \mathcal{E}_S \backslash \mathcal{E}_{\leq \beta}\}]\\
&\geq \mathbb{E}[|\langle \mathbf{x}, \boldsymbol{u}\rangle| \mathbb{1}\{\mathbf{x} \in \mathcal{E}_S \cap \mathcal{E}_{\leq \beta}\}] + \mathbb{E}[\beta\mathbb{1}\{\mathbf{x} \in \mathcal{E}_S \backslash \mathcal{E}_{\leq \beta}\}]\\
&\geq \mathbb{E}[|\langle \mathbf{x}, \boldsymbol{u}\rangle| \mathbb{1}\{\mathbf{x} \in \mathcal{E}_S \cap \mathcal{E}_{\leq \beta}\}] + \mathbb{E}[|\langle \mathbf{x}, \boldsymbol{u}\rangle| \mathbb{1}\{\mathbf{x} \in \mathcal{E}_{\leq \beta} \backslash \mathcal{E}_S\}]\\
&= \mathbb{E}[|\langle \mathbf{x}, \boldsymbol{u}\rangle| \mathbb{1}\{T, |\langle \mathbf{x}, \boldsymbol{u}\rangle| \leq \beta\}]
\end{aligned}
$$

which completes the proof. $\square$

The following corollary is an immediate result of Proposition C.2.

**Corollary C.4.** *Let $\mathcal{D}$ be a distribution on $\mathbb{R}^d \times \{0,1\}$ with standard normal $\mathbf{x}$-marginal, and $g_{\boldsymbol{w}}(\mathbf{x}, \mathrm{y}) = \mathrm{y} \cdot \mathbf{x}_{\boldsymbol{w}^\perp} \mathbb{1}\{\mathbf{x} \in h(\boldsymbol{w})\}$. Suppose $\boldsymbol{v}, \boldsymbol{w} \in \mathbb{R}^d$ are unit vectors such that $\mathrm{Pr}_{(\mathbf{x},\mathrm{y})\sim\mathcal{D}}\{\mathbf{x} \in h(\boldsymbol{v}) \cap \mathrm{y} = 1\} \leq \epsilon$ and $\theta(\boldsymbol{v}, \boldsymbol{w}) \in [0, \pi/2)$, then, if a unit vector $\boldsymbol{w}$ satisfies that $\|\mathbb{E}_{(\mathbf{x},\mathrm{y})\sim\mathcal{D}}[g_{\boldsymbol{w}}(\mathbf{x},\mathrm{y})]\|_2 < \frac{2}{5}\epsilon\sqrt{\ln \epsilon^{-1}}$, we have*

$$\mathrm{Pr}_{(\mathbf{x},\mathrm{y})\sim\mathcal{D}}\{\mathbf{x} \in h(\boldsymbol{w}) \cap \mathrm{y} = 1\} < \frac{5}{2}(\epsilon\sqrt{\ln \epsilon^{-1}})^{1/2}$$

*for some small enough $\epsilon \in [0, 1/e]$.*

*Proof.* By Cauchy's inequality and our assumption, it holds that

$$\left\langle \mathbb{E}_{(\mathbf{x},\mathrm{y})\sim\mathcal{D}}[-g_{\boldsymbol{w}}(\mathbf{x}, \mathrm{y})], \bar{\boldsymbol{v}}_{\boldsymbol{w}^\perp} \right\rangle \leq \left\| \mathbb{E}_{(\mathbf{x},\mathrm{y})\sim\mathcal{D}}[g_{\boldsymbol{w}}(\mathbf{x}, \mathrm{y})] \right\|_2 < \frac{2}{5}\epsilon\sqrt{\ln \epsilon^{-1}}$$

Then, negating Proposition 3.2 gives the desired result. $\square$

Now we are ready to prove that at least one of the halfspaces selector returned by the Projected SGD is close to the optimal one of the classifier $c \in \mathcal{C}$ in one iteration in Algorithm 1.

**Lemma C.5** (Lemma 3.4). *Let $\mathcal{D}$ be a distribution on $\mathbb{R}^d \times \{0,1\}$ with standard normal $\mathbf{x}$-marginal, and $g_{\boldsymbol{w}}(\mathbf{x}, \mathrm{y}) = \mathrm{y} \cdot \mathbf{x}_{\boldsymbol{w}^\perp} \mathbb{1}\{\mathbf{x} \in h(\boldsymbol{w})\}$. Suppose $\boldsymbol{v} \in \mathbb{R}^d$ is a unit vectors such that $\mathrm{Pr}_{(\mathbf{x},\mathrm{y})\sim\mathcal{D}}\{\mathbf{x} \in h(\boldsymbol{v}) \cap \mathrm{y} = 1\} \leq \epsilon$, if $T \geq 12d^2 \ln(2/\delta)/\epsilon^4$, $N \geq \ln(4T/\delta)/C\epsilon^2 \ln \epsilon^{-1}$ for some constant $C > 0$, and $\theta(\boldsymbol{v}, \boldsymbol{w}^{(0)}) \in [0, \pi/2)$, it holds that at least one of $\boldsymbol{w} \in \mathcal{W}$ returned by algorithm 2 will satisfies*

$$\mathrm{Pr}_{(\mathbf{x},\mathrm{y})\sim\mathcal{D}}\{\mathbf{x} \in h(\boldsymbol{w}) \cap \mathrm{y} = 1\} \leq \frac{5}{2}(\epsilon\sqrt{\ln \epsilon^{-1}})^{1/2}$$

*with probability at least $1 - \delta$ for some sufficiently small $\epsilon \in [0, 1/e]$.*

*Proof.* By Corollary B.3 with $T \geq 12d^2 \ln(2/\delta)/\epsilon^4$, there exists a $\boldsymbol{w} \in \mathcal{W}$ such that $\|\mathbb{E}_{\mathcal{D}}[g_{\boldsymbol{w}}(\mathbf{x}, \mathrm{y})]\|_2 \leq \epsilon$ with probability at least $1 - \delta/2$. Suppose $\boldsymbol{w} \in \mathcal{W}$ is indexed in the same order that the iterations happened in algorithm 2, and let $\boldsymbol{w}^{(t)}$ be the first parameter in that order such that $\|\mathbb{E}_{\mathcal{D}}[g_{\boldsymbol{w}^{(t)}}(\mathbf{x}, \mathrm{y})]\|_2 \leq \epsilon$.

Consider now the subset $S = \{\boldsymbol{w}^{(1)}, \ldots, \boldsymbol{w}^{(t-1)}\} \subset \mathcal{W}$, there are two possible cases, either there already exists a $\boldsymbol{w} \in S$ such that $\mathrm{Pr}\{h(\mathbf{x}, \boldsymbol{w}) \geq 0 \cap \mathrm{y} = 1\} \leq \frac{5}{2}(\epsilon\sqrt{\ln \epsilon^{-1}})^{1/2}$, or none of them have low error rate. The former case already satisfies the desired requirement, hence, we will focus on prove the latter case also implies the existence of a good parameter.

We first show that, by induction, every $\boldsymbol{w}^{(i)} \in \{\boldsymbol{w}^{(0)}, \ldots, \boldsymbol{w}^{(t)}\}$ satisfies $\theta(\boldsymbol{v}, \boldsymbol{w}^{(i)}) \in [0, \pi/2)$ with high probability.

For $\boldsymbol{w}^{(0)} = \boldsymbol{e}_1$, since we assumed $\theta(\boldsymbol{e}_1, \boldsymbol{v}) \in [0, \pi/2)$, it is trivially true.

Inductively, assume $\theta(\boldsymbol{v}, \boldsymbol{w}^{(i)}) \in [0, \pi/2)$. Then, due to our previous assumption in this case that $\mathrm{Pr}\{h(\mathbf{x}, \boldsymbol{w}^{(i)}) \geq 0 \cap \mathrm{y} = 1\} > \frac{5}{2}(\epsilon\sqrt{\ln \epsilon^{-1}})^{1/2}$ for every $\boldsymbol{w}^{(i)} \in S$ and some sufficiently small $\epsilon$, we can refer proposition C.2 to obtain $\langle \mathbb{E}[-g_{\boldsymbol{w}^{(i)}}(\mathbf{x}, \mathrm{y})], \bar{\boldsymbol{v}}_{\boldsymbol{w}^{(i)}\perp} \rangle \geq \frac{2}{5}\epsilon\sqrt{\ln \epsilon^{-1}}$. Notice that, in algorithm 2, the update step in algorithm 2 tells us that

$$\boldsymbol{u}^{(i+1)} = \boldsymbol{w}^{(i)} + \beta \mathbb{E}_{(\mathbf{x},\mathrm{y})\sim\hat{\mathcal{D}}^{(i+1)}}[-g_{\boldsymbol{w}^{(i)}}(\mathbf{x}, \mathrm{y})].$$

Referring lemma F.8 for $\mathbb{E}_{\hat{\mathcal{D}}^{(i+1)}}[\langle g_{\boldsymbol{w}^{(i)}}(\mathbf{x}, \mathrm{y}), \bar{\boldsymbol{v}}_{\boldsymbol{w}^{(i)}\perp} \rangle]$ and some absolute constant $C > 0$ gives

$$\mathrm{Pr}_{\mathcal{D}}\left\{ \left| \left\langle \mathbb{E}_{\hat{\mathcal{D}}^{(i+1)}}[g_{\boldsymbol{w}^{(i)}}(\mathbf{x}, \mathrm{y})] - \mathbb{E}_{\mathcal{D}}[g_{\boldsymbol{w}^{(i)}}(\mathbf{x}, \mathrm{y})], \bar{\boldsymbol{v}}_{\boldsymbol{w}^{(i)}\perp} \right\rangle \right| \geq \frac{2}{5}\epsilon\sqrt{\ln \epsilon^{-1}} \right\} \leq 2e^{-CN\epsilon^2 \ln \epsilon^{-1}}$$

which implies $\left\langle \mathbb{E}_{\hat{\mathcal{D}}^{(i+1)}}[-g_{\boldsymbol{w}^{(i)}}(\mathbf{x}, \mathbf{y})], \bar{\boldsymbol{v}}_{\boldsymbol{w}^{(i)\perp}} \right\rangle \geq 0$ with probability at least $1 - 2e^{-CN\epsilon^2 \ln \epsilon^{-1}}$ for some sufficiently small $\epsilon$. Therefore, we have

$$\left\langle \mathbb{E}_{\hat{\mathcal{D}}^{(i+1)}}[-g_{\boldsymbol{w}^{(i)}}(\mathbf{x}, \mathbf{y})], \boldsymbol{v} \right\rangle = \left\langle \mathbb{E}_{\hat{\mathcal{D}}^{(i+1)}}[-g_{\boldsymbol{w}^{(i)}}(\mathbf{x}, \mathbf{y})], \boldsymbol{v}_{\boldsymbol{w}^{(i)\perp}} \right\rangle$$

$$= \|\boldsymbol{v}_{\boldsymbol{w}^{\perp}(i)}\|_2 \left\langle \mathbb{E}_{\hat{\mathcal{D}}^{(i+1)}}[-g_{\boldsymbol{w}^{(i)}}(\mathbf{x}, \mathbf{y})], \bar{\boldsymbol{v}}_{\boldsymbol{w}^{(i)\perp}} \right\rangle$$

$$\geq 0$$

Now, by lemma C.6, we can conclude that $\left\langle \boldsymbol{w}^{(i+1)}, \boldsymbol{v} \right\rangle \geq \left\langle \boldsymbol{w}^{(i)}, \boldsymbol{v} \right\rangle$, which implies $\theta(\boldsymbol{w}^{(i+1)}, \boldsymbol{v}) \in [0, \pi/2)$ with probability at least $1 - 2e^{-CN\epsilon^2 \ln \epsilon^{-1}}$. Taking a union bound over all $T \geq t$ iterations gives that $\theta(\boldsymbol{w}^{(t)}, \boldsymbol{v}) \in [0, \pi/2)$ with probability $1 - 2Te^{-CN\epsilon^2 \ln \epsilon^{-1}}$.

At last, combining $\theta(\boldsymbol{w}^{(t)}, \boldsymbol{v}) \in [0, \pi/2)$ and the assumption that $\|\mathbb{E}_{\mathcal{D}}[g_{\boldsymbol{w}^{(t)}}(\mathbf{x}, \mathbf{y})]\|_2 \leq \epsilon$, corollary C.4 gives $\Pr\{h(\mathbf{x}, \boldsymbol{w}) \geq 0 \cap \mathbf{y} = 1\} \leq \frac{5}{2}(\epsilon\sqrt{\ln \epsilon^{-1}})^{1/2}$. Taking $N \geq \ln(4T/\delta)/C\epsilon^2 \ln \epsilon^{-1}$ completes the proof. $\qquad\square$

We need the following lemma to aid the above argument.

**Lemma C.6** (Correlation Improvement Diakonikolas et al. (2020a)). *For unit vectors $\boldsymbol{v}, \boldsymbol{w} \in \mathbb{R}^d$, let $\boldsymbol{u} \in \mathbb{R}^d$ be such that $\langle \boldsymbol{u}, \boldsymbol{v} \rangle \geq c$, $\langle \boldsymbol{u}, \boldsymbol{w} \rangle = 0$, and $\|\boldsymbol{u}\|_2 \leq 1$, with $c > 0$. Then, for $\boldsymbol{w}' = \boldsymbol{w} + \lambda\boldsymbol{u}$, we have that $\langle \bar{\boldsymbol{w}}', \boldsymbol{v} \rangle \geq \langle \boldsymbol{w}, \boldsymbol{v} \rangle + \lambda c/8$.*

## D Analysis of Algorithm 3

We prove the generalization of our conditional learning algorithms from finite classes to sparse linear classes in this section.

**Theorem D.1** (Theorem 3.5). *Let $\mathcal{D}$ be a distribution on $\mathbb{R}^d \times \{0, 1\}$ with standard normal $\mathbf{x}$-marginal, and $\mathcal{C}$ be a class of sparse linear classifiers on $\mathbb{R}^d \times \{0, 1\}$ with sparsity $s = O(1)$. If there exist a unit vector $\boldsymbol{v} \in \mathbb{R}^d$ and a classifier $c \in \mathcal{C}$ such that, for some sufficiently small $\epsilon \in [0, 1/e]$, $\Pr_{(\mathbf{x}, \mathbf{y}) \sim \mathcal{D}}\{\mathbf{x} \in h(\boldsymbol{v}) \cap c(\mathbf{x}) \neq \mathbf{y}\} \leq \epsilon$, then, with at most $\text{poly}(d, 1/\epsilon, 1/\delta)$ examples, Algorithm 3 will return a $\boldsymbol{w}^{(c)}$, with probability at least $1 - \delta$, such that $\Pr_{(\mathbf{x}, \mathbf{y}) \sim \mathcal{D}}\{\mathbf{x} \in h(\boldsymbol{w}^{(c)}) \cap c(\mathbf{x}) \neq \mathbf{y}\} = \tilde{O}(\sqrt{\epsilon})$ and run in time $\text{poly}(d, 1/\epsilon, 1/\delta)$.*

*Proof.* We first show that the returned list of Algorithm 4 will contain a classifier $c'$ such that $\Pr_{(\mathbf{x}, \mathbf{y}) \sim \mathcal{D}}\{\mathbf{x} \in h(\boldsymbol{v}) \cap c'(\mathbf{x}) \neq \mathbf{y}\} \leq 2\epsilon$.

We decompose distribution $\mathcal{D}$ into a convex combination of an inlier distribution $\mathcal{D}^*$ and a outlier distribution $\tilde{\mathcal{D}}$ in the following way. Let $\mathcal{D}^*$ be a distribution on $\mathbb{R}^d \times \{0, 1\}$ with standard normal $\mathbf{x}$-marginal such that its labels are generated by $c(\mathbf{x})$, while $\tilde{\mathcal{D}}$ be any distribution on $\mathbb{R}^d \times \{0, 1\}$ with standard normal $\mathbf{x}$-marginals. Observe that, since $\Pr\{\mathbf{x} \in h(\boldsymbol{v}) \cap c(\mathbf{x}) \neq \mathbf{y}\} \leq \epsilon$ and $\Pr\{\mathbf{x} \in h(\boldsymbol{v})\} = 1/2$, there are at least $1/2(1 - \epsilon)$ fraction (weighted by Gaussian density) of the labels of $\mathcal{D}$ is consistent with $c(\mathbf{x})$. Therefore, there must exist some $\alpha \geq 1/2(1 - \epsilon)$ such that the labels of $\mathcal{D}_{\mathbf{x}}$ can be generated by selecting labels from $\mathcal{D}^*$ with probability mass $\alpha$ and from $\tilde{\mathcal{D}}$ with probability mass $1 - \alpha$, namely $\mathcal{D} = \alpha\mathcal{D}^* + (1 - \alpha)\tilde{\mathcal{D}}$.

Hence, we can refer Theorem A.1 and Definition 1.3 to conclude that there exists a classifier $c'$ in the returned list of Algorithm 4 such that $\Pr\{\mathbf{x} \in h(\boldsymbol{v}) \cap c'(\mathbf{x}) \neq \mathbf{y}\} \leq 2\epsilon$. Meanwhile, it is easy to see that Algorithm 4 takes only $\text{poly}(d, 1/\epsilon, 1/\delta)$ examples and runs in $\text{poly}(d, 1/\epsilon, 1/\delta)$ time since $\alpha$ is a constant.

At last, by Theorem C.1, we obtained the claimed result. $\qquad\square$

## E Analysis of Hardness Results

We denote $\mathbb{Z}_q := \{0, 1, \dots, q - 1\}$, $\mathbb{R}_q := [0, q)$, and $\text{mod}_q : \mathbb{R}^d \to \mathbb{R}_q^d$ to be the function that applies $\text{mod}_q$ operation on each coordinate of $\mathbf{x}$.

**Assumption E.1** (Sub-exponential LWE Assumption). *For $q, \kappa \in \mathbb{N}, \alpha \in (0,1)$ and $C > 0$ being a sufficiently large constant, the problem $LWE(2^{O(d^\alpha)}, \mathbb{Z}_q^d, \mathbb{Z}_q^d, \mathcal{N}(0, \sigma), \mathrm{mod}_q)$ with $q \leq d^\kappa$ and $\sigma = C\sqrt{d}$ cannot be solved in $2^{O(d^\alpha)}$ time with $2^{-O(d^\alpha)}$ advantage.*

For convenience, we restate the notations been used in section 4 at first.

For simplicity, we define $\mathrm{y} \equiv \mathbb{1}\{c(\mathbf{x}) \neq \mathrm{y}'\}$ for $(\mathbf{x}, \mathrm{y}') \sim \mathcal{D}'$ and only consider the distribution $(\mathbf{x}, \mathrm{y}) \sim \mathcal{D}$ for the rest of this section. Notice that, for agnostic setting, since $\mathcal{D}'$ is adversarial, $\mathcal{D}$ is also adversarial in the worst case. Therefore, such replacement does not affect the difficulty of the problems we concerned about.

Normally, one would consider the classification loss to be the expected disagreement between the classifier and the labelling. However, it is more convenient for us to view a labelling $\mathrm{y} = 1$ as an "occurrence of error" and define the loss in terms of such occurrences. Specifically, for any subset $S \subseteq \mathbb{R}^d$ and any distribution $\mathcal{D}$ on $\mathbb{R}^d \times \{0, 1\}$, we define the classification loss as

$$\mathrm{err}_{\mathcal{D}}(S) = \Pr_{(\mathbf{x}, \mathrm{y}) \sim \mathcal{D}} \{\mathrm{y} = \mathbb{1}\{\mathbf{x} \in S\}\}. \tag{8}$$

Note that this definition of classification loss is essentially the same as the "traditional" classification loss that defined in terms of disagreement since we can convert from one to another by simply negating the labelling.

Analogously, for any subsets $S, T \subseteq \mathbb{R}^d$ and any distribution $\mathcal{D}$ on $\mathbb{R}^d \times \{0, 1\}$, we denote the conditional classification loss as

$$\mathrm{err}_{\mathcal{D}|T}(S) = \Pr_{(\mathbf{x}, \mathrm{y}) \sim \mathcal{D}} \{\mathrm{y} = \mathbb{1}\{\mathbf{x} \in S\} \mid \mathbf{x} \in T\}. \tag{9}$$

For simplicity, we write $\mathrm{err}_{\mathcal{D}|T}$ instead of $\mathrm{err}_{\mathcal{D}|T}(S)$ when $S \equiv T$.

**Lemma E.2** (Lemma 4.4). *Let $\mathcal{D}$ be any distribution on $\mathbb{R}^d \times \{0, 1\}$ and $S$ be any subset of $\mathbb{R}^d$, we have $\mathrm{err}_{\mathcal{D}}(S) = 2\mathrm{err}_{\mathcal{D}|S} \Pr_{\mathcal{D}}\{\mathbf{x} \in S\} + \Pr_{\mathcal{D}}\{\mathrm{y} = 0\} - \Pr_{\mathcal{D}}\{\mathbf{x} \in S\}$ as well as $\mathrm{err}_{\mathcal{D}}(S) = 2\mathrm{err}_{\mathcal{D}|S^c}(S) \Pr_{\mathcal{D}}\{\mathbf{x} \in S^c\} + \Pr_{\mathcal{D}}\{\mathrm{y} = 1\} - \Pr_{\mathcal{D}}\{\mathbf{x} \in S^c\}$.*

*Proof.* By the law of total probability and definition (8), we have

$$\begin{aligned}
\mathrm{err}_{\mathcal{D}}(S) &= \Pr\{\mathrm{y} = \mathbb{1}\{\mathbf{x} \in S\}\} \\
&= \Pr\{\mathrm{y} = 1 \cap \mathbf{x} \in S\} + \Pr\{\mathrm{y} = 0 \cap \mathbf{x} \notin S\} \tag{10}
\end{aligned}$$

Again, by the law of total probability, we have that

$$\begin{aligned}
\Pr\{\mathrm{y} = 0 \cap \mathbf{x} \notin S\} &= \Pr\{\mathrm{y} = 0\} - \Pr\{\mathrm{y} = 0 \cap \mathbf{x} \in S\} \\
&= \Pr\{\mathrm{y} = 0\} - \Pr\{\mathbf{x} \in S\} + \Pr\{\mathrm{y} = 1 \cap \mathbf{x} \in S\} \tag{11}
\end{aligned}$$

Taking equation (11) back into (10) gives

$$\begin{aligned}
\mathrm{err}_{\mathcal{D}}(S) &= 2\Pr\{\mathrm{y} = 1 \mid \mathbf{x} \in S\} \Pr\{\mathbf{x} \in S\} + \Pr\{\mathrm{y} = 0\} - \Pr\{\mathbf{x} \in S\} \\
&= 2\mathrm{err}_{\mathcal{D}|S} \Pr\{\mathbf{x} \in S\} + \Pr\{\mathrm{y} = 0\} - \Pr\{\mathbf{x} \in S\}
\end{aligned}$$

where the last equation holds due to definition (9). Similar to equation (11), we have

$$\Pr\{\mathrm{y} = 1 \cap \mathbf{x} \in S\} = \Pr\{\mathrm{y} = 1\} - \Pr\{\mathbf{x} \notin S\} + \Pr\{\mathrm{y} = 0 \cap \mathbf{x} \notin S\}$$

which, when plugging back to equation 10, gives

$$\begin{aligned}
\mathrm{err}_{\mathcal{D}}(S) &= 2\Pr\{\mathrm{y} = 0 \mid \mathbf{x} \notin S\} \Pr\{\mathbf{x} \notin S\} + \Pr\{\mathrm{y} = 1\} - \Pr\{\mathbf{x} \notin S\} \\
&= 2\Pr\{\mathrm{y} = \mathbb{1}\{\mathbf{x} \in S\} \mid \mathbf{x} \in S^c\} \Pr\{\mathbf{x} \in S^c\} + \Pr\{\mathrm{y} = 1\} - \Pr\{\mathbf{x} \in S^c\} \\
&= 2\mathrm{err}_{\mathcal{D}|S^c}(S) \Pr\{\mathbf{x} \in S^c\} + \Pr\{\mathrm{y} = 1\} - \Pr\{\mathbf{x} \in S^c\}.
\end{aligned}$$

The proof is completed. $\square$

**Proposition E.3** (Proposition 4.5). *Let $\mathcal{D}$ be any distribution on $\mathbb{R}^d \times \{0, 1\}$, $\mathcal{H}$ be any subset of the power set of $\mathbb{R}^d$ closed under complement, and define $\mathcal{H}_{\mathcal{D}}^{a,b} = \{S \in \mathcal{H} \mid \Pr_{\mathcal{D}}\{\mathbf{x} \in S\} \in [a, b]\}$ for any $0 \leq a \leq b \leq 1$. For any $0 \leq a \leq b \leq 1$ and $\epsilon, \delta > 0$, given sample access to $\mathcal{D}$, if there exists an algorithm $\mathcal{A}_1(\epsilon, \delta, a, b)$ runs in time $\mathrm{poly}(d, 1/\epsilon, 1/\delta)$, and outputs a subset $S_1 \in \mathcal{H}_{\mathcal{D}}^{a,b}$ such that $\mathrm{err}_{\mathcal{D}|S_1} \leq \min_{S \in \mathcal{H}_{\mathcal{D}}^{a,b}} \mathrm{err}_{\mathcal{D}|S} + \epsilon$ with probability as least $1 - \delta$, there exists another algorithm $\mathcal{A}_2(\epsilon, \delta)$, runs in time $\mathrm{poly}(d, 1/\epsilon, 1/\delta)$, and outputs a subset $S_2 \in \mathcal{H}$ such that $\mathrm{err}_{\mathcal{D}}(S_2) \leq \min_{S \in \mathcal{H}} \mathrm{err}_{\mathcal{D}}(S) + 6\epsilon$ with probability at least $1 - \delta$.*

*Proof.* We prove the proposition by showing that there exists a efficient reduction from the problem of agnostic classification to conditional classification in terms of their loss functions.

Fix a subset $S^* \in \mathcal{H}$ such that $S^* = \operatorname{argmin}_{S \in \mathcal{H}} \operatorname{err}_{\mathcal{D}}(S)$ and define $p = \Pr\{\mathbf{x} \in S^*\}$. Then, let $p_l, p_u \geq 0$ be any constants such that $p_u - p_l = \epsilon$ as well as $p \in [p_l, p_u]$.

Consider now another subset $S' \in \mathcal{H}_{\mathcal{D}}^{p_l, p_u}$ such that $S' = \operatorname{argmin}_{S \in \mathcal{H}_{\mathcal{D}}^{p_l, p_u}} \operatorname{err}_{\mathcal{D}|S}$. Notice that $S^*$ is a feasible solution for the conditional classification problem on $\mathcal{H}_{\mathcal{D}}^{p_l, p_u}$, i.e. $S^* \in \mathcal{H}_{\mathcal{D}}^{p_l, p_u}$, because $\Pr\{\mathbf{x} \in S^*\} = p \in [p_l, p_u]$ by construction.

Let $S_1$ be the subset returned by algorithm $\mathcal{A}_1(\epsilon, \delta, p_l, p_u)$. Then, with probability at least $1 - \delta$, there is

$$
\begin{aligned}
\operatorname{err}_{\mathcal{D}}(S_1) =& 2\operatorname{err}_{\mathcal{D}|S_1} \Pr\{\mathbf{x} \in S_1\} + \Pr\{y = 0\} - \Pr\{\mathbf{x} \in S_1\} \\
&\overset{(i)}{\leq} 2\left(\operatorname{err}_{\mathcal{D}|S'} + \epsilon\right) \Pr\{\mathbf{x} \in S_1\} + \Pr\{y = 0\} - \Pr\{\mathbf{x} \in S_1\} \\
&\overset{(ii)}{\leq} 2\operatorname{err}_{\mathcal{D}|S^*} \Pr\{\mathbf{x} \in S_1\} + \Pr\{y = 0\} - \Pr\{\mathbf{x} \in S_1\} + 2\epsilon \\
&\overset{(iii)}{\leq} 2\operatorname{err}_{\mathcal{D}|S^*}(p + \epsilon) + \Pr\{y = 0\} - (p - \epsilon) + 2\epsilon \\
&\overset{(iv)}{\leq} 2\operatorname{err}_{\mathcal{D}|S^*} \Pr\{\mathbf{x} \in S^*\} + \Pr\{y = 0\} - \Pr\{\mathbf{x} \in S^*\} + 5\epsilon \\
=& \operatorname{err}_{\mathcal{D}}(S^*) + 5\epsilon
\end{aligned}
\tag{12}
$$

where the first equation is derived by lemma E.2, inequality (i) holds due to the error guarantee of algorithm $\mathcal{A}_1(\epsilon, \delta, p_l, p_u)$, inequality (ii) holds because of the optimality of $S'$ as well as $S^* \in \mathcal{H}_{\mathcal{D}}^{p_l, p_u}$, inequality (iii) holds since algorithm $\mathcal{A}_1(\epsilon, \delta, p_l, p_u)$ guarantees $S_1 \in \mathcal{H}_{\mathcal{D}}^{p_l, p_u}$, which implies $p_l \leq \Pr\{\mathbf{x} \in S_1\} \leq p_u$, and, by definition, there are $p_l \geq p - \epsilon$, $p_u \leq p + \epsilon$, inequality (iv) holds because $p = \Pr\{\mathbf{x} \in S^*\}$ by definition as well as $\operatorname{err}_{\mathcal{D}|S^*} = \Pr\{y = 1 \mid \mathbf{x} \in S^*\} \leq 1$, and the last equation is, again, by referring lemma E.2.

Although we do not know what value should $p$ take exactly, we only need to guess a small range where $p$ lies in to make inequality (12) holds with high probability. Specifically, we construct algorithm $\mathcal{A}_2(\epsilon, \delta)$ by using algorithm $\mathcal{A}_1$ as a subroutine in the following way.

For $k = 1, 2, \ldots, \lceil 1/\epsilon \rceil$, we run algorithm $\mathcal{A}_1(\epsilon, \epsilon\delta/2, (k-1)\epsilon, k\epsilon)$. Observe that, when we "guessed" the correct $k$ such that $p \in [(k-1)\epsilon, k\epsilon]$, inequality (12) must holds with probability at least $1 - \epsilon\delta/2$ because of the parameters we passed into $\mathcal{A}_1$. Let $S^{(k)}$ be the solution returned by algorithm $\mathcal{A}_1$ during the $k$th call, we construct an empirical distribution $\hat{\mathcal{D}} \overset{\text{i.i.d.}}{\sim} \mathcal{D}$ and choose $S_2$ such that $\operatorname{err}_{\hat{\mathcal{D}}}(S_2) \leq \min_{k \in \lceil 1/\epsilon \rceil} \operatorname{err}_{\hat{\mathcal{D}}}(S^{(k)})$. Notice that we only need the sample size of $\hat{\mathcal{D}}$ to be polynomially large to guarantee that $\operatorname{err}_{\mathcal{D}}(S_2) \leq \min_{k \in \lceil 1/\epsilon \rceil} \operatorname{err}_{\mathcal{D}}(S^{(k)}) + \epsilon$ with probability at least $1 - \delta/2$ by lemma F.4 (Chernoff Bound). Further, by a union bound over all $\lceil 1/\epsilon \rceil$ calls of algorithm $\mathcal{A}_1$ and the estimation of classification error on $\hat{\mathcal{D}}$, we have, with probability at least $1 - \delta$, that

$$
\begin{aligned}
\operatorname{err}_{\mathcal{D}}(S_2) \leq& \min_{k \in \lceil 1/2\epsilon \rceil} \operatorname{err}_{\mathcal{D}}(S^{(k)}) + \epsilon \\
&\overset{(i)}{\leq} \operatorname{err}_{\mathcal{D}}(S^*) + 6\epsilon \\
=& \min_{S \in \mathcal{H}} \operatorname{err}_{\mathcal{D}}(S) + 6\epsilon
\end{aligned}
$$

where inequality (i) alone holds with probability at least $1 - \delta/2$ because the second argument, $\epsilon\delta/2$, we passed in algorithm $\mathcal{A}_1$ guarantees that inequality (12) holds with probability at least $1 - \epsilon\delta/2$ when we guessed $p = \Pr\{\mathbf{x} \in S^*\}$ correctly, and taking a union bound over the $\lceil 1/\epsilon \rceil$ guesses gives probability at least $1 - \delta/2$. It is easy to see that each call, $\mathcal{A}_1(\epsilon, \epsilon\delta, (k-1)\epsilon, k\epsilon)$, runs in time poly$d, 1/\epsilon, 1/\epsilon\delta$, and we only called $\mathcal{A}_1$ for at most $\lceil 1/\epsilon \rceil$ times, the resulting running time is still poly$(d, 1/\epsilon, 1/\delta)$, which completes the proof. $\square$

**Claim E.4** (Claim 4.7). *Let $\mathcal{D}$ be any distribution on $\mathbb{R}^d \times \{0, 1\}$, $\mathcal{H}$ be any subset of the power set of $\mathbb{R}^d$ closed under complement, and define $\mathcal{H}_{\mathcal{D}}^{a,b} = \{S \in \mathcal{H} \mid \Pr_{\mathcal{D}}\{\mathbf{x} \in S\} \in [a, b]\}$ for any $0 \leq a \leq b \leq 1$. For any $0 \leq a \leq b \leq 1$, $\alpha, \epsilon, \delta > 0$, given sample access to $\mathcal{D}$, if there exists*

*an algorithm $\mathcal{A}_1(\alpha, \delta, a, b)$, runs in time $\mathrm{poly}(d, 1/\alpha, 1/\delta)$, and outputs a subset $S_1 \in \mathcal{H}_{\mathcal{D}}^{a,b}$ such that $\mathrm{err}_{\mathcal{D}|S_1} \leq (1 + \alpha)\min_{S \in \mathcal{H}_{\mathcal{D}}^{a,b}} \mathrm{err}_{\mathcal{D}|S}$ with probability as least $1 - \delta$, there exists another algorithm $\mathcal{A}_2(\alpha, \epsilon, \delta)$, runs in time $\mathrm{poly}(d, 1/\alpha, 1/\epsilon, 1/\delta)$, and outputs a subset $S_2 \in \mathcal{H}$ such that $\mathrm{err}_{\mathcal{D}}(S_2) \leq (1 + \alpha)(\min_{S \in \mathcal{H}} \mathrm{err}_{\mathcal{D}}(S) + 4\epsilon)$ with probability at least $1 - \delta$.*

*Proof.* Fix a subset $S^* \in \mathcal{H}$ such that $S^* = \mathrm{argmin}_{S \in \mathcal{H}} \mathrm{err}_{\mathcal{D}}(S)$ and define $p = \Pr\{\mathbf{x} \in S^*\}$. This proof generally follows the same strategy of the analysis of proposition E.3. However, differing from the proof of proposition E.3, to complete the multiplicative reduction, we have to deal with two cases, $2\mathrm{err}_{\mathcal{D}|S} \Pr_{\mathcal{D}}\{\mathbf{x} \in S\} \leq \mathrm{err}_{\mathcal{D}}(S)$ and $2\mathrm{err}_{\mathcal{D}|S^c}(S) \Pr_{\mathcal{D}}\{\mathbf{x} \in S^c\} \leq \mathrm{err}_{\mathcal{D}}(S)$, because $\mathrm{err}_{\mathcal{D}}(S)$ can be expressed in two forms according to lemma E.2.

Briefly speaking, when prove the additive reduction, the additive error will be carried through from conditional classification loss to classification loss no matter if $2\mathrm{err}_{\mathcal{D}|S} \Pr_{\mathcal{D}}\{\mathbf{x} \in S\} \leq \mathrm{err}_{\mathcal{D}}(S)$ because $\mathrm{err}_{\mathcal{D}}(S)$ is affinely related to $\mathrm{err}_{\mathcal{D}|S}$ by lemma E.2. However, whether a multiplicative error can be passed from one loss to another depends on whether $2\mathrm{err}_{\mathcal{D}|S} \Pr_{\mathcal{D}}\{\mathbf{x} \in S\} \leq \mathrm{err}_{\mathcal{D}}(S)$, which, of course, is not always true. Nonetheless, it is easy to see either $2\mathrm{err}_{\mathcal{D}|S} \Pr_{\mathcal{D}}\{\mathbf{x} \in S\} \leq \mathrm{err}_{\mathcal{D}}(S)$ or $2\mathrm{err}_{\mathcal{D}|S^c}(S) \Pr_{\mathcal{D}}\{\mathbf{x} \in S^c\} \leq \mathrm{err}_{\mathcal{D}}(S)$ based on lemma E.2: observe that $\Pr\{y = 0\} - \Pr\{\mathbf{x} \in S^*\} + \Pr\{y = 1\} - \Pr\{\mathbf{x} \notin S^*\} = 0$, so either $\Pr\{y = 0\} - \Pr\{\mathbf{x} \in S^*\}$ or $\Pr\{y = 1\} - \Pr\{\mathbf{x} \notin S^*\}$ must be nonnegative. We show that the multiplicative factor can be preserved through the reduction for both of these cases.

**Case I**, $\Pr\{y = 0\} - \Pr\{\mathbf{x} \in S^*\} \geq 0$. Let $p_l, p_u \geq 0$ be any constants such that $p_u - p_l = \epsilon$ as well as $p \in [p_l, p_u]$. Consider now another subset $S' \in \mathcal{H}_{\mathcal{D}}^{p_l, p_u}$ such that $S' = \mathrm{argmin}_{S \in \mathcal{H}_{\mathcal{D}}^{p_l, p_u}} \mathrm{err}_{\mathcal{D}|S}$. Notice that $S^*$ is a feasible solution for the conditional classification problem on $\mathcal{H}_{\mathcal{D}}^{p_l, p_u}$, i.e. $S^* \in \mathcal{H}_{\mathcal{D}}^{p_l, p_u}$, because $\Pr\{\mathbf{x} \in S^*\} = p \in [p_l, p_u]$ by construction.

Let $S_1$ be the subset returned by algorithm $\mathcal{A}_1(\epsilon, \delta, p_l, p_u)$. Then, with probability at least $1 - \delta$, there is

$$
\begin{aligned}
\mathrm{err}_{\mathcal{D}}(S_1) =& 2\mathrm{err}_{\mathcal{D}|S_1} \Pr\{\mathbf{x} \in S_1\} + \Pr\{y = 0\} - \Pr\{\mathbf{x} \in S_1\} \\
&\overset{(i)}{\leq} 2(1 + \alpha)\mathrm{err}_{\mathcal{D}|S'} \Pr\{\mathbf{x} \in S_1\} + \Pr\{y = 0\} - \Pr\{\mathbf{x} \in S_1\} \\
&\overset{(ii)}{\leq} 2(1 + \alpha)\mathrm{err}_{\mathcal{D}|S^*} \Pr\{\mathbf{x} \in S_1\} + \Pr\{y = 0\} - \Pr\{\mathbf{x} \in S_1\} \\
&\overset{(iii)}{\leq} 2(1 + \alpha)\mathrm{err}_{\mathcal{D}|S^*} (p + \epsilon) + \Pr\{y = 0\} - (p - \epsilon) \\
&\overset{(iv)}{\leq} 2(1 + \alpha)\mathrm{err}_{\mathcal{D}|S^*} \Pr\{\mathbf{x} \in S^*\} + (1 + \alpha)(\Pr\{y = 0\} - \Pr\{\mathbf{x} \in S^*\}) + 3(1 + \alpha)\epsilon \\
=& (1 + \alpha)(\mathrm{err}_{\mathcal{D}}(S^*) + 3\epsilon)
\end{aligned}
\tag{13}
$$

where the first equation is derived by lemma E.2, inequality (i) holds due to the error guarantee of algorithm $\mathcal{A}_1(\epsilon, \delta, p_l, p_u)$, inequality (ii) holds because of the optimality of $S'$ as well as $S^* \in \mathcal{H}_{\mathcal{D}}^{p_l, p_u}$, inequality (iii) holds since algorithm $\mathcal{A}_1(\epsilon, \delta, p_l, p_u)$ guarantees $S_1 \in \mathcal{H}_{\mathcal{D}}^{p_l, p_u}$, which implies $p_l \leq \Pr\{\mathbf{x} \in S_1\} \leq p_u$, and, by definition, there are $p_l \geq p - \epsilon$, $p_u \leq p + \epsilon$, inequality (iv) holds because $p = \Pr\{\mathbf{x} \in S^*\}$ by definition, $\mathrm{err}_{\mathcal{D}|S^*} = \Pr\{y = 1 \mid \mathbf{x} \in S^*\} \leq 1$, and $\Pr\{y = 0\} - \Pr\{\mathbf{x} \in S^*\} \geq 0$ by assumption, the last equation is, again, by referring lemma E.2.

**Case II**, $\Pr\{y = 1\} - \Pr\{\mathbf{x} \notin S^*\} \geq 0$. Let $p_l, p_u \geq 0$ be any constants such that $p_u - p_l = \epsilon$ as well as $1 - p \in [p_l, p_u]$. Further, let $\mathcal{D}_0$ be the distribution on $\mathbb{R}^d \times \{0, 1\}$ constructed by flipping the labels of $\mathcal{D}$. Notice that, for any $S \in \mathcal{H}$, we have, by definition 9, that

$$
\begin{aligned}
\mathrm{err}_{\mathcal{D}_0|S} =& \Pr_{(\mathbf{x}, y) \sim \mathcal{D}_0}\{y = \mathbb{1}\{\mathbf{x} \in S\} \mid \mathbf{x} \in S\} \\
=& \Pr_{(\mathbf{x}, y) \sim \mathcal{D}_0}\{y = 1 \mid \mathbf{x} \in S\} \\
\overset{(i)}{=}& \Pr_{(\mathbf{x}, y) \sim \mathcal{D}}\{y = 0 \mid \mathbf{x} \in S\} \\
=& \Pr_{(\mathbf{x}, y) \sim \mathcal{D}}\{y = \mathbb{1}\{\mathbf{x} \in S^c\} \mid \mathbf{x} \in S\} \\
=& \mathrm{err}_{\mathcal{D}|S}(S^c)
\end{aligned}
\tag{14}
$$

where equation (i) is because $\mathcal{D}_0$ has reversed labelling from $\mathcal{D}$ so that every $\mathrm{y} = 1$ in $\mathcal{D}_0$ is $\mathrm{y} = 0$ in $\mathcal{D}$, and the last equation is by definition (9).

Consider now another subset $S' \in \mathcal{H}_{\mathcal{D}_0}^{p_l,p_u}$ such that $S' = \mathrm{argmin}_{S \in \mathcal{H}_{\mathcal{D}_0}^{p_l,p_u}} \mathrm{err}_{\mathcal{D}_0|S}$. Notice that $S^{*c}$ is a feasible solution for the conditional classification problem on $\mathcal{H}_{\mathcal{D}}^{p_l,p_u}$, i.e. $S^* \in \mathcal{H}_{\mathcal{D}}^{p_l,p_u}$, because $\Pr\{\mathbf{x} \in S^{*c}\} = \Pr\{\mathbf{x} \notin S^*\} = 1 - p \in [p_l, p_u]$ by construction. Observe now that, since $\mathcal{D}_0$ only differ from $\mathcal{D}$ on the labelling, any subset $S \in \mathcal{H}_{\mathcal{D}}^{p_l,p_u}$ must also be in $\mathcal{H}_{\mathcal{D}_0}^{p_l,p_u}$, vice versa. Therefore, we also have $S' \in \mathcal{H}_{\mathcal{D}}^{p_l,p_u}$ as well as $S^{*c} \in \mathcal{H}_{\mathcal{D}_0}^{p_l,p_u}$

Let $S_1$ be the subset returned by algorithm $\mathcal{A}_1(\epsilon, \delta, p_l, p_u)$ given sample access to $\mathcal{D}_0$. Then, with probability at least $1 - \delta$, there is

$$
\begin{aligned}
\mathrm{err}_{\mathcal{D}}(S_1^c) =& 2\mathrm{err}_{\mathcal{D}|S_1}(S_1^c) \Pr\{\mathbf{x} \in S_1\} + \Pr\{\mathrm{y} = 1\} - \Pr\{\mathbf{x} \in S_1\} \\
&\overset{(i)}{=} 2\mathrm{err}_{\mathcal{D}_0|S_1} \Pr\{\mathbf{x} \in S_1\} + \Pr\{\mathrm{y} = 1\} - \Pr\{\mathbf{x} \in S_1\} \\
&\overset{(ii)}{\leq} 2(1+\alpha) \mathrm{err}_{\mathcal{D}_0|S'} \Pr\{\mathbf{x} \in S_1\} + \Pr\{\mathrm{y} = 1\} - \Pr\{\mathbf{x} \in S_1\} \\
&\overset{(iii)}{\leq} 2(1+\alpha) \mathrm{err}_{\mathcal{D}_0|S^{*c}} \Pr\{\mathbf{x} \in S_1\} + \Pr\{\mathrm{y} = 1\} - \Pr\{\mathbf{x} \in S_1\} \\
&\overset{(iv)}{=} 2(1+\alpha) \mathrm{err}_{\mathcal{D}|S^{*c}}(S^*) \Pr\{\mathbf{x} \in S_1\} + \Pr\{\mathrm{y} = 1\} - \Pr\{\mathbf{x} \in S_1\} \\
&\overset{(v)}{\leq} 2(1+\alpha) \mathrm{err}_{\mathcal{D}|S^{*c}}(S^*) (1 - p + \epsilon) + \Pr\{\mathrm{y} = 1\} - (1 - p - \epsilon) \\
&\overset{(vi)}{\leq} 2(1+\alpha) \mathrm{err}_{\mathcal{D}|S^{*c}}(S^*) \Pr\{\mathbf{x} \notin S^*\} + (1+\alpha)(\Pr\{\mathrm{y} = 1\} - \Pr\{\mathbf{x} \notin S^*\}) + 3(1+\alpha)\epsilon \\
&= (1+\alpha)(\mathrm{err}_{\mathcal{D}}(S^*) + 3\epsilon)
\end{aligned}
\tag{15}
$$

where the first equation is derived by lemma E.2, inequality (i) holds through using equation (14) reversely on $\mathrm{err}_{\mathcal{D}|S_1}(S_1^c)$, inequality (ii) holds due to the error guarantee of algorithm $\mathcal{A}_1(\epsilon, \delta, p_l, p_u)$, inequality (iii) holds because of the optimality of $S'$ as well as $S^{*c} \in \mathcal{H}_{\mathcal{D}_0}^{p_l,p_u}$ as we discussed previously, inequality (iv) holds by applying equation 14 on $\mathrm{err}_{\mathcal{D}_0|S^{*c}}$, inequality (v) holds since algorithm $\mathcal{A}_1(\epsilon, \delta, p_l, p_u)$ guarantees $S_1 \in \mathcal{H}_{\mathcal{D}_0}^{p_l,p_u}$, which implies $p_l \leq \Pr\{\mathbf{x} \in S_1\} \leq p_u$, and, by definition that $1 - p \in [p_l, p_u]$, there are $p_l \geq 1 - p - \epsilon$, $p_u \leq 1 - p + \epsilon$, inequality (vi) holds because $1 - p = \Pr\{\mathbf{x} \in S^{*c}\} = \Pr\{\mathbf{x} \notin S^*\}$ by definition, $\mathrm{err}_{\mathcal{D}|S^{*c}}(S^*) = \Pr\{\mathrm{y} = 0 \mid \mathbf{x} \notin S^*\} \leq 1$, and $\Pr\{\mathrm{y} = 1\} - \Pr\{\mathbf{x} \notin S^*\} \geq 0$ by assumption, the last equation is, again, by referring lemma E.2.

Given inequalities (13) and (15), we can conclude that, when $\Pr\{\mathbf{x} \in S^*\}$ is known, we can always use $\mathcal{A}_1$ to find a subset $S$ such that, with probability at least $1 - \delta$, $\mathrm{err}_{\mathcal{D}}(S) \leq (1+\alpha)(\mathrm{err}_{\mathcal{D}}(S^*) + 3\epsilon)$.

Then, the construction and analysis of $\mathcal{A}_2$ is rather identical to those of $\mathcal{A}_2$ in the proof of proposition E.3. We will then refer the proof of proposition E.3 to completes the analysis. $\square$

## F  GAUSSIAN PROPERTIES AND CONCENTRATION TOOLS

In this section, we show some common properties of Gaussian distributions for completeness.

**Definition F.1** (Sub-gaussian norm Vershynin (2018)). *For any random variable* $\mathrm{x} \sim \mathcal{D}$ *on* $\mathbb{R}$*, we define* $\|\mathrm{x}\|_{\psi_2} = \inf\left\{t > 0 \mid \mathbb{E}_{\mathrm{x} \sim \mathcal{D}}[e^{\mathrm{x}^2/t^2}] \leq 2\right\}$.

**Fact F.2** (Gaussian $\psi_2$-norm). *Let* $\mathrm{z} \sim \mathcal{N}(0, \sigma^2)$*, we have* $\|\mathrm{z}\|_{\psi_2} = \sqrt{8/3}\sigma$.

**Fact F.3** (Gaussian Tail Bound). *Let* $\mathrm{z} \sim \mathcal{N}(0, \sigma^2)$*, we have* $\Pr\{\mathrm{z} \geq t\} \leq e^{-t^2/2\sigma^2}$.

**Lemma F.4** (Chernoff Bound). *Let* $\mathrm{x}_1, \ldots, \mathrm{x}_m$ *be a sequence of* $m$ *independent Bernoulli trials, each with probability of success* $\mathbb{E}[\mathrm{x}_i] = p$*, then with* $\gamma \in [0, 1]$*, we have*

$$
\Pr\{\left|\frac{1}{m}\sum_{i=1}^{m} \mathrm{x}_i - p\right| > \gamma\} \leq 2e^{-2m\gamma^2}.
$$

**Lemma F.5** (General Hoeffding Bound (Vershynin, 2018)). *Let* $x_1, \ldots, x_m$ *be a sequence of* $m$ *independent mean-zero sub-gaussian random variables. Then, for all* $t \geq 0$ *and some absolute constant* $c > 0$, *we have*

$$\Pr\left\{\left|\frac{1}{m}\sum_{i=1}^{m} x_i\right| \geq t\right\} \leq 2\exp\left(-\frac{cm^2t^2}{\sum_{i=1}^{m}\|x_i\|_{\psi_2}^2}\right).$$

**Lemma F.6.** *Let* $\mathcal{D}$ *be any distribution on* $\mathbb{R}^d \times \{0, 1\}$ *with* $\mathbf{x}$-*marginal such that* $\|\langle\mathbf{x}, \boldsymbol{u}\rangle\|_{\psi_2} \leq K$ *for some unit vector* $\boldsymbol{u} \in \mathbb{R}^d$. *For any event* $T \subseteq \mathbb{R}^d$, *we have* $\|y \cdot \langle\mathbf{x}, \boldsymbol{u}\rangle\mathbb{1}\{\mathbf{x} \in T\}\|_{\psi_2} \leq K$.

*Proof.* Because $y$ and $\mathbb{1}\{\mathbf{x} \in T\}$ are boolean valued, we have

$$\mathbb{E}[\exp\left((y \cdot \langle\mathbf{x}, \boldsymbol{u}\rangle\mathbb{1}\{\mathbf{x} \in T\})^2/K^2\right)] \leq \mathbb{E}[\exp(\langle\mathbf{x}, \boldsymbol{u}\rangle^2/K^2)]$$

$$\overset{(i)}{\leq} \mathbb{E}[\exp(\langle\mathbf{x}, \boldsymbol{u}\rangle^2/\|\langle\mathbf{x}, \boldsymbol{u}\rangle\|_{\psi_2}^2)]$$

$$\leq 2$$

where inequality (i) holds because $\mathbb{E}[\exp(\langle\mathbf{x}, \boldsymbol{u}\rangle^2/t^2)]$ is a decreasing function of $t^2$, and the last inequality is by Definition F.1. By the same definition, the above inequality implies the claimed result. $\qquad\square$

**Lemma F.7** (Lemma 2.6.8 in Vershynin (2018)). *If* $x \sim \mathcal{D}$ *is a sub-gaussian random variable on* $\mathbb{R}$ *such that* $\|x\|_{\psi_2} \leq K$, *then there exists some absolute constant* $C$ *such that* $\|x - \mathbb{E}_{\mathcal{D}}[x]\|_{\psi_2} \leq CK$.

**Corollary F.8.** *Let* $\mathcal{D}$ *be any distribution on* $\mathbb{R}^d \times \{0, 1\}$ *with standard normal* $\mathbf{x}$-*marginal and* $\hat{\mathcal{D}} \overset{i.i.d.}{\sim} \mathcal{D}$ *be an* $m$-*sample. Define* $g_{\boldsymbol{w}}(\mathbf{x}, y) = y \cdot \mathbf{x}_{\boldsymbol{w}^\perp}\mathbb{1}\{\mathbf{x} \in h(\boldsymbol{w})\}$. *Fix some* $\boldsymbol{v} \in \mathbb{R}^d$, *then, for any* $\boldsymbol{w} \in \mathbb{R}^d$, *it holds that*

$$\Pr\left\{\left|\left\langle\underset{\hat{\mathcal{D}}}{\mathbb{E}}[g_{\boldsymbol{w}}(\mathbf{x}, y)] - \underset{\mathcal{D}}{\mathbb{E}}[g_{\boldsymbol{w}}(\mathbf{x}, y)], \bar{\boldsymbol{v}}\right\rangle\right| > t\right\} \leq e^{-Cmt^2}$$

*where* $C > 0$ *is an absolute constant.*

*Proof.* Let's first notice that, since $\mathbf{x}_{\boldsymbol{w}^\perp}$ is a projection of $\mathbf{x}$ to a space of lower dimension, the variance of $\langle\bar{\boldsymbol{v}}, \mathbf{x}_{\boldsymbol{w}^\perp}\rangle$ must be no larger than that of $\langle\bar{\boldsymbol{v}}, \mathbf{x}\rangle$ and, hence, $\|\langle\bar{\boldsymbol{v}}, \mathbf{x}_{\boldsymbol{w}^\perp}\rangle\|_{\psi_2} \leq \sqrt{8/3}$ by Fact F.2. Then, combining Lemma F.6 and Lemma F.7 results to $\|\langle g_{\boldsymbol{w}}(\mathbf{x}, y), \bar{\boldsymbol{v}}\rangle\|_{\psi_2} \leq C'\sqrt{8/3}$ for some $C' > 0$. At last, applying Lemma F.5 on $\langle g_{\boldsymbol{w}}(\mathbf{x}, y), \bar{\boldsymbol{v}}\rangle$ gives the claimed tail bound. $\qquad\square$

**Lemma F.9.** *Let* $x \sim \mathcal{N}(0, 1)$, *then* $\Pr\{x \geq t\} \geq \frac{1}{\sqrt{2\pi}(t+1)}e^{-t^2/2}$ *for every* $t \geq 0$.

*Proof.* Define $f : \mathbb{R} \to \mathbb{R}$ as

$$f(t) = \sqrt{2\pi}\Pr\{x \geq t\} - \frac{1}{t+1}e^{-t^2/2}$$

$$= \int_t^{+\infty}e^{-x^2/2}dx - \frac{1}{t+1}e^{-t^2/2}.$$

Observe that $f(0) = \sqrt{\pi/2} - 1 > 0$ and

$$\nabla_t f(t) = -e^{-t^2/2} - \left(-\frac{1}{(t+1)^2}e^{-t^2/2} - \frac{t}{t+1}e^{-t^2/2}\right) \tag{16}$$

$$= -\frac{t}{(t+1)^2}e^{-t^2/2} \tag{17}$$

$$\leq 0 \tag{18}$$

for $t \geq 0$. Furthermore, we have $\lim_{t \to +\infty} f(t) = 0$, which implies $f(t)$ is always positive on $t \in [0, +\infty)$ and, hence, the claimed result. $\qquad\square$