# OpenReview forum: "Distribution-Specific Agnostic Conditional Classification With Halfspaces"
_ICLR.cc/2025/Conference — ICLR 2025 Poster_

### Official Review · Reviewer_qivV · 2024-10-29

**Soundness:** 3
**Presentation:** 3
**Contribution:** 2
**Rating:** 5
**Confidence:** 4

**Summary:**

This paper studies the agnostic conditional classification problem for a finite class of hypotheses $\mathcal{C}$. In the classic agnostic learning setting, a learner is asked to output a hypothesis $c \in \mathcal{C}$ that achieves a small error over the whole domain of examples. In the conditional classification setting, a learner is instead asked to output a subset $S$ of the domain and a hypothesis $c$ such that the conditional error over $S$ is small. The motivation is that over some subset of the domain, the error rate of a hypothesis could be much lower than the error rate over the whole domain. This paper focuses on a special case of the conditional classification problem, where the marginal distribution is a standard Gaussian and the subset output by the learner is the positive region of some homogeneous halfspace. The paper gives an algorithm that runs in time $poly(|\mathcal{C}|,1/\epsilon,d)$ and outputs a hypothesis $c \in \mathcal{C}$ and a homogeneous halfspace $h$ such that the conditional error of $c$ over $h$ is at most $\tilde{O}(\sqrt{opt}+\epsilon)$, where $opt$ is the smallest conditional error of some $c$ over all possible homogeneous halfspaces. Technically, the algorithm can be seen as an application of agnostic learning homogeneous halfspaces under Gaussian with respect to conditional error (examples in the halfspaces are labeled by 0, noisy examples have artificial labels 1). Given such a halfspace learning algorithm, the learner can brute force and compute an approximate best halfspace for each hypothesis in the class. On the other hand, the paper reduces the problem of agnostic learning Gaussian halfspaces to the conditional classification problem and uses the known cryptographic hardness result to show that it is impossible to approximate the optimal conditional classification error within an additive error in polynomial time.

**Strengths:**

1. The presentation of the paper is clear. The main body of the paper gives a clear technical review of the algorithms and the proof of the hardness result.
2. Technically, this paper gives a novel algorithm for agnostic learning homogeneous Gaussian with conditional error. Previously known algorithms for agnostic learning halfspaces all concern the error over the whole domain. It is not clear whether we can apply these algorithms directly to get a good hypothesis with a small conditional error. So, the new algorithm might be of interest to the community

**Weaknesses:**

1. From my point of view, the main weakness of this paper is the motivation of using homogeneous halfspaces as a selector under Gaussian distribution. I am not sure whether the conditional error itself alone is a meaningful metric, because, in selective classification, one should also consider the coverage of the output selector (accept rate of the selector). For example, one may output two selectors with different accept rates but the same conditional error. In this case, it makes no sense to use the one with a lower accept rate. However, such an important characterization is not considered by the paper. Under Gaussian distribution, all homogeneous halfspace selectors have a reject rate of 1/2, which is fairly high. It does not make sense to use a selector that has to throw away half of the examples. More importantly, this equal reject rate makes the risk-coverage tradeoff curve disappear in the setting studied by this paper. These two limitations of using a halfspace selector under Gaussian make the result a little bit strange from the perspective of selective classification and look more like an application of halfspace learning.

2. The main algorithm can only deal with a class of finite concepts. In fact, finiteness is very crucial in the paper as the main algorithm in the paper computes a good selector for every single concept in the class and does not make use of any structure of the concept classes. Though, at the end of Section 1.1, the authors mention that one can deal with larger classes via robust list learning, the paper does not give a very clear explanation of how to use robust list learning to selectively learn a general class.  For example, to selectively learn a VC class $H$, it is not clear how to set up the parameters in the corresponding robust list learning. The authors also mentioned an efficient list-learning algorithm for sparse linear separators but did not explicitly connect this example with selective learning. I believe more explanations are needed in the future version of the paper.

3. As mentioned by the authors, the error guarantee $\tilde{O}(\sqrt{opt})+\epsilon$ is not optimal, and thus the hardness result in the paper does not rule out the possibility of having an algorithm with much better error guarantee.

4. The abstract in the OpenReview states the algorithm can learn a selector up to error $O(opt)$, which is different from the results obtained by the paper.

**Questions:**

1. Can you help formally explain how to use robust list learning to deal with larger hypothesis classes? Why learning $O(1)$-sparse linear separators is used as an example here? Is it because the length of the list can be controlled? What if we consider other simple hypothesis classes such as high-dimensional boxes or dense linear separators?

2. Previous agnostic halfspace learning papers such as [DKTZ22] show that under Gaussian as long as the angle between the current hypothesis and the target is small, one can achieve error $O(opt+\epsilon)$ and thus those algorithms all converge in the last iterate. Can you provide any intuition of why similar arguments cannot be used here and why the algorithm can only converge in average and achieve $\sqrt{opt}+\epsilon$?

3. Please make comments on the weakness that I pointed out above. I would raise my score based on the response.


[DKTZ22]Diakonikolas I, Kontonis V, Tzamos C, Zarifis N. Learning general halfspaces with adversarial label noise via online gradient descent. International Conference on Machine Learning 2022 Jun 28 (pp. 5118-5141). PMLR.

---

> ### Author Response · Authors · 2024-11-18
>
> Thanks for the suggestions! We will respond to the weakness you pointed out and your questions respectively.
>
> For the first weakness, there might be some misunderstanding about our motivation for considering conditional error. First of all, you are right that it would be ideal to be able to maximize the coverage for any given risk level or, equivalently, minimize the risk for any given coverage. Recall that in the problem of conditional classification, we seek to minimize the classification error conditioned on any selector with some fixed coverage. Under Gaussian distributions, if we are able solve the conditional classification problem with the class of general halfspace selectors, we can always achieve the optimal risk-coverage trade-off (by simply searching through all reasonable thresholds). However, this ideal situation seems hard to reach as partly justified by our negative results, and our results on approximating conditional classification error for homogeneous halfspace selectors represent the current state of the art. Moreover, in the task of conditional classification, we are also very interested in cases where the coverage is even significantly smaller than $1/2$, since we expect to achieve significantly smaller error at lower coverage. The risk-coverage model commonly studied in selective learning suggests lower coverage rate usually allows smaller optimal error. And indeed, for example, the work by Hainline et al. on sparse conditional linear regression (AISTATS 2019) found that this occurs in many of the standard small benchmark data sets for regression, at least for conditions specified by small 2-DNF formulas. While the task Hainline et al. studied differs from our task in two important respects (we are interested in classification on halfspace subsets), we expect something similar to occur.
>
> The the second weakness can be addressed by our answer to a strongly related question that almost every reviewer has asked about, we refer you to the answer we posted separately for details, where we discuss how our approach allows us to learn sparse linear representations in polynomial time. It is an interesting open question how far our results can be extended.
>
> For the third weakness, it is true that there does exist a relatively large gap between the lower and upper bounds we currently can achieve. However, even if it is possible to obtain a better error guarantee, it may come at the cost of a higher-order polynomial running time.
>
> For the last weakness, we are sorry for the typo in the abstract, it should indeed be $\tilde{O}(\sqrt{\mathrm{opt}})$.
>
> For your first question, again, please see our common answer.
>
> For your second question, we will answer it in parts. The online gradient descent algorithm in [DKTZ22] is guaranteed to converge in the last iteration because it is non-stochastic that their gradient descent step are always running on the same data set (non-stochastic). On the other hand, our Algorithm 2 is a stochastic gradient descent process, whose convergence is usually presented in average in standard analysis. In fact, we believe it is possible to adopt the "Angle Contraction" argument in [DKTZ22] to our algorithms for homogeneous halfspace selectors, but it is not likely to improve the performance guarantee asymptotically. This brought us to the second part of your question. Briefly speaking, our error guarantee is eventually determined by the lower bound on the projection of the gradient $\mathbb{E}[g_{\mathbf{w}}]$ onto the optimal normal vector $\mathbf{v}$ (Proposition 3.2). The proof strategy of Proposition 3.2 actually resembles to that of Lemma 3.5 in [DKTZ22], where they decomposed the projection into a noiseless part and a noisy part. However, as far as we are concerning conditional classification error, it seems impossible to isolate a term that is completely noiseless while having a significant contribution to the projection simultaneously. As a result, our decomposition gives two
> "noisy" terms, one of which corresponds to the optimal error (contributes negatively) and the other one corresponds to the sub-optimal error (contributes positively). With our current argument, we need a better lower bound on the term corresponding to the sub-optimal error to improve over the $\sqrt{\mathrm{opt}} + \epsilon$ error bound.

---

> > ### Comment · Reviewer_qivV · 2024-11-25
> >
> > I thank the authors for their response and will keep my rate.

---

### Official Review · Reviewer_XCyY · 2024-10-31

**Soundness:** 3
**Presentation:** 3
**Contribution:** 3
**Rating:** 8
**Confidence:** 3

**Summary:**

The paper considers agnostic conditional classification when the underlying marginal distribution is Gaussian and the selector is a half-space. First, the authors prove a polynomial time algorithm that returns a classifier-selector pair such that the associated risk is at most the square root of optimal risk. Given this, a natural question is whether one can achieve an error that is at most $\varepsilon$ away (additively) from the optimal risk. Unfortunately, the authors prove a hardness result showing that no polynomial algorithm can achieve this.

**Strengths:**

1. The paper is well-written and is easy to follow. I particularly like Section 3.2 where the paper guides us through the proof sketch, while including propositions, and lemmas required for the proof.

2. The algorithm appears both natural and intuitive. While not strictly necessary, it would be helpful if the authors could provide additional context around two aspects of the proposed algorithm. First, what is the intuition behind initializing PSGD for both $ w^{(0)} $ and $-w^{(0)} $? Second, what motivates the choice of ReLU as the convex surrogate? Although the paper includes some description, it is unclear whether ReLU's usefulness here is due to the separators being halfspaces, or if the goal is more related to obtaining gradients of a specific form by using tools from Diakonikolas et al. (2020b).

**Weaknesses:**

The authors list a few limitations of the work in the final section of the paper. Another major weakness of the paper is the fact that Theorem 3.1 only holds for a finite class of classifiers $\mathcal{C}$. Typically, finite hypothesis class $\mathcal{C}$ is justified because some form of discretization-based arguments generalize the result to infinite class as well. For example, usually, proving risk bounds for half-spaces can be reduced to proving risk bounds for finite discretization of half-spaces. However, Theorem 3.1 cannot even handle the discretization of half-spaces because the size of $\varepsilon$ discretization of half-spaces on $d$ dimension is roughly $\varepsilon^{-d}$.  Given that the runtime of Algorithm 1 is $O(d\,  |\mathcal{C}|/\varepsilon^6)$, taking $|\mathcal{C}| \approx \varepsilon^{-d}$ makes the runtime exponential in $d$.

**Questions:**

1. How essential is the choice of the Gaussian distribution for this analysis? I skimmed the proofs in the supplementary materials, and it appears that the arguments primarily rely on the symmetry and tail probability bounds of the Gaussian. Do you think the same results could be extended to symmetric sub-Gaussian (or, ideally, just sub-Gaussian) random variables?

2. Is anything known about lower bounds on the risk of polynomial algorithms in this setting? In their discussion, the authors mention that the $O(\sqrt{\varepsilon}) $ rate appears suboptimal. I was wondering if the authors have any reasons to support this belief.

---

> ### Author Response · Authors · 2024-11-18
>
> Thanks for the suggestions! We will respond to the weakness you pointed out and your questions respectively.
>
> To answer the first question in the Strengths section, notice that the lower bound in the claim of Proposition 3.2 holds only if the angle between $\mathbf{w}$ and $\mathbf{v}$ is less than $\pi/2$. Since Proposition 3.2 will be used to show at least one of the parameters returned by Algorithm 2 is approximately optimal inductively in Lemma 3.5, we need to guarantee that the angle between $\mathbf{w}^{(0)}$ and $\mathbf{v}$ is also less than $\pi/2$ in each run of Algorithm 2. Obviously, exactly one of $\mathbf{w}^{(0)}$ and $-\mathbf{w}^{(0)}$ satisfies the requirement.
>
> To answer the second question in the Strengths section, notice that the most important part of our argument (Proposition 3.2 specifically) is to lower bound the projection of the gradient $\mathbb{E}[g_{\mathbf{w}}]$ onto the optimal normal vector. Observe that our assumptions only give us information on $\Pr\\{y=1\ |\ \mathbf{x}\in h(\mathbf{v})\\}$ and $\Pr\\{y=1\ |\ \mathbf{x}\in h(\mathbf{w})\\}$, therefore, we do want to make sure $\mathbb{E}[g_{\mathbf{w}}]$ is always zero on $\mathbf{x}\notin h(\mathbf{w})$, which, in turn, requires $L_{\mathcal{D}}(\mathbf{w})$ to be zero on $\mathbf{x}\in h(\mathbf{w})$. Therefore, ReLU seems to be a natural choice as our surrogate loss in this context.
>
> The weakness you have pointed out can be strengthened by our answer to a strongly related question that almost every reviewer has asked about, we refer you to the answer we posted separately for details, where we explained how our approach can be used to learn sparse linear representations in polynomial time.
>
> For your first question, we recently found that it is possible to generalize our results to broader classes of distributions such as log-concave ones, however, with compromising the performance guarantee to $O(\mathrm{opt}^{1/4})$. To the best of our knowledge, (even symmetric) sub-gaussian tail alone may not be enough for our results to hold as we also need properties such as anti-concentration on all directions.
>
> For the second question, it seems clear that  $\sqrt{\epsilon}$ is statistically sub-optimal, since we can view our classifier as producing three labels (true, false, and abstain), computed by two halfspaces, and we can approach $\epsilon$ simply by minimizing the empirical error on a sufficiently large sample. So the only question is whether there is a computational barrier. Here, our reasons to suspect that $\sqrt{\epsilon}$ is sub-optimal are simply as follows. First, it is still consistent with the known negative results, even for general halfspaces. Second, it was possible to reach $O(\epsilon)$ error for agnostic learning of halfspaces w.r.t. Gaussian data. Third, we also know that in the case of other, simpler list-decodable robust estimation tasks (e.g., mean estimation in the referenced work by Kothari-Steinhardt-Steurer, STOC 2018) it was possible to use $k$th moments to obtain $\epsilon^{1-O(1/k)}$ error rather than $\sqrt{\epsilon}$ as in the original works, and  for list-decodable linear regression it is similarly possible to approach optimal error by spending more on the running time (see for example Bakshi-Kothari SODA 2021) so there does not seem to be a hard barrier at some suboptimal error rate. It is natural to guess that something similar occurs here as well.

---

> > ### Comment · Reviewer_XCyY · 2024-11-22
> >
> > I thank the authors for their response.

---

### Official Review · Reviewer_bBb3 · 2024-11-01

**Soundness:** 3
**Presentation:** 3
**Contribution:** 3
**Rating:** 6
**Confidence:** 3

**Summary:**

This work studies the problem of conditional classification. The problem of conditional classification is the following: Given a finite classifiers of class $\mathcal{C}$ and a set of concepts $\mathcal{H}$ that have non-trivial probability over the marginal, the task is to find a pair $(c,h)$ for $c\in \mathcal{C}$ and $h\in \mathcal{H}$ such that $\Pr[y\neq c(x)\mid x\in h]$ is small. They consider $\mathcal{H}$ to be the class of homogeneous halfspaces. They prove that there exists a polynomial time algorithm for the above task that achieves error $\tilde{O}(\sqrt{opt})$ where $opt$ is the minimum error possible. They also prove a lower bound that says conditional classification is hard for general halfspaces.

**Strengths:**

1) The reduction from conditional classification to one sided agnostic classification seems novel. Although the reduction is not complicated, it is neat.
2) The proofs seem correct and intuitive.
3) I believe that this work makes important progress in the study of conditional classification.
4) Their algorithm runs in polynomial time (although they obtain a multiplicative approximation).

**Weaknesses:**

1) The lower bound for general halfspaces is for the objective $opt+\epsilon$. However, the upper bound only achieves $\sqrt{opt}$ for homogenous halfspaces. Is it the case that $\sqrt{opt}$ is not possible for general halfspaces efficiently? Their lower bound does not seem to exactly complement their upper bound.
2) The statements of the theorems have grammatical errors. For instant, in proposition 3.2, the last line says if {property A}, there is {property B}. "There is" should be replaced by "it holds that" or "we have".
3) The presentation is a bit unclear in some places. Further expanded in Questions.

**Questions:**

1) In section 1.3, it would be helpful to state both theorems corresponding to the contributions within this section or provide a link to their statements for easy reference.
2) The connection to list learning is unclear. In particular, why is Theorem 2.1 stated in the preliminaries? Where is it used in your proof?

---

> ### Author Response · Authors · 2024-11-18
>
> Thanks for the suggestions! We will respond to the weakness you pointed out and your questions respectively.
>
> For the first weakness you pointed out, we admit that there does exist a relatively large margin between the lower bound and the upper bound we currently can achieve. To the best of our knowledge, it is not very clear on whether we can achieve the same $\tilde{O}(\sqrt{\mathrm{opt}})$ error on general halfspaces under standard normal marginals using a similar approach. In particular, the optimality analysis of our algorithms is mostly based on the observation that, when a halfspace, $h(\mathbf{w})$, is sub-optimal (the disagreement region w.r.t. the optimal halfspace, $h(\mathbf{v})$, is non-empty), the disagreement region will contribute positively to the projection of the gradient $\mathbb{E}[g_{\mathbf{w}}]$ onto the optimal normal vector $\mathbf{v}$ in homogeneous cases (see the claim of Proposition 3.2). However, in the general cases, the contribution from the disagreement region will no longer necessarily be positive, especially when the halfspace is far away from the origin. Therefore, we strongly believe a new technique is needed to handle the case of general halfspace selectors.
>
> For the rest of the weaknesses and the first question, we thank you for the advices and will revise our paper accordingly.
>
> The second question is a common question that almost every reviewer has asked about, we refer you to the answer we posted separately for details, where we explained how our approach allows us to learn sparse linear representations in polynomial time.

---

> > ### Comment · Reviewer_bBb3 · 2024-11-22
> >
> > I thank the authors for their response. I am satisfied by their response and keep my score.

---

### Official Review · Reviewer_ZDsQ · 2024-11-01

**Soundness:** 3
**Presentation:** 3
**Contribution:** 3
**Rating:** 6
**Confidence:** 2

**Summary:**

This paper studies distribution-specific agnostic conditional classification for halfspaces. In particular, they consider sparse linear classifiers, subsets defined by halfspaces, and Gaussian feature distributions. Both positive and negative results are provided. On the positive side, they authors give the first polynomial time PAC learning algorithm when the selectors are homogenous half spaces with error guarantee $\tilde{O}(\sqrt{\epsilon})$. On the negative side, the authors show that under standard cryptographic assumptions and even under gaussian features, estimating the conditional classification error is computationally hard. This proof involved reducing approximating agnostic classification, which is known to be hard, to approximating conditional classification.

**Strengths:**

- The problem setting is interesting and well-motivated
- The proof techniques seem non-trivial and novel
- Both the positive and negative results, in my opinion, and important contributions to this area.

**Weaknesses:**

I don't have any specific weakness in mind. That said, the paper does feel very dense and would benefit from more informal theorems/lemmas/propositions in the main text, and more details in the appendix.

**Questions:**

- Theorem 1 states that Algorithm 1 outputs a good half space selector **for each** concept $c \in C$. But, don't you need to ouput a single selector that works well across all concepts in $C$? If I am not mistaken, I do not see a Theorem in the paper that quantifies the error of the selector output in Step 11 of Algorithm 1.

- Based on Algorithm 1, it seems like the class of concepts needs to be finite? Is there anyway to relax this assumption and have the input $T, N$ depend on the VC dimension of $C$?

---

> ### Author Response · Authors · 2024-11-18
>
> Thanks for your suggestions! We will try to adjust our content arrangement according to your suggestions.
>
> For your first question, we are sorry for the confusion. Actually, Theorem 3.1 gives the performance guarantee for line 11 of Algorithm 1, that is, it provides the error bound for the final output pair $(c', \mathbf{w}^{c'})$ over all $c\in\mathcal{C}$ (line 11). Lemma 3.5 provides the error guarantee of Algorithm 2, which outputs the halfspace selectors, for each $c\in\mathcal{C}$. More specifically, Algorithm 1 simply enumerate all $c\in\mathcal{C}$ (so, yes, $\mathcal{C}$ in Algorithm 1 is finite, and we will elaborate how a robust list-learning algorithm can help generalize our approach to work with more general hypothesis classes when answering your second question), where we call Algorithm 2 to find a sequence of halfspace selectors for each $c\in\mathcal{C}$ such that at least one of these selectors will approximately minimize the conditional error of this $c$. The statistical error of the hidden good selector in the sequence for each $c\in\mathcal{C}$ is bounded in Lemma 3.5. Then, using standard error estimation, we can select a halfspace $\mathbf{w}^{(c)}$ from the output sequence by Algorithm 2 that is at least as competitive as the hidden good selector for each $c\in\mathcal{C}$. Eventually, at line 11 of Algorithm 1, we simply select out the classifier-selector pair from the previously obtained $\\{(c, \mathbf{w}^{(c)})\ |\  c\in\mathcal{C}\\}$ with the smallest estimation error, whose statistical error is bounded by Theorem 3.1.
>
> The second question is a common question that almost every reviewer has asked about, we refer you to the answer we posted separately for details, where we explain how our approach can be applied to learn sparse linear representations in polynomial time. Our analysis applies to any class that has a robust list-decodable learning algorithm. However it is an open question which classes are learnable in this model -- it is not known whether every VC class has such algorithms.

---

> > ### Comment · Reviewer_ZDsQ · 2024-11-25
> >
> > I thank the authors for their response. I will keep my score.

---

### Official Review · Reviewer_gfkp · 2024-11-03

**Soundness:** 3
**Presentation:** 3
**Contribution:** 2
**Rating:** 6
**Confidence:** 3

**Summary:**

This paper studies conditional classification with homogeneous halfspaces as selectors under the agnostic learning setting with Gaussian marginals. That is, given access to samples from underlying distribution $D$ with adversarial label noise, the algorithm aims to output a classifier-selector(halfspace) pair, such that the classifier makes low prediction error conditioned on the instances in the given halfspace. When there exists a pair $(c^*,v^*)$ that achieves $opt$ error rate, this paper proposes an algorithm that can output a pair $(c,w)$ that achieves error rate of $\tilde{O}(\sqrt{opt})$. The algorithm is based on Projected SGD for choosing the appropriate halfspace and then output the classifier-halfspace pair with minimal loss.

On the other hand, the paper shows that agnostic conditional learning with general halfspace selectors is computationally hard.

**Strengths:**

This paper studies an interesting problem, which considers agnostic classification under a conditional distribution. It models the conditional distribution using a homogeneous halfspace, and complement the result using a hardness example for non-homogeneous halfspace. The paper also links the problem to many other important problems in machine learning, including learning under the list-decodable setting, learning with abstention, etc. It will be interesting to investigate the deep connection between these problems.

The theoretical analysis is generally sound to me. The paper reads smoothly, with results well presented.

**Weaknesses:**

The paper claims that the classifiers are sparse, but it seems the aim is to limit the size of the classifier set. The paper has very limited discussion on this matter, which makes the claim of learning "sparse" classifiers a bit misleading.

Although agnostic conditional learning with general halfspaces is computational hard, the current algorithm for homogeneous halfspace selector also depends on the size of the classifier set $\lvert\mathcal{C}\rvert$. The paper claims that $\lvert\mathcal{C}\rvert$ is small, I wonder how applicable the current algorithm is to the agnostic conditional learning problem for general class of binary classifiers.

The algorithm relies strongly on the Gaussian marginals assumption, such that any halfspace has probability mass of $1/2$. Is there any way we can extend this assumption? Or, the whole analysis only works for Gaussians?

**Questions:**

I didn't get the part of choosing ReLU function as the convex surrogate approximation of the conditional classification error? It seems $L_D(w)$ also blocks the information for one-sided class $y=1$. However, $g_w(x,y)$ seems to be one-sided.

---

> ### Author Response · Authors · 2024-11-18
>
> Thanks for the suggestions! We will respond to the weakness you pointed out and your questions respectively.
>
> We are sorry for the confusion. Since this is a common question that almost every reviewer has asked about, we refer you to the separate common answer for details, where we explain how our approach can also work with sparse linear representations while maintaining polynomial running time in the dimension. The role of sparsity is indeed that the algorithm we describe in Theorem 2.1 (Appendix A) crucially relies on it. It is not so obvious how to extend our method to general classes of binary classifiers.
>
> For the third weakness, the short answer is yes. We recently found that our analysis can be generalized to broader classes of distributions, such as centered log-concave distributions. However, the performance guarantee will also be compromised to $O(\mathrm{opt}^{1/4})$. Also, similar to the case of Gaussian distribution, we heavily rely on the assumption that every homogeneous halfspace cannot be statistically too small, which is satisfied by any centered log-concave distribution. Nonetheless, the reliance of such an assumption is due to the lack of approaches to minimize the conditional classification error $\Pr\\{y = 1\ |\ \mathbf{x}\in h(\mathbf{w})\\}$ directly. That is, the generalized approach still seeks to minimize the joint error $\Pr\\{y = 1\cap \mathbf{x}\in h(\mathbf{w})\\}$, where we can argue that the corresponding conditional classification error cannot differs too much because $\Pr \\{\mathbf{x}\in h(\mathbf{w}) \\}$ cannot be too small.
>
> For your question, we are sorry for the confusion, but the term "one-sided" loss essentially refers to the classification error within the $\mathbf{x}\in h(\mathbf{w})$ side for some halfspace $h(\mathbf{w})$. So, the "one-sided" loss $L_{\mathcal{D}}(\mathbf{w})$ blocks the information of $\mathbf{x}\notin h(\mathbf{w})$. In particular, the most important part of our argument (Proposition 3.2 specifically) is to lower bound the projection of the gradient $\mathbb{E}[g_{\mathbf{w}}]$ onto the optimal normal vector. Observe that our assumptions only give us information on $\Pr\\{y=1\ |\ \mathbf{x}\in h(\mathbf{v})\\}$ and $\Pr\\{y=1\ |\ \mathbf{x}\in h(\mathbf{w})\\}$, therefore, we do want to make sure $\mathbb{E}[g_{\mathbf{w}}]$ is always zero on $\mathbf{x}\notin h(\mathbf{w})$, which, in turn, requires $L_{\mathcal{D}}(\mathbf{w})$ to be zero on $\mathbf{x}\in h(\mathbf{w})$. Therefore, ReLU seems to be a natural choice as our surrogate loss in this context.

---

> ### Comment · Reviewer_gfkp · 2024-11-26
> **Official Comment by Reviewer gfkp**
>
> I thank the authors for their response to address my questions. I will keep my current review.

---

### Author Response · Authors · 2024-11-18
**Answer to The Common Question**

As a critical implication of our results, we show how to generalize our conditional classification approach to work with sparse linear classifiers by combining a robust list-learning algorithm (Theorem 2.1) and our algorithm for finite classes of representations (Theorem 3.1). In particular, if we think of $\mathcal{C}_0$ as the class of sparse linear representations for the task of agnostic conditional classification, then we only need to guarantee that the classifier $c^*\in\mathcal{C}_0$ of the global minimizer $(c^*, h^*)$ (as a classifier-selector pair) for the conditional classification error is also in $\mathcal{C}$ so that we can use our current approach to find a good selector. To obtain such a $\mathcal{C}$, we view $\mathcal{D}$ as a convex combination of the inlier distribution $\mathcal{D}^*$ and outlier distribution $\tilde{\mathcal{D}}$ as in definition 1.3, where the labels of $\mathcal{D}^*$ is generated by $c^*(\mathbf{x})$ while $\tilde{\mathcal{D}}$ can be arbitrary. Now observe that we can refer Theorem 2.1 to obtain a finite list of sparse linear classifier that contains $c^*$, and then use Theorem 3.1 to find a good homogeneous halfspace selector for $c^*$.

As for extending the method to other families of classifiers, this is an open question. While we don't know how to learn either boxes or dense separators, the referenced works on robust clustering (Kothari-Steinhardt-Steurer STOC 2018) yield a kind of learning of Euclidean balls (i.e., the clusters). Polynomial-time algorithms for list-decodable (dense) linear regression (as opposed to classification) have been found [1,2], and we noted in the paper that methods are known for other list-decodable estimation tasks, but not much else is known for learning classifiers at present. It is an interesting broader open question which families classifiers are learnable in the robust list-decodable setting.

[1] S. Karmalkar, A. Klivans, and P. Kothari. List-decodable linear regression. In NeurIPS 2019.

[2] P. Raghavendra and M. Yau. List-decodable learning with sum-of-squares. In SODA 2020.

---

### Meta-Review · Area_Chair_iGd4 · 2024-12-11

**Metareview:**

This work presents a new algorithm for agnostically learning conditional classifiers under Gaussian distribution with a suboptimal error rate. It also shows that under certain cryptographic assumptions, optimal PAC learning of this problem is computationally hard. Reviewers agree that the paper makes important contribution. On the other side, reviewers also have concerns that the results only hold for polynomially finite VC classes and that coverage is not considered. Overall, given that this is a fairly non-trivial learning theory problem, the message from the paper appears inspiring and merits accept.

**Additional Comments On Reviewer Discussion:**

Two crucial questions were raised: 1) how to deal with more general concept classes, 2) how to bound coverage. For the first question, authors provided a workaround via assuming the existence of list-decodable learner; for the second, authors argued that it is relevant but may not be critical to establish coverage guarantee in the current context. Though one reviewer still has concern on 2), overall the reviews are positive.

---

### Decision · Program_Chairs · 2025-01-22

Accept (Poster)